

# TaskBench: Benchmarking Large Language Models for Task Automation

**Yongliang Shen**[1], **Kaitao Song**[2†], **Xu Tan**[2], **Wenqi Zhang**[1],
**Kan Ren**[2], **Siyu Yuan**[3], **Weiming Lu**[1†], **Dongsheng Li**[2], **Yueting Zhuang**[1†]

Zhejiang University[1], Microsoft Research Asia[2], Fudan University[3]
{syl,luwm,zhangwenqi}@zju.edu.cn, syyuan21@m.fudan.edu.cn
{kaitaosong, xuta, dongsli}@microsoft.com

https://github.com/microsoft/JARVIS

## Abstract

In recent years, the remarkable progress of large language models (LLMs) has sparked interest in task automation, which involves decomposing complex tasks described by user instructions into sub-tasks and invoking external tools to execute them, playing a central role in autonomous agents. However, there is a lack of systematic and standardized benchmarks to promote the development of LLMs in task automation. To address this, we introduce TASKBENCH, a comprehensive framework to evaluate the capability of LLMs in task automation. Specifically, task automation can be divided into three critical stages: task decomposition, tool selection, and parameter prediction. To tackle the complexities inherent in these stages, we introduce the concept of Tool Graph to represent decomposed tasks and adopt a back-instruct method to generate high-quality user instructions. We propose TASKEVAL, a multi-faceted evaluation methodology that assesses LLM performance across these three stages. Our approach combines automated construction with rigorous human verification, ensuring high consistency with human evaluation. Experimental results demonstrate that TASKBENCH effectively reflects the capabilities of various LLMs in task automation. It provides insights into model performance across different task complexities and domains, pushing the boundaries of what current models can achieve. TASKBENCH offers a scalable, adaptable, and reliable benchmark for advancing LLM-based autonomous agents [1].

## 1 Introduction

Due to the recent advances of large language models (LLMs) [1, 2, 3, 4, 5, 6], LLM-empowered autonomous agents [7, 8, 9, 10] have unveiled remarkable potential towards artificial general intelligence and become a new rising trend in the realm of AI research. Generally, within the realm of LLM-empowered autonomous agents, task automation is considered as the most important component, which aims to leverage LLMs to autonomously analyze user instructions and accomplish their objectives. Consequently, many researchers attempt to delve deeper into LLM to enable more intelligent task automation [11, 12, 13, 14]. However, it is worth noting that a critical challenge in advancing this area is the lack of a systematic and standardized benchmark to thoroughly evaluate the capability of LLMs in automating tasks. Therefore, creating such a benchmark to facilitate research in this area has become an urgent need.

---

[†] Corresponding author.
[1] https://github.com/microsoft/JARVIS/tree/main/taskbench

38th Conference on Neural Information Processing Systems (NeurIPS 2024) Track on Datasets and Benchmarks.

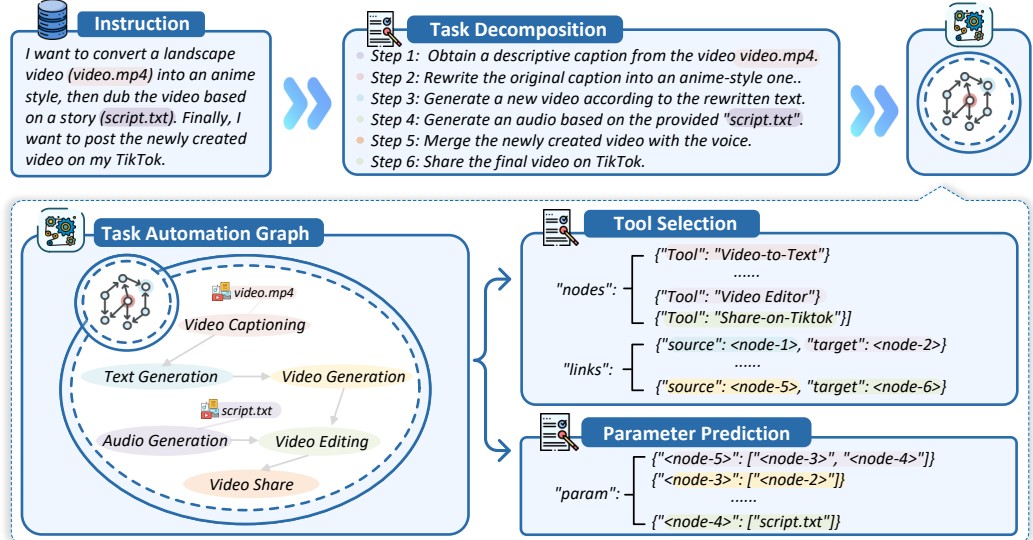

Figure 1: LLM-Based Task Automation: This process involves task decomposition, tool selection, and parameter prediction by LLM-based agents to autonomously complete tasks. Upon receiving a user request, the large language model performs task decomposition, generating a sequence of task steps. Simultaneously, it predicts the tool invocation graph, which encompasses both the selection of appropriate tools and the prediction of their parameters.

Constructing a benchmark for task automation presents unique challenges beyond traditional NLP tasks. As shown in Figure 1, task automation involves multiple stages: task decomposition, tool selection, and parameter prediction. This complexity requires sophisticated evaluation metrics to assess LLM performance comprehensively. Real-world scenarios often feature intricate user instructions with multiple subtasks and tool dependencies, necessitating benchmarks that can simulate this complexity accurately. A key challenge lies in modeling tool dependencies, which is crucial for effective task automation but difficult to capture in both datasets and evaluation metrics. Additionally, generating a high-quality, diverse dataset that reflects real-world scenarios is challenging, as it requires creating instructions that are both natural and aligned with realistic tool usage patterns. These challenges underscore the need for a carefully designed benchmark that can accurately evaluate LLMs' capabilities in task automation across various dimensions and scenarios.

In light of these challenges, we found that existing benchmarks fall short of adequately showcasing the full potential of LLMs in autonomous task completion. Traditional LLM benchmarks, such as MMLU [15], GSM8K [16], and AGIEval [17] assess general knowledge and reasoning skills. Recent tool-augmented LLM benchmarks have made significant strides but still fall short in various aspects. APIBench [18], ToolBench [19], and MetaTool [20] have advanced the evaluation of LLMs' tool usage capabilities, but they often lack comprehensive assessment across all stages of task automation. For instance, MetaTool primarily focuses on tool selection, while ToolQA [21] emphasizes question-answering accuracy. Similarly, benchmarks like ToolAlpaca [22] and API-Bank [23] offer valuable insights but are limited in their evaluation scope and tool complexity modeling.

To address these limitations, we introduce TASKBENCH, a comprehensive framework for evaluating LLMs' capabilities in task automation. Our approach integrates innovative data generation techniques with rigorous evaluation methods. We begin by introducing the concept of a Tool Graph (TG), which represents relationships and dependencies between various tools, overcoming the limitations of simple API documentation or template-based approaches used in existing benchmarks. Based on this Tool Graph, we employ a back-instruct strategy to generate high-quality user instructions from sampled subgraphs, ensuring diverse and realistic task scenarios. To guarantee the naturalness and utility of the dataset, we implement a multi-step quality control process, including structured sampling, self-critic mechanisms (both LLM-based and rule-based), and extensive human verification. These methods significantly enhance data quality, surpassing the verification approaches used in most existing benchmarks. Finally, we propose TASKEVAL, a multi-faceted evaluation methodology that

| Benchmark | TaskBench | APIBench | ToolBench | MetaTool | API-Bank |
|---|---|---|---|---|---|
| **Data Generation** | Tool Graph + Back-Instruct | API Doc | Manual + LLM | Template Generation (Diverse, Keyword, Emotional) | API Doc |
| **Tool Dependency** | ✓ | ✗ | ✗ | ✗ | ✗ |
| **Quality Control** | LLM Self-critique + Rule-based Self-critique + Human Verification | Human Verification | Human Verification | Human Verification | Auto-generation + Human Verification |
| **Evaluation** | Task Decomposition + Tool Selection + Parameter Prediction | Tool Selection | Tool Selection + Parameter Prediction | Tool Usage Awareness + Tool Selection | API call correctness, response quality |
| **Tool Complexity** | Single tool to complex tool graph structures | Mainly single tool | Single to multi-tool | Single to multi-tool | Single to multi-tool |
| **Dataset Scale** | 17,331 samples | 2,365 samples | 12,657 samples | 21,127 samples | 6,135 interactions |

Table 1: Comparison of TaskBench with Selected Tool-Augmented LLM Benchmarks

comprehensively assesses LLM performance across task decomposition, tool selection, and parameter prediction stages, offering a more thorough evaluation than existing tool-augmented benchmarks.

Our contributions can be summarized as follows:

- We introduce TASKBENCH, a novel benchmark for evaluating LLMs in task automation, featuring an innovative data generation process that addresses the data deficiency and limitations in existing tool-augmented LLM benchmarks.

- We present TASKEVAL, a comprehensive evaluation framework that quantitatively assesses LLMs' capabilities in task decomposition, tool selection, and parameter prediction, providing more nuanced insights than the metrics used in current benchmarks.

- The experimental results on different LLMs and additional dataset analysis demonstrate that our proposed TASKBENCH can effectively reflect the capability of LLMs in multiple dimensions with the support of TASKEVAL and show high correlations with human evaluation.

## 2   Related Works

Large language models (ChatGPT [2], GPT-4 [4], LLAMA [5, 24], Bard [6]) have drawn the development of autonomous agents (e.g., AutoGPT [7], HuggingGPT [8], BabyAGI [9]). These applications can be considered as a form of task automation, which uses LLMs as the controller to analyze user instructions and search for the most suitable solution (e.g., external models) to obtain answers. With the increasing demand for advanced task automation, numerous benchmarks have emerged to assess the capability of large language models LLMs to effectively interact with external tools. However, these benchmarks vary significantly in terms of data generation, tool dependency modeling, quality control mechanisms, and evaluation, as shown in Table 1.

Several benchmarks assess the interaction between LLMs and tools, primarily focusing on tool usage and API calls. APIBench [18] and ToolBench [19] use API documentation to generate tasks, but their reliance on template-based sampling can limit the logical consistency of generated tasks. MetaTool [20] introduces template-based tool generation for decision-making, focusing on whether a tool is needed, though it lacks support for modeling complex tool dependencies. ToolAlpaca [22]

explores self-instruct to generate tasks but does not adequately address dependencies between tools. Similarly, AgentBench [25] evaluates models across simulated environments but focuses more on agent-like behavior than tool interactions. Other benchmarks, like ToolAlpaca [22], ToolQA [21] and GPT4Tools [26], concentrate on task completion or QA correctness in specific domains. While these works explore how well models complete tasks by consulting external tools, they rely heavily on human verification or simple template-based generation, potentially limiting their scalability and diversity.

Unlike prior benchmarks that rely solely on API documentation or template-based task generation, TaskBench introduces a Tool Graph to model real-world dependencies between tools, simulating complex tool interactions more accurately. Furthermore, the Back-Instruct strategy aligns tool subgraphs with instructions, significantly reducing the risk of hallucination and improving data authenticity. Its unique tool graph approach, focus on dependency modeling, and multi-domain coverage make it a more reliable, practical, and scalable benchmark for evaluating LLM capabilities in complex task automation scenarios.

# 3 TaskBench

In this section, we introduce the construction of TASKBENCH, the benchmark meticulously designed to facilitate the development of LLMs in task automation. Specifically, unlike previous methods which use collection or instruction methods, TASKBENCH can consider the complex relationships among multiple tasks to simulate more practical and complex user instruction. Figure 2 illustrates the entire process of our method to build the datasets. More details will be introduced in the following subsections.

## 3.1 Preliminary: Tool Graph

Task automation in real-world scenarios often involves complex user instructions encompassing multiple sub-tasks, each potentially requiring the invocation of specific tools [27]. These sub-tasks may have temporal or resource dependencies. To capture this complexity, we introduce the concept of the Tool Graph (TG). A TG is formally defined as $\mathcal{G} = T, D$, where $T = t_1, t_2, \ldots, t_n$ represents a collection of tools, and $D$ is a set of dependencies $(t_a, t_b)$ indicating that tool $t_a$ depends on tool $t_b$. This structure offers a more effective way to organize tools and their relationships compared to traditional taxonomy trees used in [19]. In the next subsection, we will introduce how to build a tool graph and utilize it to formulate our benchmark.

## 3.2 Dataset Construction

To accomplish user intent, LLMs usually adopt a stepwise process (e.g., task decomposition→tool selection→parameter prediction) to analyze the user request and convert it into multiple executable tasks. Therefore, it is essential to construct the dataset and allow LLMs to evaluate their automation capability in the above process.

To guarantee that the generated user instructions could cover the expected tasks and dependencies, we adopt a back-instruct strategy to simulate data. More specifically, it can summarized as three steps: 1) we first collect a tool repository and build a tool graph $\mathcal{G}$ with a collection of tools and their dependencies; 2) then we sample a sub-graph from $\mathcal{G}$, to obtain a specified structure; 3) based on the sampled tool sub-graph, we use LLMs to generate user instruction via back-instruct. More details are introduced as below:

### 3.2.1 Tool Graph Construction

Building a tool graph requires us to collect many standalone tools from different sources. When combining different tools together, the dependencies among tools could be diverse, encompassing resource dependencies, temporal dependencies, environment dependencies, and so on. In our research, we mainly investigate two of them: resource and temporal dependencies. For the former one, it means the two tools can have a connection if the input type of tool $t_a$ can match the output type of tool $t_b$. For the latter one, we devise tool graphs that highlight temporal dependencies, allowing any two tools

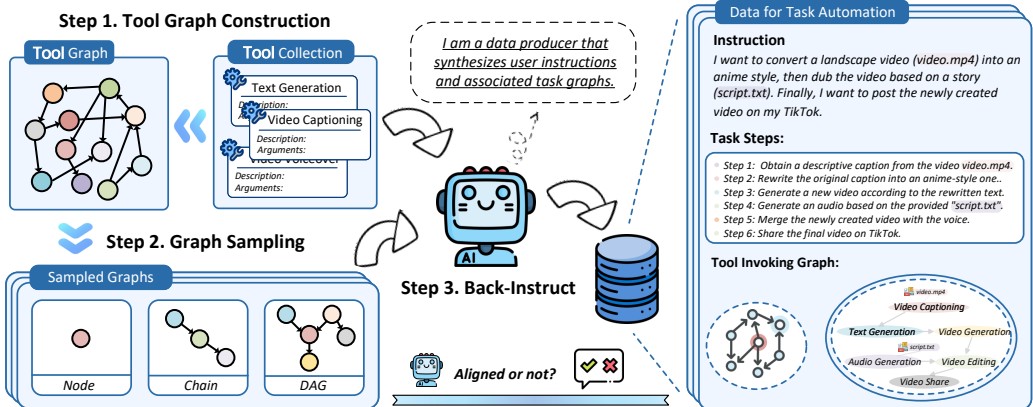

Figure 2: Construction of the TASKBENCH: Initially, the toolbox is transformed into a tool graph by establishing connections based on tool dependencies. We then sample diverse subgraphs from this tool graph, which can be individual nodes, linear chains, or directed acyclic graphs (DAGs). Using these sampled tool subgraphs, we "back-instruct" the LLM to inversely craft user instructions, task steps, and tool invocation graphs. Additionally, we employ critics to assess the consistency between the generated tool invocation graphs and the corresponding sampled tool subgraphs.

to be linked to illustrate their order. In this work, we choose three scenarios to build the datasets for our benchmark[2]:

**Hugging Face**   Hugging Face [28] provides a wide variety of AI models to cover massive tasks across language, vision, audio, video, and so on. Each task defined by Hugging Face can be viewed as a tool to address a specific task. Specifically, each tool in Hugging Face has determined the type of its input and output. Hence, if tool $t_a$ and $t_b$ have a connection, the input type of $t_a$ should match the output type of $t_b$. Guided by this principle, we constructed Hugging Face's tool graph, comprising 23 tools and 225 edges.

**Multimedia**   In contrast to the Hugging Face tools, which are tailored for AI tasks, the multimedia tools is broader in scope. It provides more user-centric tools like file downloader, video editor, and so on. The policy for tool connections is the same as the Hugging Face domain. Finally, we could construct a tool graph over multimedia tools with 40 nodes and 449 edges.

**Daily Life APIs**   Sometimes, we also need some daily life services, including web search, shopping, and etc. Hence, these daily life APIs can also be considered as tools for specific tasks. However, it is worth noting that the type of dependencies among these APIs is predominantly temporal. Therefore, two daily life APIs have a successive order if they are connected. In this scenario, we can build a tool graph with 40 nodes and 1,560 edges.

### 3.2.2   Sampling on Tool Graph

Based on the above steps, we can sample a sub-graph from the constructed TG and keep the connections of sampled tools from the TG to capture the dependencies between tools. Following the setting of HuggingGPT, we categorize the sub-structure of a TG into three types: node, chain, and directed acyclic graph (DAG). Each type embodies a specific pattern for tool invocation:

- **Node** represents standalone tool invocations, suitable for addressing simple tasks necessitating only a single tool.
- **Chain** corresponds to sequential tool invocations, where tools need to be stepwise executed to complete a task.
- **DAG** depicts more intricate tool invocations. A tool might rely on multiple preceding tools or influence several subsequent tools.

---

[2]We present more details about these tool graphs on different domains in Appendix A.8

By sampling sub-graphs from these three substructures, we can emulate a variety of valid tool invocation patterns for user instruction. We represent the tool subgraph in $\mathcal{G}$ as $\mathcal{G}_s = \{T_s, D_s\}$, where $T_s = \{t_{s1}, t_{s2}, \ldots, t_{sk}\}$ with $k < n$ and $D_s = \{(t_{sa}, t_{sb})\}$, such that $t_{sa}$ and $t_{sb}$ belong to $T_s$. The sampling of the tool graph can be described as:

$$\text{Sample}(\mathcal{G}, \text{mode}, \text{size}) \rightarrow \mathcal{G}_s, \tag{1}$$

where the mode specifies the sampling mode (e.g., Nodes, Chains, DAGs), and the size indicates the number of tools (Here we set its range as $\{1, 2, ..., 10\}$). These factors determine the topological nature and magnitude of the tool sub-graph in user instructions, respectively.

### 3.2.3 Back-Instruct

Next, based on the sampled sub-graph $\mathcal{G}_s$, we use LLMs to synthesize user instructions. We term this process BACK-INSTRUCT, which can considered as a data engine to convert the sampled tools into user instruction. Specifically, given a sampled subgraph $\mathcal{G}_s$, we formulate the following BACK-INSTRUCT procedure, empowering LLMs to produce the corresponding instructions $I$:

$$\text{BackInstruct}_1(\mathcal{G}_s = (T_s, D_s)) \rightarrow I. \tag{2}$$

Here, the sampled sub-graph $\mathcal{G}_s$ can instruct LLMs to generate user requests covering these related sub-tasks, and further with their dependencies. Such a strategy ensures the complexity and quality of the generated data.

Specifically, we note that sampled sub-graphs can only provide information on tool invocation skeletons, lacking the critical parameters for tool execution. Therefore, based on the generated instruction $I$ in Eqn. 2, we let the LLM to populate the parameters for the tool subgraphs and generate the final tool invocation graph $\bar{\mathcal{G}}$ along with the corresponding task decomposition steps $P$:

$$\text{BackInstruct}_2(\mathcal{G}_s = (T_s, D_s), I) \rightarrow \{P, \bar{\mathcal{G}}\}. \tag{3}$$

### 3.2.4 Quality Control Mechanisms

To ensure high-quality data, we implement a multi-step quality control process:

**Self-Critic Mechanisms:** A we introduce a self-critic mechanism to check and filter out the generated instruction to guarantee quality. Here, we offer two variants: LLM-based and rule-based. The former aims to use LLM to check the alignments between the generated data and the sampled tool sub-graph. While the latter uses straightforward rules to determine the alignment between the tool graphs in created data and the sampled tool graphs. Here, we use the nodes and edges of the sampled graph to determine the consistency. Figure 2 illustrates each step of our data engine to simulate user instructions.

**Human Verification:** We implemented a rigorous human verification process to further ensure the quality and coherence of the dataset. Human experts reviewed the generated instructions across different task complexities and confirmed their logical consistency and alignment with the intended tasks. Before beginning the verification process, we provided detailed instructions and conducted a calibration session for all experts. This session was essential to standardize the evaluation criteria across different reviewers and ensure consistency in their assessments. During this calibration, we provided sample cases and discussed the scoring criteria (naturalness, complexity, alignment) to align the understanding among the experts. This verification process covered instructions generated from sub-graphs of varying sizes, ensuring that the instructions are meaningful and practical. This comprehensive review provides additional assurance that the dataset meets high standards of quality.

Based on the above steps, we build TASKBENCH across three domains, which use GPT-4 as the data engine. The ratio of different modes (i.e., Node, Chain, DAG) is set as $3 : 7 : 8$ for sampling and the ratio for the number of different tools is set as $\{0.1, 0.2, 0.3, 0.2, 0.1, 0.05, 0.025, 0.025, 0.025\}$. More detailed designs about our data engine and the statistics of the constructed datasets are provided in the Appendix A.5.

### 3.3 Evaluation of the Dataset Quality

To illustrate the quality of the TASKBENCH datasets, we conducted comprehensive human evaluations based on generated samples. Additionally, we performed a case study and an error analysis of the datasets. For further details, please refer to Appendix A.1 and Appendix A.2.

**Evaluation Metrics** To evaluate the quality of datasets constructed by Back-Instruct, we designed three metrics: two metrics (i.e., Naturalness and Complexity) to assess the quality of the instructions, and one metric (i.e., Alignment) to evaluate the tool invocation graph. More details are in below:

- **Naturalness:** This metric evaluates how reasonable the instructions are, considering factors such as the typicality of dependencies between tools and their relevance to real-world applications.
- **Complexity:** This metric assesses the complexity of the instructions by examining aspects such as the depth of the task, the number of tools involved, and the interrelations among these tools.
- **Alignment:** This measures how well the tool invocation graphs align with the instructions, specifically evaluating whether the graphs effectively fulfill the user's commands.

Each metric is scored on a scale from 1 to 5. These metrics are designed to assess the effectiveness and faithfulness of our TASKBENCH in task automation.

**Comparison with Baselines** To make a fair comparison, we choose two additional baselines to compare our Back-Instruct:

- **Back-Instruct (Ours):** e sample tool subgraphs and then backtranslate to instructions and further refine the tool invocation graph.
- **Back-Instruct w/o edges:** Unlike our complete Back-Instruct, this version omits edges from the sampled tool subgraphs, retaining only the tool node information in the prompts.
- **Self-Instruct:** As described by [29], this method uses manually labeled demonstrations and descriptions of all tools. We utilize GPT-4 to autonomously select tools and generate instructions along with tool invocation graphs.

Table 2: Human evaluation (rating from 1 to 5) on samples constructed by different methods. Average score rating from three human experts.

| Methods | Naturalness↑ | Complexity↑ | Alignment↑ | Overall↑ |
|---|---|---|---|---|
| Back-Instruct | **3.89** | **4.01** | **3.66** | **3.85** |
| Back-Instruct w/o edges | 3.44 | 3.27 | 3.62 | 3.44 |
| Self-Instruct | 2.18 | 2.01 | 3.64 | 2.61 |

**Evaluation Results** During the human evaluation, we randomly selected 50 samples from our TASKBENCH and invited three domain experts to assess their quality. To ensure a fair and unbiased evaluation, all samples were anonymized. We provided canonical samples to help these experts calibrate their assessment criteria during the annotation process. The final results are the average scores from all experts' ratings, and are detailed in Table 2.

We observed that all methods (Self-Instruct and Back-Instruct) effectively ensured alignment. However, our method, Back-Instruct, achieved the highest scores in Naturalness and Complexity. We attribute these superior results to the realistic resource and temporal dependencies in our sampled tool subgraphs, which enable us to generate more natural and complex instructions, especially in scenarios involving multiple tools. This graph structure guides the generation process, resulting in more natural and complex instructions that reflect realistic task scenarios. The performance difference between back-instruct with and without edges further underscores the importance of capturing tool dependencies. Including edge information in the Tool Graph allows for a more comprehensive understanding of tool relationships, resulting in more natural and complex instructions.

## 4  TaskEval

To comprehensively evaluate LLMs' capabilities in task automation, we introduce TASKEVAL, a systematic evaluation framework that assesses three critical aspects: task decomposition, tool selection, and parameter prediction. Unlike existing benchmarks that focus on isolated aspects of tool usage or API interactions, TASKEVAL provides a holistic assessment of the entire task automation process. To ensure standardized evaluation, we employ a consistent prompting strategy that guides each model through a structured sequence: first decomposing user requests into sub-tasks, then

Table 3: Evaluation for tool selection. Node F1 *(n-F1)* and Edge F1 *(e-F1)* for node and edge prediction. For nodes, a prediction is deemed positive if the predicted node's ID aligns with any of the ground-truth node labels. For edges, both the source and target nodes of a predicted edge must correspond exactly. Normalized Edit Distance *(NED)* measures the normalized number of operations required to correct the prediction for chain structure.

| | **TOOL SELECTION - Predicts tools and their dependencies.** | | | | | | | |
|---|---|---|---|---|---|---|---|---|
| | **Node** | **Chain** | | | **DAG** | | **Overall** | |
| **LLM** | *n-F1* ↑ | *n-F1* ↑ | *e-F1* ↑ | *NED* ↓ | *n-F1* ↑ | *e-F1* ↑ | *n-F1* ↑ | *e-F1* ↑ |
| **Hugging Face Tools** | | | | | | | | |
| gpt-4 | 84.34 | 80.79 | 55.73 | 39.70 | 82.86 | 56.39 | 81.54 | 54.70 |
| gemini-pro | 77.46 | 76.12 | 45.51 | 43.10 | 79.05 | 49.36 | 76.62 | 43.50 |
| claude-2 | 69.83 | 80.67 | 48.11 | 40.03 | 84.52 | 53.40 | 79.00 | 43.51 |
| gpt-3.5-turbo | 56.91 | 72.63 | 39.92 | 46.52 | 73.79 | 38.55 | 69.49 | 33.36 |
| text-davinci-003 | 40.71 | 66.05 | 36.04 | 48.57 | 64.64 | 34.19 | 59.38 | 29.37 |
| mistral-7b-v0.3 | 60.74 | 67.00 | 25.70 | 52.74 | 68.55 | 26.37 | 65.96 | 21.91 |
| codellama-13b | 43.68 | 55.65 | 17.80 | 62.23 | 52.87 | 13.19 | 53.16 | 14.64 |
| nous-hermes-13b | 58.66 | 52.39 | 9.01 | 62.48 | 51.99 | 6.33 | 53.62 | 8.29 |
| vicuna-13b-v1.5 | 51.74 | 50.37 | 8.40 | 66.83 | 52.46 | 9.06 | 50.82 | 7.28 |
| llama-2-13b-chat | 43.59 | 49.87 | 8.22 | 64.99 | 49.60 | 9.11 | 48.47 | 7.30 |
| **Multimedia Tools** | | | | | | | | |
| gpt-4 | 97.13 | 89.70 | 69.29 | 28.93 | 92.32 | 71.64 | 90.90 | 69.27 |
| gemini-pro | 73.61 | 82.65 | 55.50 | 35.62 | 85.29 | 57.80 | 81.54 | 52.07 |
| claude-2 | 66.16 | 83.95 | 59.22 | 33.41 | 82.98 | 54.28 | 80.94 | 53.01 |
| text-davinci-003 | 59.15 | 76.87 | 50.79 | 38.54 | 79.00 | 50.69 | 73.97 | 45.81 |
| gpt-3.5-turbo | 53.55 | 76.81 | 50.30 | 39.05 | 78.65 | 49.52 | 72.83 | 44.02 |
| mistral-7b-v0.3 | 64.00 | 78.32 | 41.12 | 40.75 | 79.96 | 41.36 | 76.11 | 35.34 |
| codellama-13b | 43.70 | 66.89 | 28.77 | 46.35 | 68.68 | 28.79 | 62.78 | 24.61 |
| vicuna-13b-v1.5 | 66.64 | 59.18 | 16.49 | 54.17 | 61.40 | 13.95 | 60.61 | 14.78 |
| nous-hermes-13b | 60.58 | 58.53 | 9.47 | 56.02 | 59.39 | 9.57 | 58.97 | 8.90 |
| llama-2-13b-chat | 38.02 | 45.14 | 1.62 | 65.29 | 45.95 | 2.11 | 43.87 | 1.63 |
| **Daily Life APIs** | | | | | | | | |
| gpt-4 | 95.97 | 97.06 | 83.47 | 38.69 | 96.41 | 42.01 | 96.91 | 80.53 |
| claude-2 | 79.57 | 95.36 | 80.68 | 39.93 | 93.85 | 41.04 | 93.52 | 75.31 |
| gemini-pro | 76.15 | 92.79 | 64.58 | 41.64 | 89.68 | 28.42 | 90.75 | 59.45 |
| gpt-3.5-turbo | 52.18 | 90.80 | 70.66 | 43.50 | 86.94 | 30.85 | 85.37 | 60.67 |
| text-davinci-003 | 68.49 | 82.15 | 60.12 | 47.14 | 76.81 | 24.54 | 80.42 | 54.90 |
| codellama-13b | 89.75 | 87.80 | 65.92 | 44.42 | 83.61 | 27.47 | 87.73 | 63.16 |
| mistral-7b-v0.3 | 81.55 | 80.52 | 50.95 | 51.80 | 79.17 | 25.04 | 80.54 | 45.87 |
| vicuna-13b-v1.5 | 80.59 | 73.74 | 13.24 | 51.43 | 67.92 | 5.62 | 75.67 | 12.48 |
| nous-hermes-13b | 82.50 | 71.17 | 3.55 | 53.47 | 70.65 | 2.86 | 73.45 | 3.50 |
| llama-2-13b-chat | 34.11 | 57.61 | 20.13 | 67.06 | 56.18 | 8.42 | 55.77 | 17.02 |

selecting appropriate tools with parameters, and finally constructing a complete tool invocation graph. For our evaluations, we primarily focus on the GPT series [1, 2, 4], Gemini [3], Claude [30] and open-source LLMs [5, 31, 32, 33, 34, 35]. For comprehensive evaluations of other open-source LLMs [36, 37, 38], please refer to Appendix A.7.

## 4.1 Task Decomposition

To evaluate LLMs' ability to understand and break down complex tasks, we assess the quality of task decomposition through three complementary ROUGE metrics: **Rouge-1 *(R1)***, **Rouge-2 *(R2)***, and **Rouge-L *(RL)***. Our analysis, presented in Table 12, reveals several key findings: (1) GPT-4 consistently demonstrates superior task decomposition abilities, achieving approximately 10% higher scores in both R1 and R2 compared to other models. This performance gap indicates its enhanced capability to understand and structure complex tasks. (2) Codellama-13b shows particular strength in the "Daily Life APIs" domain, achieving scores of 89.86 in R1 and 83.27 in R2, suggesting that code-centric pre-training enhances the ability to understand structured task sequences. (3) The performance gap between models widens as task complexity increases, indicating that advanced reasoning capabilities become more crucial for complex task decomposition.

Table 4: Evaluation for parameter prediction of tools. *t-F1* evaluate the pair of (task, parameter name), *v-F1* evaluate the triple of (task, parameter name, parameter value).

| | | **Node** | | **Chain** | | **DAG** | | **Overall** | |
|---|---|---|---|---|---|---|---|---|---|
| | **LLM** | *t-F1*↑ | *v-F1*↑ | *t-F1* ↑ | *v-F1*↑ | *t-F1* ↑ | *v-F1*↑ | *t-F1* ↑ | *v-F1*↑ |
| **Hugging Face Tools** | gpt-4 | 80.05 | 74.10 | 76.66 | 58.15 | 78.24 | 60.03 | 77.31 | 60.86 |
| | gemini-pro | 67.63 | 56.54 | 66.60 | 46.35 | 70.41 | 50.56 | 67.12 | 48.54 |
| | claude-2 | 48.07 | 32.14 | 66.35 | 45.57 | 68.59 | 48.19 | 63.00 | 43.08 |
| | text-davinci-003 | 38.51 | 27.43 | 56.90 | 38.76 | 57.03 | 38.90 | 52.53 | 36.04 |
| | gpt-3.5-turbo | 37.70 | 19.81 | 60.96 | 41.15 | 61.33 | 42.89 | 55.88 | 36.32 |
| | mistral-7b-v0.3 | 29.18 | 13.19 | 46.18 | 26.09 | 45.49 | 28.73 | 42.41 | 23.40 |
| | codellama-13b | 20.09 | 12.58 | 36.40 | 21.31 | 33.43 | 20.48 | 32.06 | 18.87 |
| | nous-hermes-13b | 46.38 | 31.06 | 35.55 | 13.81 | 33.06 | 13.69 | 37.51 | 17.66 |
| | vicuna-13b-v1.5 | 29.80 | 20.54 | 32.14 | 13.57 | 32.16 | 15.23 | 31.61 | 15.38 |
| | llama-2-13b-chat | 25.71 | 13.11 | 28.99 | 11.14 | 30.04 | 13.60 | 28.34 | 11.85 |
| **Multimedia Tools** | gpt-4 | 95.64 | 87.12 | 85.60 | 69.83 | 87.57 | 72.79 | 87.06 | 72.31 |
| | gemini-pro | 62.21 | 50.48 | 72.99 | 55.21 | 76.13 | 58.79 | 71.67 | 54.82 |
| | claude-2 | 53.81 | 24.02 | 75.60 | 58.12 | 72.41 | 52.43 | 71.63 | 51.58 |
| | gpt-3.5-turbo | 44.94 | 11.96 | 70.53 | 47.76 | 71.82 | 47.95 | 65.91 | 40.80 |
| | text-davinci-003 | 60.30 | 20.78 | 69.91 | 44.76 | 71.91 | 45.76 | 68.48 | 40.70 |
| | mistral-7b-v0.3 | 30.70 | 14.65 | 61.42 | 41.79 | 62.32 | 42.93 | 55.52 | 36.40 |
| | codellama-13b | 32.01 | 16.10 | 52.30 | 32.51 | 53.08 | 33.79 | 48.19 | 29.13 |
| | vicuna-13b-v1.5 | 52.72 | 35.55 | 39.31 | 21.00 | 40.05 | 21.40 | 41.62 | 23.62 |
| | nous-hermes-13b | 50.11 | 37.80 | 41.98 | 17.89 | 43.99 | 20.04 | 43.60 | 21.69 |
| | llama-2-13b-chat | 28.49 | 17.01 | 30.26 | 9.66 | 31.00 | 11.35 | 29.99 | 11.32 |
| **Daily Life APIs** | gpt-4 | 95.83 | 76.21 | 97.23 | 70.67 | 95.95 | 69.65 | 97.02 | 71.14 |
| | claude-2 | 78.12 | 59.43 | 94.72 | 65.30 | 91.83 | 66.39 | 92.71 | 64.72 |
| | gemini-pro | 69.88 | 45.41 | 91.66 | 57.93 | 88.50 | 53.91 | 88.95 | 56.22 |
| | gpt-3.5-turbo | 43.81 | 28.77 | 89.21 | 61.11 | 83.88 | 56.13 | 81.97 | 55.66 |
| | text-davinci-003 | 61.68 | 45.53 | 80.68 | 54.54 | 76.51 | 51.91 | 78.37 | 53.40 |
| | codellama-13b | 86.34 | 71.20 | 84.31 | 61.51 | 80.42 | 60.18 | 84.26 | 62.38 |
| | mistral-7b-v0.3 | 65.86 | 50.67 | 72.03 | 49.71 | 70.52 | 48.35 | 71.21 | 49.73 |
| | vicuna-13b-v1.5 | 83.63 | 67.71 | 61.80 | 44.54 | 57.14 | 41.72 | 64.27 | 47.31 |
| | nous-hermes-13b | 79.69 | 63.29 | 62.64 | 45.32 | 63.26 | 45.74 | 64.47 | 47.22 |
| | llama-2-13b-chat | 10.39 | 7.32 | 38.89 | 25.37 | 36.43 | 23.40 | 35.11 | 22.94 |

## 4.2 Tool Selection

To assess LLMs' ability to select and connect appropriate tools, we evaluate their construction of tool invocation graphs, where nodes represent individual tools and edges capture dependencies between them. We introduce three complementary metrics designed to capture different aspects of tool selection accuracy: (1) **Node F1 (n-F1)**: Evaluates the accuracy of tool selection by comparing the predicted tools with the reference set, measuring the model's ability to identify appropriate tools for each sub-task. (2) **Edge F1 (e-F1)**: Assesses the model's understanding of tool dependencies by comparing the predicted connections between tools with the reference graph, capturing the ability to understand tool interaction patterns. (3) **Normalized Edit Distance (NED)**: Specifically measures the sequential accuracy in chain structures, evaluating how well models understand and maintain the correct order of tool operations.

Results in Table 3 reveal several important patterns: (1) Edge prediction consistently proves more challenging than node prediction, with F1 score differences of approximately 20% across all models, indicating that understanding tool relationships is more complex than identifying individual tools. (2) Performance varies significantly with task structure complexity - while open-source models like Mistral-7b and CodeLlama-13b compete well with GPT-3.5-Turbo on simpler node structures, they show notable limitations when handling more complex dependencies. (3) GPT-4 maintains more consistent performance across different structure types, suggesting better generalization to complex tool interactions.

### 4.3 Parameter Prediction

To evaluate LLMs' ability to correctly configure tools for execution, we assess parameter prediction through two comprehensive metrics: (1) **Parameter Name F1 (t-F1)**: Measures the accuracy in identifying required parameters for each tool, evaluating the model's understanding of tool specifications; (2) **Parameter Name & Value F1 (v-F1)**: Evaluates both parameter identification and value assignment accuracy, assessing the model's ability to provide correct and contextually appropriate parameter values.

The results are detailed in Table 4. GPT-4 demonstrates remarkable robustness in capturing the nuances of both parameter names and values, essential for precise task execution. LLMs such as Claude-2 and Gemini-Pro show competitive results in some domains but still fall short of the benchmarks set by GPT-4. In contrast, open-source LLMs, while performing adequately in some categories, generally exhibit lower *v-F1* scores. This discrepancy highlights a critical area for improvement in task automation capabilities, particularly in the precision of parameter prediction. This insight points to the need for advancements in model training to enhance the effectiveness of LLMs in real-world applications.

### 4.4 Analysis

Our comprehensive evaluation reveals several key factors that influence task automation performance:

**Fundamental Capabilities**    1) *Reasoning*: The success of LLMs in task automation largely depends on their ability to solve complex problems and reason effectively. For instance, the GPT series exhibits superior reasoning skills in mathematical and coding tasks, indicative of its robust capabilities in task planning and tool usage. 2) *Instruction Following*: Models specifically fine-tuned for instruction following, such as Vicuna-13b and WizardLLM-13b, tend to outperform others like Llama-2-13b. Notably, WizardLLM-13b exhibits a marked improvement over Vicuna-13b, highlighting the impact of sophisticated instruction fine-tuning on performance.

**Contributing Factors**    1) *Code Pre-training*: Models with extensive code pre-training, such as Code-Llama, surpass other LLMs in task automation. Our data shows an average improvement of 4.45% in tool prediction and 12.76% in parameter prediction across various domains, underscoring the necessity of structured text for connecting automation stages. 2) *Alignment*: Models employing human alignment techniques (e.g., the GPT series with RLHF) show enhanced task automation capabilities compared to their open-source counterparts, indicating that RLHF promotes more generalized reasoning skills and mitigates instruction-specific overfitting.

### 4.5 Consistency with Human Evaluation

To validate TASKEVAL's effectiveness as a benchmark, we analyze its correlation with human evaluations using two statistical measures: Kendall's $\tau$ and Spearman's $\rho$. Our results in Table 9 demonstrate strong correlations (average $\tau = 0.89$, $\rho = 0.78$), confirming that our automated metrics align well with human judgment of task automation quality.

## 5  Conclusion

In this paper, we introduce TASKBENCH, a benchmark designed to evaluate the performance of LLMs in task automation. We begin by outlining the three critical stages of task automation for LLMs: task decomposition, tool selection, and tool parameter prediction. The performance in these stages reflects the overall capability of LLMs in task automation, motivating the construction of specialized evaluation datasets. To this end, we present the concept of Tool Graph, which aggregates various tools along with their interconnections. Using our curated datasets, we further introduce TASKEVAL, which comprises systematic evaluation metrics for task automation. The experimental results reveal the performance of current mainstream LLMs in task automation and analyze the factors influencing their autonomous task execution. The results also validate the effectiveness of TASKBENCH in assessing LLMs' performance in task automation. Looking forward, we plan to expand our benchmark to include more domains and develop more advanced metrics to further explore the potential of LLMs in task automation and the development of powerful autonomous agents.

## Acknowledgement

This work is supported by National Natural Science Foundation of China (No. 62436007), the National Natural Science Foundation of China (No. 62376245), the "Pioneer" and "Leading Goose" R&D Programs of Zhejiang (No. 2024C01034), the Fundamental Research Funds for the Central Universities (226-2024-00170), the project of the Donghai Laboratory (Grant no. DH-2022ZY0013), MOE Engineering Research Center of Digital Library and ZJU Kunpeng&Ascend Center of Excellence.

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

# A  Appendix

## A.1  Case Study of Back-Instruct

We draw some cases to intuitively show the differences between Back-Instruct, Back-Instruct w/o edges and Self-Instruct [29], as shown in Table 5. From these examples, we observe that our Back-Instruct can generate examples with more comprehensive and interconnected tool usage, reflecting higher naturalness and complexity in instruction generation.

## A.2  Error Analysis

### A.2.1  Error Analysis on TaskBench Dataset

Despite the advanced instruction generation and labeling capabilities of gpt-4, we admit that it is challenging to guarantee the correctness of all generated samples. To better understand our dataset and assess its accuracy, we conduct human evaluations to provide a thorough error analysis. Here, we first randomly sampled 148 samples, and our labeling team identified 18 error samples (nearly 12%) from the sampled data. We attribute these incorrect samples to five distinct error categories. Typical examples and the proportions for each category are shown in Table 6 and Figure 3:

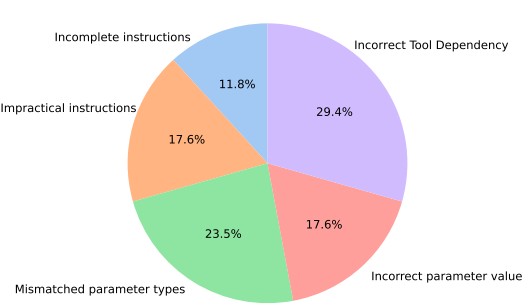

Figure 3: Error Analysis on TASKBENCH

- **Incorrect Instructions**:
  - **Incomplete instructions**: This error occurs when the instructions lack the necessary details or resources for successful completion.
  - **Impractical instructions**: The instructions could be irrational or impossible to execute with the capabilities of current tools.
- **Parameter Errors:**
  - **Mismatched parameter types**: This error occurs when the parameters provided do not match the expected types for the used tool.
  - **Incorrect parameter value**: This error is evident when the values provided for the parameters are incorrect or not suitable for the task.
- **Incorrect Tool Dependency**: This error type refers to the incorrect linking or sequencing of tools required for a task.

Based on these observed errors, we conclude that it is necessary to build a more elaborate prompt (e.g., more detailed tool-use specification and demonstrations) to describe tool parameters and tool dependencies when generating the tool invocation graph. Besides, we will also introduce more high-quality criteria to continuously improve our dataset in addition to our rule-based and LLM-based critics.

### A.2.2  Error Analysis of Different LLMs in Predicting Tool Invocation Graph

We analyze the failures in predicting the tool invocation graph that occur during task automation inference. These failures can be categorized into three main groups: incorrect tool names, incorrect tool dependencies, and mismatched tool parameters. For our analysis, we randomly selected 50 predictions, and the distribution of each error type across different LLMs is detailed in Table 7. We observed that:

- gpt-4 demonstrates the fewest errors in all categories, indicating a higher accuracy in predicting the tool invocation graph.
- gpt-3.5-turbo and code-llama-13b show a progressively higher number of errors. Notably, the 'Incorrect Tool Dependency' is the most common across all models, highlighting the challenge LLMs face in predicting accurate parameters for tools.

Table 5: Comparative analysis of Back-Instruct, Back-Instruct w/o edges, and Self-Instruct.

| Method | Tools | Instruction | Tool Invocation Graph |
|---|---|---|---|
| Back-Instruct | `"sampled_nodes": [{"id": "Audio-to-Image"}, {"id": "Image Colorizer"}, {"id": "Image Stitcher"}, {"id": "Image Style Transfer"}, {"id": "Image-to-Text"}], "sampled_links": [{"source": "Image Style Transfer", "target": "Image Stitcher"}, {"source": "Audio-to-Image", "target": "Image Colorizer"}, {"source": "Image Colorizer", "target": "Image Style Transfer"}]` | I am working on a project where I've created some music and I've saved that music in an 'example.wav' file. I also have an image 'example.jpg' that I want to impart some visual appeal. Could you help me create a visual representation of my audio file that merges well with my image, applies a colorful vibe to it with 'example_style.jpg' as reference, and finally, if there is any text in my final stylized image, please extract it? | `{"task_nodes": [{"task": "Audio-to-Image", "arguments": ["example.wav"]}, {"task": "Image Stitcher", "arguments": ["example.jpg", "<node-0>"]}, {"task": "Image Colorizer", "arguments": ["<node-1>", "example_style.jpg"]}, {"task": "Image-to-Text", "arguments": ["<node-2>"]}], "links": [{"source": "Audio-to-Image", "target": "Image Stitcher"}, {"source": "Image Stitcher", "target": "Image Colorizer"}, {"source": "Image Colorizer", "target": "Image-to-Text"}]}` |
| Back-Instruct w/o edges | `"sampled_nodes": [{"id": "Audio-to-Image"}, {"id": "Image Colorizer"}, {"id": "Image Stitcher"}, {"id": "Image Style Transfer"}, {"id": "Image-to-Text"}]` | I would like to create a stylized collage from 'example.jpg' and the sound of the 'example.wav' file. The visual style should be inspired by the similar image search result. The collage should be colorized, and I would like any text in the final image to be extracted. | `{"task_nodes": [{"task": "Audio-to-Image", "arguments": ["example.wav"]}, {"task": "Image Colorizer", "arguments": ["Image Style Transfer"]}, {"task": "Image Search (by Image)", "arguments": ["example.jpg"]}, {"task": "Image Stitcher", "arguments": ["Audio-to-Image", "example.jpg"]}, {"task": "Image Style Transfer", "arguments": ["Image Stitcher", "Image Search (by Image)"]}, {"task": "Image-to-Text", "arguments": ["Image Colorizer"]}], "links": [{"source": "Audio-to-Image", "target": "Image Stitcher"}, {"source": "Image Colorizer", "target": "Image-to-Text"}, {"source": "Image Search (by Image)", "target": "Image Style Transfer"}, {"source": "Image Stitcher", "target": "Image Style Transfer"}, {"source": "Image Style Transfer", "target": "Image Colorizer"}]}` |
| Self-Instruct | `"sampled_nodes": [{"id": "Audio-to-Image"}, {"id": "Image Colorizer"}, {"id": "Image Stitcher"}, {"id": "Image Style Transfer"}, {"id": "Image-to-Text"}]` | I have a black and white image of an old newspaper (example.jpg) and I want to colorize it first, then apply the style of a modern newspaper (example.png) to it, and finally extract the text from the stylized image. | `{"task_nodes": [{"task": "Image Colorizer", "arguments": ["example.jpg"]}, {"task": "Image Style Transfer", "arguments": ["Image Colorizer", "example.png"]}, {"task": "Image-to-Text", "arguments": ["Image Style Transfer"]}], "links": [{"source": "Image Colorizer", "target": "Image Style Transfer"}, {"source": "Image Style Transfer", "target": "Image-to-Text"}]}` |

Further, we present specific cases in Table 8 to elucidate the nature of prediction errors in these LLMs. Given the following example, gpt-4 correctly interpreted the task in the given example, underscoring its advanced task automation capabilities. Conversely, gpt-3.5-turbo and code-llama-13b omitted a critical tool ('Audio Downloader'), resulting in a 'Missing Required Tool' error. Additionally,

Table 6: Error Analysis on TASKBENCH.

| Error Type | Example |
|---|---|
| Incomplete instructions | I have a long text and I would like to get a summarized version of it, then generate an image that represents the main idea of the summarized text. |
| Impractical instructions | I have a text: 'This training vid is amazing! Speed it up by 1.5x please!'. Analyze the sentiment, expand it, find the video URL and adjust the video speed. |
| Mismatched parameter types | I want to find articles related to climate change and analyze their sentiment. Please translate non-English articles to English.

`{"task_nodes": [{"task": "Text Search", "arguments": ["climate change"]}, {"task": "Text Sentiment Analysis", "arguments": ["<node-0>"]}, {"task": "Text Translator", "arguments": ["<node-1>"]}], "task_links": [{"source": "Text Search", "target": "Text Sentiment Analysis"}, {"source": "Text Sentiment Analysis", "target": "Text Translator"}]}` |
| Incorrect parameter value | I have two audio files from online lectures at the following URLs: 'example1.wav' and 'example2.wav'. I want them combined into a single audio file, transcribe the speech into text, and check the text for grammatical errors.

`{"task_nodes": [{"task": "Audio Downloader", "arguments": ["example1.wav", "example2.wav"]}, {"task": "Audio Splicer", "arguments": ["<node-0>"]}, {"task": "Audio-to-Text", "arguments": ["<node-1>"]}, {"task": "Text Grammar Checker", "arguments": ["<node-2>"]}], "task_links": [{"source": "Audio Downloader", "target": "Audio Splicer"}, {"source": "Audio Splicer", "target": "Audio-to-Text"}, {"source": "Audio-to-Text", "target": "Text Grammar Checker"}]}` |
| Incorrect Tool Dependency | I want to create a more engaging version of this short text: 'Join us for a fun-filled evening!' and find some videos related to its sentiment.

`{"task_nodes": [{"task": "Article Spinner", "arguments": ["<node-2>"]}, {"task": "Text Expander", "arguments": ["Join us for a fun-filled evening!"]}, {"task": "Text Sentiment Analysis", "arguments": ["<node-1>"]}, {"task": "Video Search", "arguments": ["<node-2>"]}], "task_links": [{"source": "Text Expander", "target": "Text Sentiment Analysis"}, {"source": "Text Sentiment Analysis", "target": "Article Spinner"}, {"source": "Text Sentiment Analysis", "target": "Video Search"}]}` |

Table 7: Error Distribution in Different LLMs.

| | Required Tool Missing | Tool Dependency Error | Tool Parameter Error |
|---|---|---|---|
| gpt-4 | 0 | 2 | 3 |
| gpt-3.5-turbo | 2 | 8 | 11 |
| code-llama-13b | 4 | 9 | 13 |

code-llama-13b encountered compounded errors, including 'Tool Parameter Error' and 'Incorrect Tool Dependency'.

```
Instruction:
I need an audio file downloaded from 'https://www.example.com/
example.wav', then please reduce the background noise and apply
 a reverb effect according to my instruction 'reverb 50\%'.
Finally, combine it with the audio file 'example.wav'.
Gold tool invocation graph:
"task_nodes": [
{"task": "Audio Downloader", "arguments": ["https://www.example
.com/example.wav"]},
{"task": "Audio Noise Reduction", "arguments": ["<node-0>"]},
{"task": "Audio Effects", "arguments": ["<node-1>", "reverb
50%"]},
{"task": "Audio Splicer", "arguments": ["<node-2>", "example.
wav"]}]
```

Table 8: Case Study of Prediction Errors in LLMs.

| LLM | Predicted Tool Invocation Graph | Error Type |
|---|---|---|
| gpt-4 | `{"task_nodes": [{"task": "Audio Downloader", "arguments": ["https://www.example.com/example.wav"]}, {"task": "Audio Noise Reduction", "arguments": ["<node-0>"]}, {"task": "Audio Effects", "arguments": ["<node-1>", "reverb 50%"]}, {"task": "Audio Splicer", "arguments": ["<node-2>", "example.wav"]}]}` | None |
| gpt-3.5-turbo | `{"task_nodes": [{"task": "Audio Noise Reduction", "arguments": ["""https://www.example.com/example.wav""]}, {"task": "Audio Effects", "arguments": ["<node-0>", "reverb 50%"]}, {"task": "Audio Splicer", "arguments": ["<node-1>", "example.wav"]}]}` | Missing Required Tool |
| code-llama-13b | `{"task_nodes": [{"task": "Audio Noise Reduction", "arguments": ["example.wav"]}, {"task": "Audio Effects", "arguments": ["<node-0>", "reverb 50%"]}, {"task": "Audio Splicer", "arguments": ["<node-1>", "<node-0>"]}]}` | Missing Required Tool Tool Parameter Error Incorrect Tool Dependency |

## A.3   Metrics for Ranking Consistency

To compute the consistency of two rankings where the number of observations is n, we introduce two correlation coefficients: Spearman's $\rho$ and Kendall's $\tau$. In our work, they refer to the human and TASKBENCH rankings of large language models in terms of task automation capabilities.

**Spearman's $\rho$**   measures the strength and direction of the rank association between two variables. To calculate Spearman's $\rho$, start by assigning ranks to each observation in both sets of data. For any tied values, assign the average rank. Next, compute the difference in ranks between the two datasets for each observation, and then square these differences. The coefficient is calculated as follow:

$$\rho = 1 - \frac{6 \times \text{ sum of squared rank differences}}{n\left(n^2 - 1\right)} \tag{4}$$

**Kendall's $\tau$**   is calculated based on the consistency and inconsistency of pairs between two rankings. For both rankings, we will consider all possible pairs of items in them. For each pair of items, if the relative position is correct in both rankings, then we call this a "consistent pair". If the relative position is wrong, then we call this an "inconsistent pair".

$$\tau = \frac{(\text{number of concordant pairs }) - (\text{ number of discordant pairs })}{n(n-1)/2} \tag{5}$$

**Ranking Consistency Between TASKBENCH and Human Evaluation**   We report the above metrics of TASKBENCH to investigate the consistency with human evaluations. The results are shown in Table 9. We find that the average values for Kendall's $\tau$ and Spearman's $\rho$ are 0.89 and 0.78, respectively. This indicates a very positive correlation between human evaluation and our TASKBENCH, which further validates the effectiveness of our proposed framework for dataset construction.

Table 9: Alignment of TASKBENCH with human evaluation. Kendall's $\tau$ alculates the proportion of aligned pairs, while Spearman's $\rho$ measures the correlation between the ranks of elements.

| Correlation Metric | Hugging Face Tools | Multimedia Tools | Daily Life APIs | Average |
|---|---|---|---|---|
| Kendall's $\tau$ | 0.89 | 0.83 | 0.94 | 0.89 |
| Spearman's $\rho$ | 0.78 | 0.62 | 0.93 | 0.78 |

## A.4   Analysis

### A.4.1   Different Number of Tools

The greater the number of tool nodes the tool graphs contain, the more challenging it is for LLM to perform task automation. To make a clear understanding of the correlation between the number of nodes in the tool graph and the performance of LLMs in task automation, we conduct a detailed statistical analysis in Table 10. This analysis includes various metrics such as node-wise F1, node set accuracy, edge-wise F1, edge set accuracy, and graph accuracy, which measure the exact-match accuracy of the node set, edge set, and the entire graph, respectively.

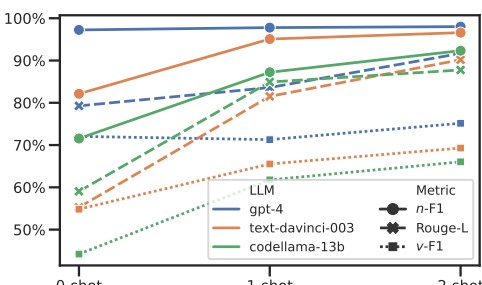

Figure 4: Performance in the few-shot setting for the Daily Life APIs domain. *R-L* for task decomposition; *n-F1* for tool selection; and *v-F1* for parameter prediction.

From Table 10, we observed that as the number of tools in the tool graphs increases, there is a clear downward trend in various performance metrics such as node set accuracy, edge set accuracy, and graph accuracy. This trend confirms that tool graphs with a higher number of tools present more complex challenges for LLMs in task automation. Specifically, the result shows a significant drop in performance metrics when moving from simpler graphs (1-2 tools) to more complex ones (3 or more tools). For instance, while single-node graphs achieve a high graph accuracy of 96.16%, this metric falls to 39.31% for graphs with 6 tools and further decreases to 25.00% for 8-node graphs.

This correlation between the number of tools and the difficulty of the test cases can be attributed to the increased complexity in understanding and processing more extensive and intricate links between tools. As the number of tools grows, LLMs must handle a larger set of possible dependencies, which

Table 10: Task automation performance with different number of tools on GPT-4

| # tool nodes | supports | node set accuracy | edge set accuracy | graph accuracy |
|---|---|---|---|---|
| 1 (single) | 2,059 | 96.16 | - | 96.16 |
| 2 | 278 | 86.33 | 84.53 | 84.53 |
| 3 | 1,313 | 67.93 | 60.70 | 60.39 |
| 4 | 1,280 | 64.29 | 75.62 | 54.37 |
| 5 | 731 | 54.03 | 70.53 | 41.58 |
| 6 | 290 | 50.34 | 39.31 | 39.31 |
| 7 | 151 | 49.66 | 36.42 | 36.42 |
| 8 | 60 | 35.00 | 25.00 | 25.00 |
| 9 | 55 | 38.18 | 21.81 | 21.81 |
| 10 | 64 | 39.06 | 31.25 | 31.25 |
| overall | 6,281 | 73.52 | 67.55 | 67.25 |

significantly challenges their predictive and analytical capabilities. The results from this analysis underline the importance of continuous advancements in LLM capabilities to keep pace with the increasing complexity of tasks in various domains.

Table 11: Performance in the few-shot setting. Rouge-L (*R-L*) reflects the performance on task decomposition; Node F1 (*n-F1*) and Edge F1 (*e-F1*) indicate the performance on tool selection; and Parameter Name F1 (*t-F1*) and Parameter Name & Value F1 (*v-F1*) indicate the performance on parameter prediction.

| Shot | LLM | $R\text{-}L\uparrow$ | $n\text{-}F1\uparrow$ | $e\text{-}F1\uparrow$ | $t\text{-}F1\uparrow$ | $v\text{-}F1\uparrow$ |
|---|---|---|---|---|---|---|
| 0-shot | gpt-4 | 79.27 | 97.23 | 34.52 | 97.11 | 72.05 |
| | gpt-3.5-turbo | 48.72 | 86.45 | 51.77 | 83.85 | 56.34 |
| | text-davinci-003 | 55.28 | 82.13 | 50.75 | 80.63 | 54.83 |
| | codellama-13b | 59.05 | 71.55 | 36.57 | 63.04 | 44.21 |
| | wizardlm-13b | 40.58 | 65.39 | 20.87 | 55.96 | 38.56 |
| | vicuna-13b-v1.5 | 48.34 | 68.32 | 26.73 | 51.47 | 35.71 |
| | nous-hermes-13b | 36.58 | 50.44 | 13.96 | 36.32 | 25.07 |
| 1-shot | gpt-4 | 83.60 | 97.78 | 50.56 | 97.82 | 71.28 |
| | gpt-3.5-turbo | 76.16 | 90.87 | 51.14 | 90.18 | 62.31 |
| | text-davinci-003 | 81.52 | 95.08 | 72.00 | 94.73 | 65.52 |
| | codellama-13b | 84.91 | 87.21 | 54.89 | 83.71 | 61.76 |
| | vicuna-13b-v1.5 | 75.52 | 73.96 | 11.17 | 62.39 | 45.81 |
| | nous-hermes-13b | 73.04 | 72.69 | 2.76 | 63.77 | 46.48 |
| | wizardlm-13b | 75.96 | 67.93 | 12.26 | 53.59 | 39.08 |
| 2-shot | gpt-4 | 91.69 | 98.02 | 52.49 | 97.94 | 75.15 |
| | gpt-3.5-turbo | 89.07 | 96.03 | 39.72 | 95.28 | 69.04 |
| | text-davinci-003 | 90.18 | 96.59 | 72.37 | 96.19 | 69.29 |
| | codellama-13b | 87.76 | 92.33 | 64.17 | 88.60 | 66.03 |
| | nous-hermes-13b | 78.88 | 82.42 | 43.55 | 74.68 | 54.39 |
| | wizardlm-13b | 80.20 | 79.25 | 33.51 | 70.69 | 52.20 |
| | vicuna-13b-v1.5 | 79.67 | 79.84 | 37.50 | 71.45 | 51.82 |

### A.4.2 Few-shot Setting

In-context learning is a crucial capability for LLMs, that can improve the performance of LLMs by providing a few examples. In TASKBENCH, we also provide a fixed number of demonstrations in the designed prompt to advance the capability of LLMs in automation. Therefore, we also investigate the effect of the number of demonstrations in our setting. The results are reported in Table 11 and Figure 4. We can find that as the number of demonstrations increases, it can receive significant improvements of LLMs at different dimensions (e.g., task decomposition, tool selection, and parameter prediction). For example, codellama-13b with a 2-shot setting can obtain 20.78% and 21.82% improvements to

the zero-shot setting in *n-F1* and *v-F1*. These results underscore the effect of the demonstrations in improving LLMs for task automation.

## A.5 Details about Back-Instruct and TaskBench

Table 15 reports the statistical information of the tool graph and our constructed TASKBENCH datasets across three domains. Notably, it is evident that the two critics we introduced play a crucial role in improving data quality. The rule-based and LLM-based critics respectively filter out an average of 15.13% and 22.73% of the samples. In addition, we invited human experts to revise and filter the data. And finally, we obtained 61.76%, 62.71%, and 60.42% of the aligned samples for the three datasets, respectively.

We utilize the following prompt template for the "Back-Instruct" Data Engine. Each sample is generated through a single prompting. We assign "instruction", "tool invocation graph", and "self-critics" to specific fields in the prompt, and then populate the relevant fields to complete the data generation in a single prompt.

```
Given a tool graph where tools serve as nodes and invoking chains
between tools act as edges, the following tools (nodes) are available
with their respective descriptions and input/output types:
{NODES}

These tools can be connected as follows, where the directed edges
represent invoking chains between tools:
{EDGES}

Based on the above tool graph, please skillfully generate the
corresponding task steps, user request, and tool invoking graph.

Requirements:
1. The generated user request should be clear, self-contained (with
user-specified text, image, video, audio, and content included within
the request) and practical (designed to help users solve a tangible
problem). The task steps must strictly adhere to the tool graph (nodes
 and edges) and be reasonable. The tool invoking graph must align with
 both the task steps and the provided tool graph.
2. The user request should be decomposable into task steps that the
tool invoking graph can solve.
3. Each task step must correspond to a tool node in both the tool
graph and the tool invoking graph. The number of task steps must equal
 the number of nodes. Each tool node can only be used once.
4. If the user request requires image/audio/video resources, use files
 named 'example.[jpg/mp4/wav/png]'.
5. The dependencies among task steps must be consistent with the edges
 of both the tool graph and the tool invoking graph.

Now, please generate your result (with random seed {seed}) in a strict
 JSON format:

{
"task_steps": [step description for one or more steps],
"user_request": "your high-quality and self-contained synthesized
request",
"invoking_graph": {
    "nodes": [
        {
            "task": "tool name",
            "arguments": [either user-specified text or resource file
            'example.[jpg/mp4/wav/png]' from the user request, or the
            name of the dependent tool whose output this node requires
            ]
        }
    ],
    "links": [{"source": "tool name i", "target": "tool name j"}]
```

```
        },
"check_by_teacher": "This field is filled by your strict and well-
trained teacher, minor mistakes are complete intolerable to him. He
evaluated whether your synthesized user request, tool invoking graph
are valid and whether they are aligned with the given tool graph (
strictly checked step by step according to the above requirements).
Some comments from him place here (start with 'Let me check your
result step by step, and evaluate the 'Executable' and 'Correct' of
the tool invoking graph (Executable means that the tool invoking graph
 executed successfully, regardless of alignment with the given tool
graph. While Correct implies that the tool invoking graph are not only
 'Executable' but also strictly consistent (with strictly same nodes
and same edges) with the given tool graph). After carefully evaluating
, found some mistakes:' and end with a conclusion: 'Conclusion:
Executable: no/yes, Correct: no/yes'.)
}:
```

## A.6   Prompt for Inference

For a fair and standardized evaluation, we provide prompt templates for inference.

```
# Tools List #
FOR tool in {tool_list}:
    {tool["id"]: {tool["description"]}}
    Parameters: {tool["parameter"]}
END FOR

# GOAL #:
Based on the above tools, I want you to generate task steps and a task
 graph (tool invocation graph, including nodes and edges) to address
the # USER REQUEST #. The format must be in strict JSON format, like:
{
    "task_steps": [step description for one or more steps],
    "task_nodes": [{
        "task": "tool name must be from # TOOL LIST #",          "
        arguments": [a concise list of arguments for the tool. This
        can be original text, a user-mentioned filename, or the tag '<
        node-j>' (starting from 0) to refer to the output of the j-th
        node.]
        }]
    "task_links": [{"source": "task name i", "target": "task name j
    "}],
}.

# REQUIREMENTS #:
1. The generated task steps and task nodes must resolve the given user
 request # USER REQUEST # perfectly. Task names must be selected from
# TASK LIST #.
2. The task steps should align strictly with the task nodes, and the
number of task steps should be the same as the task nodes.
3. The dependencies among task steps should be consistent with the
argument dependencies of the task nodes.
4. The tool arguments should align with the parameter field of # TASK
LIST #.

# EXAMPLE #:
FOR demo IN {demos}:
# USER REQUEST #:
{demo["user_request"]}
# RESULT #:
{(demo["result"])}
END FOR

# USER REQUEST #:
```

```
{{user_request}}
Now, please generate your result in strict JSON format:
# RESULT #:
```

## A.7 Detail Comparison of LLMs

We present the performance of more open-source large language models [36, 37, 38] on TASKBENCH. The performance metrics for task decomposition, tool selection, and parameter prediction are shown in Table 12, Table 13, and Table 14, respectively.

Table 12: Evaluation for task decomposition. The scores for Rouge-1 (*R1*), Rouge-2 (*R2*), and Rouge-L (*RL*) for the generated step descriptions in comparison to the ground truth steps.

| TASK DECOMPOSITION TASK - Step-by-step task decomposition | | | | | | | | |
|---|---|---|---|---|---|---|---|---|
| **LLM** | **Hugging Face Tools** | | | **Multimedia Tools** | | | **Daily Life APIs** | | |
| | *R1* ↑ | *R2* ↑ | *RL* ↑ | *R1* ↑ | *R2* ↑ | *RL* ↑ | *R1* ↑ | *R2* ↑ | *RL* ↑ |
| gpt-4 | 52.42 | 30.38 | 47.21 | 60.84 | 40.08 | 56.22 | 85.07 | 72.36 | 82.16 |
| gemini-pro | 45.96 | 24.23 | 40.06 | 53.02 | 31.51 | 46.76 | 54.36 | 27.92 | 47.28 |
| claude-2 | 44.21 | 21.12 | 37.87 | 48.85 | 23.59 | 43.23 | 82.26 | 69.88 | 78.99 |
| text-davinci-003 | 36.68 | 17.61 | 31.86 | 49.23 | 27.97 | 43.44 | 68.27 | 50.30 | 62.01 |
| gpt-3.5-turbo | 42.99 | 21.58 | 36.96 | 49.66 | 28.51 | 43.60 | 58.53 | 39.90 | 53.30 |
| mistral-7b-v0.3 | 41.04 | 19.94 | 34.20 | 50.73 | 28.97 | 43.85 | 73.03 | 57.73 | 68.98 |
| codellama-13b | 38.75 | 18.37 | 33.39 | 44.46 | 23.30 | 39.21 | 89.86 | 83.27 | 86.12 |
| wizardlm-13b | 34.47 | 15.38 | 28.76 | 35.87 | 17.55 | 30.51 | 82.02 | 72.43 | 77.57 |
| vicuna-13b-v1.5 | 37.12 | 17.03 | 30.77 | 44.75 | 23.75 | 38.92 | 81.76 | 71.76 | 76.86 |
| nous-hermes-13b | 37.36 | 16.91 | 30.50 | 35.73 | 16.11 | 29.17 | 78.49 | 68.04 | 73.42 |
| codellama-7b | 38.97 | 18.62 | 33.25 | 43.76 | 22.93 | 38.97 | 56.98 | 38.83 | 50.68 |
| baichuan-13b-chat | 19.93 | 5.97 | 17.94 | 20.41 | 3.77 | 17.50 | 49.43 | 27.25 | 42.74 |
| llama-3-8b-inst | 6.32 | 1.13 | 5.58 | 13.51 | 4.36 | 11.58 | 42.53 | 29.79 | 38.69 |
| llama-2-13b-chat | 39.37 | 18.64 | 32.60 | 26.16 | 7.88 | 22.57 | 45.39 | 22.42 | 38.29 |
| internlm-chat-7b | 20.53 | 7.16 | 17.73 | 16.64 | 3.56 | 14.65 | 42.94 | 21.02 | 36.91 |
| vicuna-7b-v1.5 | 27.17 | 10.02 | 22.69 | 39.46 | 19.83 | 32.73 | 40.26 | 21.19 | 33.13 |
| llama-2-7b-chat | 24.12 | 8.68 | 19.31 | 34.51 | 15.91 | 29.57 | 37.06 | 16.49 | 29.02 |

## A.8 Tools in the Tool Graph

We show some of the tools used in the construction of the tool graph, including the tool name, tool description and parameters of the tool. In the Daily Life APIs domain, we resorted to manual construction because of the scarcity of publicly available APIs. We crafted 40 APIs tailored to common daily life activities such as shopping, education, and travel. Our focus is solely on producing the API documentation without implementing the actual functionality. Some of the tools on the Hugging Face tools, Multimedia tools and Daily Life APIs domains are shown in Table 16, Table 17, and Table 18, respectively.

In order to visualize the complete tool graph we constructed, we take the Multimedia domain as an example to render the tool graph with resource dependencies. As shown in Figure 5 and 6, nodes in the graph denote tools, and directed edges indicate that the output type of the source tool matches the input type of the target tool.

Table 13: Evaluation for tool selection. *n-F1* and *e-F1* for node and edge prediction. *NED* measures the normalized number of operations required to correct the prediction for chain structure.

| | **LLM** | **Node** | **Chain** | | | **DAG** | | **Overall** | |
|---|---|---|---|---|---|---|---|---|---|
| | | *n-F1* ↑ | *n-F1*↑ | *e-F1* ↑ | *NED* ↓ | *n-F1*↑ | *e-F1* ↑ | *n-F1*↑ | *e-F1* ↑ |
| **Hugging Face Tools** | gpt-4 | 84.34 | 80.79 | 55.73 | 39.70 | 82.86 | 56.39 | 81.54 | 54.70 |
| | gemini-pro | 77.46 | 76.12 | 45.51 | 43.10 | 79.05 | 49.36 | 76.62 | 43.50 |
| | claude-2 | 69.83 | 80.67 | 48.11 | 40.03 | 84.52 | 53.40 | 79.00 | 43.51 |
| | gpt-3.5-turbo | 56.91 | 72.63 | 39.92 | 46.52 | 73.79 | 38.55 | 69.49 | 33.36 |
| | text-davinci-003 | 40.71 | 66.05 | 36.04 | 48.57 | 64.64 | 34.19 | 59.38 | 29.37 |
| | mistral-7b-v0.3 | 60.74 | 67.00 | 25.70 | 52.74 | 68.55 | 26.37 | 65.96 | 21.91 |
| | codellama-13b | 43.68 | 55.65 | 17.80 | 62.23 | 52.87 | 13.19 | 53.16 | 14.64 |
| | baichuan-13b-chat | 58.29 | 52.82 | 8.07 | 61.52 | 53.29 | 7.82 | 53.85 | 7.65 |
| | nous-hermes-13b | 58.66 | 52.39 | 9.01 | 62.48 | 51.99 | 6.33 | 53.62 | 8.29 |
| | llama-2-13b-chat | 43.59 | 49.87 | 8.22 | 64.99 | 49.60 | 9.11 | 48.47 | 7.30 |
| | vicuna-13b-v1.5 | 51.74 | 50.37 | 8.40 | 66.83 | 52.46 | 9.06 | 50.82 | 7.28 |
| | codellama-7b | 18.81 | 47.70 | 8.52 | 63.55 | 45.20 | 7.17 | 37.59 | 5.35 |
| | llama-3-8b-inst | 13.01 | 8.81 | 2.86 | 96.00 | 6.62 | 1.88 | 9.27 | 2.64 |
| | vicuna-7b-v1.5 | 36.20 | 44.79 | 3.24 | 69.40 | 43.94 | 2.00 | 42.87 | 2.76 |
| | wizardlm-13b | 54.69 | 54.50 | 2.22 | 60.55 | 52.93 | 0.92 | 54.40 | 2.05 |
| | llama-2-7b-chat | 14.89 | 32.61 | 0.71 | 81.01 | 31.47 | 1.38 | 27.30 | 0.74 |
| | internlm-chat-7b | 33.98 | 22.86 | 0.81 | 85.69 | 22.01 | 1.22 | 24.39 | 0.83 |
| | longchat-7b-v1.5 | 44.97 | 49.11 | 0.52 | 65.74 | 48.41 | 1.04 | 48.18 | 0.56 |
| **Multimedia Tools** | gpt-4 | 97.13 | 89.70 | 69.29 | 28.93 | 92.32 | 71.64 | 90.90 | 69.27 |
| | gemini-pro | 73.61 | 82.65 | 55.50 | 35.62 | 85.29 | 57.80 | 81.54 | 52.07 |
| | claude-2 | 66.16 | 83.95 | 59.22 | 33.41 | 82.98 | 54.28 | 80.94 | 53.01 |
| | text-davinci-003 | 59.15 | 76.87 | 50.79 | 38.54 | 79.00 | 50.69 | 73.97 | 45.81 |
| | gpt-3.5-turbo | 53.55 | 76.81 | 50.30 | 39.05 | 78.65 | 49.52 | 72.83 | 44.02 |
| | mistral-7b-v0.3 | 64.00 | 78.32 | 41.12 | 40.75 | 79.96 | 41.36 | 76.11 | 35.34 |
| | codellama-13b | 43.70 | 66.89 | 28.77 | 46.35 | 68.68 | 28.79 | 62.78 | 24.61 |
| | codellama-7b | 40.43 | 56.15 | 16.90 | 54.36 | 57.55 | 16.71 | 53.29 | 14.76 |
| | vicuna-13b-v1.5 | 66.64 | 59.18 | 16.49 | 54.17 | 61.40 | 13.95 | 60.61 | 14.78 |
| | nous-hermes-13b | 60.58 | 58.53 | 9.47 | 56.02 | 59.39 | 9.57 | 58.97 | 8.90 |
| | wizardlm-13b | 55.13 | 50.57 | 4.92 | 58.46 | 49.38 | 5.52 | 51.24 | 4.82 |
| | llama-3-8b-inst | 22.47 | 14.90 | 5.67 | 92.91 | 15.64 | 6.18 | 16.01 | 5.34 |
| | baichuan-13b-chat | 45.59 | 41.96 | 4.95 | 64.28 | 42.05 | 8.46 | 42.51 | 5.19 |
| | longchat-7b-v1.5 | 43.54 | 42.72 | 4.25 | 67.09 | 44.83 | 5.30 | 43.08 | 3.95 |
| | vicuna-7b-v1.5 | 36.22 | 48.29 | 4.79 | 63.49 | 48.26 | 4.09 | 46.06 | 4.26 |
| | llama-2-13b-chat | 38.02 | 45.14 | 1.62 | 65.29 | 45.95 | 2.11 | 43.87 | 1.63 |
| | llama-2-7b-chat | 16.49 | 30.00 | 0.94 | 76.13 | 28.81 | 1.23 | 26.47 | 0.91 |
| | internlm-chat-7b | 36.39 | 22.21 | 1.17 | 84.65 | 22.53 | 1.03 | 23.60 | 1.14 |
| **Daily Life APIs** | gpt-4 | 95.97 | 97.06 | 83.47 | 38.69 | 96.41 | 42.01 | 96.91 | 80.53 |
| | claude-2 | 79.57 | 95.36 | 80.68 | 39.93 | 93.85 | 41.04 | 93.52 | 75.31 |
| | gemini-pro | 76.15 | 92.79 | 64.58 | 41.64 | 89.68 | 28.42 | 90.75 | 59.45 |
| | gpt-3.5-turbo | 52.18 | 90.80 | 70.66 | 43.50 | 86.94 | 30.85 | 85.37 | 60.67 |
| | text-davinci-003 | 68.49 | 82.15 | 60.12 | 47.14 | 76.81 | 24.54 | 80.42 | 54.90 |
| | codellama-13b | 89.75 | 87.80 | 65.92 | 44.42 | 83.61 | 27.47 | 87.73 | 63.16 |
| | mistral-7b-v0.3 | 81.55 | 80.52 | 50.95 | 51.80 | 79.17 | 25.04 | 80.54 | 45.87 |
| | codellama-7b | 40.19 | 62.00 | 31.11 | 59.14 | 58.19 | 13.35 | 59.33 | 27.23 |
| | llama-3-8b-inst | 25.22 | 57.06 | 27.70 | 72.49 | 45.60 | 7.27 | 52.93 | 23.47 |
| | llama-2-13b-chat | 34.11 | 57.61 | 20.13 | 67.06 | 56.18 | 8.42 | 55.77 | 17.02 |
| | vicuna-7b-v1.5 | 46.51 | 54.01 | 17.43 | 65.38 | 51.68 | 10.68 | 52.73 | 14.23 |
| | longchat-7b-v1.5 | 34.20 | 49.91 | 18.17 | 69.96 | 53.53 | 11.93 | 47.26 | 14.44 |
| | wizardlm-13b | 92.27 | 65.74 | 14.51 | 55.80 | 63.80 | 9.20 | 69.34 | 14.18 |
| | vicuna-13b-v1.5 | 90.59 | 73.74 | 13.24 | 51.43 | 67.92 | 5.62 | 75.67 | 12.48 |
| | baichuan-13b-chat | 52.50 | 52.60 | 11.59 | 69.27 | 52.08 | 6.53 | 52.55 | 10.61 |
| | internlm-chat-7b | 33.08 | 29.28 | 7.06 | 86.26 | 22.22 | 3.62 | 29.14 | 6.63 |
| | llama-2-7b-chat | 20.11 | 31.68 | 5.40 | 83.87 | 30.88 | 2.80 | 30.17 | 4.27 |
| | nous-hermes-13b | 92.50 | 71.17 | 3.55 | 53.47 | 70.65 | 2.86 | 73.45 | 3.50 |

Table 14: Evaluation for tool parameter prediction. Parameter Name F1 *(t-F1)* evaluates (task, parameter name) pairs, while *v-F1* assesses (task, parameter name, parameter value) triples.

**TOOL PARAMETER PREDICTION TASK - Predicts parameters for the tool execution.**

| | LLM | Node | | Chain | | DAG | | Overall | |
|---|---|---|---|---|---|---|---|---|---|
| | | *t-F1*↑ | *v-F1*↑ | *t-F1*↑ | *v-F1*↑ | *t-F1*↑ | *v-F1*↑ | *t-F1*↑ | *v-F1*↑ |
| **Hugging Face Tools** | gpt-4 | 80.05 | 74.10 | 76.66 | 58.15 | 78.24 | 60.03 | 77.31 | 60.86 |
| | gemini-pro | 67.63 | 56.54 | 66.60 | 46.35 | 70.41 | 50.56 | 67.12 | 48.54 |
| | claude-2 | 48.07 | 32.14 | 66.35 | 45.57 | 68.59 | 48.19 | 63.00 | 43.08 |
| | text-davinci-003 | 38.51 | 27.43 | 56.90 | 38.76 | 57.03 | 38.90 | 52.53 | 36.04 |
| | gpt-3.5-turbo | 37.70 | 19.81 | 60.96 | 41.15 | 61.33 | 42.89 | 55.88 | 36.32 |
| | mistral-7b-v0.3 | 29.18 | 13.19 | 46.18 | 26.09 | 45.49 | 28.73 | 42.41 | 23.40 |
| | codellama-13b | 20.09 | 12.58 | 36.40 | 21.31 | 33.43 | 20.48 | 32.06 | 18.87 |
| | nous-hermes-13b | 46.38 | 31.06 | 35.55 | 13.81 | 33.06 | 13.69 | 37.51 | 17.66 |
| | wizardlm-13b | 43.97 | 25.90 | 37.34 | 12.48 | 38.43 | 13.79 | 38.76 | 15.35 |
| | vicuna-13b-v1.5 | 29.80 | 20.54 | 32.14 | 13.57 | 32.16 | 15.23 | 31.61 | 15.38 |
| | baichuan-13b-chat | 46.18 | 29.46 | 30.29 | 9.55 | 30.10 | 10.37 | 33.17 | 13.53 |
| | longchat-7b-v1.5 | 34.94 | 19.37 | 33.07 | 11.39 | 34.06 | 13.75 | 33.57 | 13.94 |
| | llama-2-13b-chat | 25.71 | 13.11 | 28.99 | 11.14 | 30.04 | 13.60 | 28.34 | 11.85 |
| | vicuna-7b-v1.5 | 20.82 | 12.56 | 25.85 | 10.10 | 26.09 | 10.94 | 24.65 | 10.81 |
| | codellama-7b | 13.31 | 4.48 | 27.47 | 11.97 | 24.94 | 12.36 | 22.50 | 9.20 |
| | internlm-chat-7b | 20.52 | 14.08 | 14.29 | 4.76 | 14.44 | 5.62 | 15.41 | 6.64 |
| | llama-3-8b-inst | 8.14 | 3.49 | 6.54 | 4.64 | 5.77 | 4.02 | 6.02 | 4.19 |
| | llama-2-7b-chat | 7.61 | 2.46 | 15.53 | 2.81 | 15.42 | 4.15 | 13.05 | 2.79 |
| **Multimedia Tools** | gpt-4 | 95.64 | 87.12 | 85.60 | 69.83 | 87.57 | 72.79 | 87.06 | 72.31 |
| | gemini-pro | 62.21 | 50.48 | 72.99 | 55.21 | 76.13 | 58.79 | 71.67 | 54.82 |
| | claude-2 | 53.81 | 24.02 | 75.60 | 58.12 | 72.41 | 52.43 | 71.63 | 51.58 |
| | gpt-3.5-turbo | 44.94 | 11.96 | 70.53 | 47.76 | 71.82 | 47.95 | 65.91 | 40.80 |
| | text-davinci-003 | 60.30 | 20.78 | 69.91 | 44.76 | 71.91 | 45.76 | 68.48 | 40.70 |
| | mistral-7b-v0.3 | 30.70 | 14.65 | 61.42 | 41.79 | 62.32 | 42.93 | 55.52 | 36.40 |
| | codellama-13b | 32.01 | 16.10 | 52.30 | 32.51 | 53.08 | 33.79 | 48.19 | 29.13 |
| | codellama-7b | 31.79 | 23.10 | 39.42 | 24.50 | 40.52 | 26.98 | 38.04 | 24.45 |
| | vicuna-13b-v1.5 | 52.72 | 35.55 | 39.31 | 21.00 | 40.05 | 21.40 | 41.62 | 23.62 |
| | nous-hermes-13b | 50.11 | 37.80 | 41.98 | 17.89 | 43.99 | 20.04 | 43.60 | 21.69 |
| | wizardlm-13b | 49.79 | 33.59 | 36.88 | 14.87 | 36.61 | 18.68 | 39.10 | 18.74 |
| | vicuna-7b-v1.5 | 28.79 | 17.79 | 29.73 | 12.48 | 31.38 | 14.12 | 29.72 | 13.74 |
| | longchat-7b-v1.5 | 31.06 | 21.12 | 26.97 | 11.07 | 28.43 | 14.16 | 27.89 | 13.41 |
| | baichuan-13b-chat | 40.41 | 27.87 | 25.80 | 8.50 | 25.87 | 10.13 | 28.04 | 11.77 |
| | llama-2-13b-chat | 28.49 | 17.01 | 30.26 | 9.66 | 31.00 | 11.35 | 29.99 | 11.32 |
| | llama-3-8b-inst | 16.71 | 9.18 | 21.09 | 6.88 | 25.13 | 8.84 | 21.26 | 8.68 |
| | internlm-chat-7b | 24.01 | 16.04 | 12.45 | 4.81 | 13.21 | 5.54 | 13.75 | 6.09 |
| | llama-2-7b-chat | 14.00 | 7.03 | 19.73 | 5.38 | 19.20 | 5.78 | 18.27 | 5.84 |
| **Daily Life APIs** | gpt-4 | 95.83 | 76.21 | 97.23 | 70.67 | 95.95 | 69.65 | 97.02 | 71.14 |
| | claude-2 | 78.12 | 59.43 | 94.72 | 65.30 | 91.83 | 66.39 | 92.71 | 64.72 |
| | gemini-pro | 69.88 | 45.41 | 91.66 | 57.93 | 88.50 | 53.91 | 88.95 | 56.22 |
| | gpt-3.5-turbo | 43.81 | 28.77 | 89.21 | 61.11 | 83.88 | 56.13 | 81.97 | 55.66 |
| | text-davinci-003 | 61.68 | 45.53 | 80.68 | 54.54 | 76.51 | 51.91 | 78.37 | 53.40 |
| | codellama-13b | 86.34 | 71.20 | 84.31 | 61.51 | 80.42 | 60.18 | 84.26 | 62.38 |
| | mistral-7b-v0.3 | 65.86 | 50.67 | 72.03 | 49.71 | 70.52 | 48.35 | 71.21 | 49.73 |
| | nous-hermes-13b | 79.69 | 63.29 | 62.64 | 45.32 | 63.26 | 45.74 | 64.47 | 47.22 |
| | vicuna-13b-v1.5 | 83.63 | 67.71 | 61.80 | 44.54 | 57.14 | 41.72 | 64.27 | 47.31 |
| | wizardlm-13b | 89.27 | 72.96 | 50.68 | 36.48 | 49.03 | 35.75 | 55.00 | 40.53 |
| | codellama-7b | 31.62 | 21.16 | 56.33 | 37.20 | 52.56 | 33.46 | 52.99 | 34.81 |
| | vicuna-7b-v1.5 | 27.71 | 19.81 | 38.25 | 25.82 | 37.16 | 24.65 | 36.30 | 24.67 |
| | baichuan-13b-chat | 32.47 | 21.72 | 38.31 | 24.24 | 36.84 | 21.84 | 37.48 | 23.77 |
| | llama-2-13b-chat | 10.39 | 7.32 | 38.89 | 25.37 | 36.43 | 23.40 | 35.11 | 22.94 |
| | llama-3-8b-inst | 6.40 | 5.35 | 36.62 | 25.91 | 24.69 | 17.28 | 29.16 | 20.81 |
| | longchat-7b-v1.5 | 14.99 | 12.11 | 28.37 | 19.60 | 31.25 | 22.22 | 25.73 | 18.18 |
| | internlm-chat-7b | 18.67 | 15.22 | 19.56 | 13.50 | 14.48 | 10.80 | 19.21 | 13.48 |
| | llama-2-7b-chat | 6.60 | 4.21 | 16.85 | 10.53 | 16.95 | 10.46 | 14.94 | 9.34 |

Table 15: Statistics for the TASKBENCH. We report the number of nodes and links of the tool graphs. "# Avg. Nodes" and "# Avg. Links" stands for the average number of nodes and links involved in one sample. We also report the sample number and average request length for the datasets.

| Statistic | Hugging Face Tools | Multimedia Tools | Daily Life APIs |
|---|---|---|---|
| # Nodes of Tool Graph | 23 | 40 | 40 |
| # Links of Tool Graph | 225 | 449 | 1,560 |
| # Avg. Nodes | 3.47 | 3.68 | 3.82 |
| # Avg. Links | 2.46 | 2.68 | 2.8 |
| # Samples | 12,217 | 8,904 | 7,150 |
| - Node / Chain / DAG | 3,270 / 4,302 / 4,645 | 2,117 / 3,145 / 3,642 | 1,277 / 2,716 / 3,157 |
| Avg. Request Length | 41.21 | 39.15 | 38.64 |
| - Node / Chain / DAG | 28.42 / 45.72 / 46.04 | 24.71 / 43.55 / 43.73 | 12.36 / 44.49 / 44.23 |
| self-critic  Both critics | 8,456 (69.22%) | 6,281 (70.54%) | 5,432 (75.97%) |
| self-critic  LLM-based critic | 9,042 (74.01%) | 6,959 (78.16%) | 5,694 (79.63%) |
| self-critic  Rule-based critic | 10,289 (84.22%) | 7,363 (82.69%) | 6,271 (87.70%) |
| Human Verification | 7,546 (61.76%) | 5,584 (62.71%) | 4,320 (60.42%) |

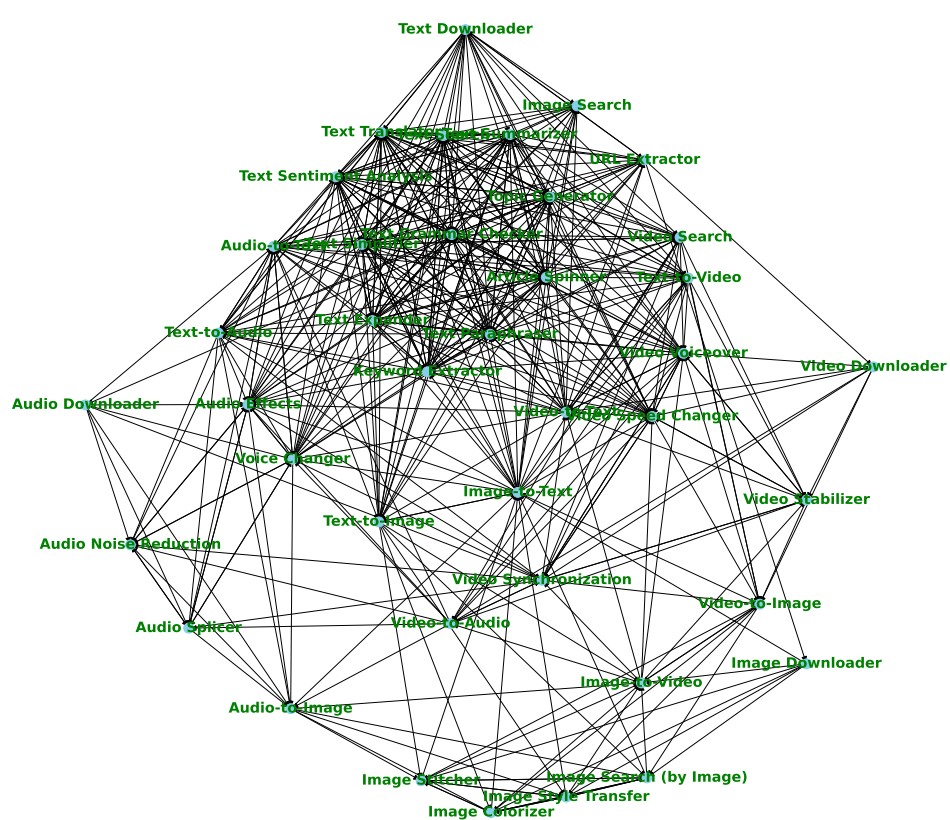

Figure 5: Constructed tool graph with resource dependencies on the Multimedia Tools domain.

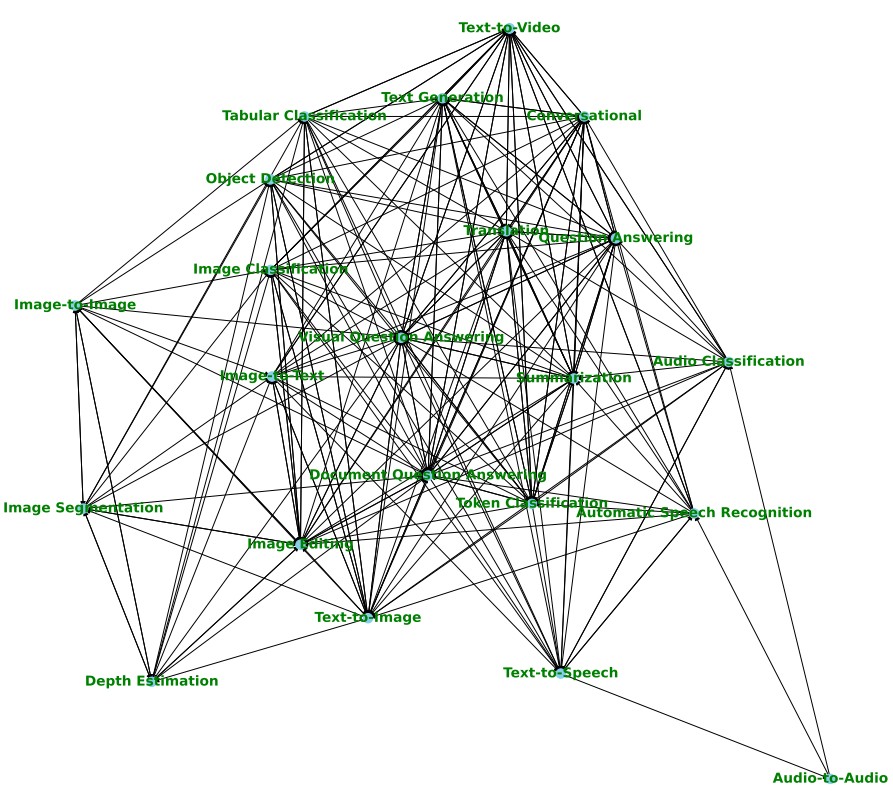

Figure 6: Constructed tool graph with resource dependencies on the Hugging Face Tools domain.

Table 16: Hugging Face tools and their descriptions

| Name | Description | Parameters |
|---|---|---|
| Translation | Translation is the task of converting text from one language to another. | ['text'] |
| Summarization | Summarization is the task of producing a shorter version of a document while preserving its important information. Some models can extract text from the original input, while other models can generate entirely new text. | ['text'] |
| Question Answering | Question Answering models can retrieve the answer to a question from a given text, which is useful for searching for an answer in a document. | ['text', 'text'] |
| Text Generation | Generating text is the task of producing new text. These models can, for example, fill in incomplete text or paraphrase. | ['text'] |
| Object Detection | Object Detection models allow users to identify objects of certain defined classes. Object detection models receive an image as input and output the images with bounding boxes and labels on detected objects. | ['image'] |
| Image Classification | Image classification is the task of assigning a label or class to an entire image. Images are expected to have only one class for each image. Image classification models take an image as input and return a prediction about which class the image belongs to. | ['image'] |
| Image-to-Image | Image-to-image is the task of transforming a source image to match the characteristics of a target image or a target image domain. Any image manipulation and enhancement is possible with image to image models. | ['image'] |
| Image-to-Text | Image to text models output a text from a given image. Image captioning or optical character recognition can be considered as the most common applications of image to text. | ['image'] |
| Text-to-Image | Generates images from input text. These models can be used to generate images based on text prompts. | ['text'] |
| Text-to-Video | Generates videos from input text. These models can be used to generate videos based on text prompts. | ['text'] |
| Visual Question Answering | Visual Question Answering is the task of answering questions based on an image. | ['image', 'text'] |
| Image Segmentation | Image Segmentation divides an image into segments where each pixel in the image is mapped to an object. This task has multiple variants such as instance segmentation, panoptic segmentation and semantic segmentation. | ['image'] |
| Depth Estimation | Depth estimation is the task of predicting depth of the objects present in an image. | ['image'] |
| Text-to-Speech | Text-to-Speech (TTS) is the task of generating natural sounding speech given text input. TTS models can be extended to have a single model that generates speech for multiple speakers and multiple languages. | ['text'] |
| Automatic Speech Recognition | Automatic Speech Recognition (ASR), also known as Speech to Text (STT), is the task of transcribing a given audio to text. It has many applications, such as voice user interfaces. | ['audio'] |
| Audio-to-Audio | Audio-to-Audio is a family of tasks in which the input is an audio and the output is one or multiple generated audios. Some example tasks are speech enhancement and source separation. | ['audio'] |
| Audio Classification | Audio classification is the task of assigning a label or class to a given audio. It can be used for recognizing which command a user is giving or the emotion of a statement, as well as identifying a speaker. | ['audio'] |
| Image Editing | Image editing is the task of modifying an image to match a given text description. It can be used to modify the attributes of an image, such as the color of an object or the background. | ['text', 'image'] |

Table 17: Multimedia tools and their descriptions

| Name | Description | Parameters |
|---|---|---|
| Image Downloader | Downloads an image from a given URL. | ['url'] |
| Video Downloader | Downloads a video from a given URL. | ['url'] |
| Audio Downloader | Downloads an audio file from a given URL. | ['url'] |
| Text Downloader | Downloads the text content from a given URL. | ['url'] |
| Text Search | Searches for a specific text or keyword on the internet. | ['text'] |
| Image Search | Searches for images on the internet based on a given query. | ['text'] |
| URL Extractor | Extracts URL from text | ['text'] |
| Video Search | Searches for videos on the internet based on a given query. | ['text'] |
| Text-to-Video | Generates a video based on a given text description. | ['text'] |
| Text-to-Audio | Generates an audio file based on a given text description. | ['text'] |
| Image-to-Text | Extracts text from an input image using Optical Character Recognition (OCR). | ['image'] |
| Audio-to-Text | Transcribes speech from an audio file into text. | ['audio'] |
| Video-to-Text | Transcribes speech from a video file into text. | ['video'] |
| Audio Noise Reduction | Reduces background noise or unwanted sounds from a given audio file. | ['audio'] |
| Audio Effects | Applies various audio effects to a given audio file according to human instruction, such as reverb, chorus, or equalization. | ['audio', 'text'] |
| Audio Splicer | Combines two audio files into a single output file. | ['audio', 'audio'] |
| Voice Changer | Modifies the characteristics of a recorded voice according to human instruction, such as tone, pitch, or gender. | ['audio', 'text'] |
| Text Summarizer | Summarizes a given text into a shorter version while retaining the main points. | ['text'] |
| Text Translator | Translates a given text from one language to english. | ['text'] |
| Text Sentiment Analysis | Analyzes the sentiment of a given text, identifying if it is positive, negative, or neutral. | ['text'] |
| Text Grammar Checker | Checks a given text for grammatical errors and suggests corrections. | ['text'] |
| Text Simplifier | Rewrites a given text in a simpler and more understandable manner. | ['text'] |
| Keyword Extractor | Extracts the most important keywords and phrases from a given text. | ['text'] |
| Text Paraphraser | Rewrites a given text using different words while maintaining its original meaning. | ['text'] |
| Topic Generator | Generates a list of relevant topics or ideas based on a given input. | ['text'] |
| Audio-to-Image | Generates an image that visually represents a given audio, such as a waveform or spectrogram. | ['audio'] |
| Video-to-Audio | Extracts the audio track from a given video file. | ['video'] |
| Video-to-Image | Extracts a still image from a given video. | ['video'] |
| Image Stitcher | Stitches together two input images to create a panorama or collage. | ['image', 'image'] |
| Image Colorizer | Adds color to a black and white input image using deep learning techniques. | ['image'] |
| Video Stabilizer | Stabilizes a shaky input video to produce a smoother output video. | ['video'] |
| Video Speed Changer | Adjusts the playback speed of a given video according to human instruction, either speeding it up or slowing it down. | ['video', 'text'] |
| Video Synchronization | Synchronizes the timing of an existing voiceover or audio file with the visuals of a given video. | ['video', 'audio'] |

Table 18: Daily Life APIs and their descriptions

| API Name | API Description | Parameter Names |
|---|---|---|
| get_news_for_topic | Get the news for a specific topic | ['topic'] |
| stock_operation | Do a specific operation on a specific stock | ['stock', 'operation'] |
| book_flight | Book a flight for a specific date, from a specific location to a specific destination | ['date', 'from', 'to'] |
| book_hotel | Book a specific hotel for a specific date | ['date', 'name'] |
| book_car | Book a car for a specific date, in a specific location | ['date', 'location'] |
| online_shopping | Buy a product from a specific website | ['website', 'product'] |
| send_email | Send an email to a specific email address | ['email_address', 'content'] |
| send_sms | Send an sms to a specific phone number | ['phone_number', 'content'] |
| share_by_social_network | Share a specific content by a specific social network | ['content', 'social_network'] |
| book_restaurant | Book a specific restaurant for a specific date | ['date', 'name'] |
| search_by_engine | Search a specific query by a specific search engine | ['query', 'engine'] |
| apply_for_job | Apply for a specific job | ['job'] |
| see_doctor_online | See a specific doctor for a specific disease | ['disease', 'doctor'] |
| consult_lawyer_online | Consult a specific lawyer for a specific legal issue | ['issue', 'lawyer'] |
| enroll_in_course | Enroll in a specific course at a specific university | ['course', 'university'] |
| buy_insurance | Buy a specific insurance from a specific insurance company | ['insurance', 'company'] |
| online_banking | Do a specific banking operation online at a specific bank | ['instruction', 'bank'] |
| daily_bill_payment | Pay a specific bill | ['bill'] |
| sell_item_online | Sell a specific item at a specific online store | ['item', 'store'] |
| do_tax_return | Do the tax return for a specific year | ['year'] |
| apply_for_passport | Apply for a passport | ['country'] |
| pay_for_credit_card | Pay for a specific credit card | ['credit_card'] |
| auto_housework_by_robot | Let a robot do a housework by following a specific instruction | ['instruction'] |
| auto_driving_to_destination | Let a car drive to a specific destination | ['destination'] |
| deliver_package | Deliver a specific package to a specific destination | ['package', 'destination'] |
| order_food_delivery | Order a specific food to be delivered to a specific location at a specific platform | ['food', 'location', 'platform'] |
| order_taxi | Order a taxi to a specific location at a specific platform | ['location', 'platform'] |
| play_music_by_title | Play a specific music by a specific title | ['title'] |

