# OpenReview forum: "TaskBench: Benchmarking Large Language Models for Task Automation"
_NeurIPS.cc/2024/Datasets_and_Benchmarks_Track — NeurIPS 2024 Track Datasets and Benchmarks Poster_

### Official Review · Reviewer_BPJD · 2024-07-21
**Good contribution but concerns with regards to quality and utility**

**Rating:** 6
**Confidence:** 4

**Review:**

- Dataset Quality: Because the dataset is completely synthetically generated, I am skeptical of its general quality, especially from the naturalness and alignment to humans side. For example. just because two tools have compatible input-output types is not sufficient to generate a natural request. Here are a couple of examples from the daily life APIs data:

> "I need to see Dr. David for my flu, then apply for a medical transcriptionist job at Company ABC"

and

> "I need to deliver a package (example.jpg) to Dr. Smith, consult with him about a rash, book a table at Mediterranean Palace for December 25th..."

These don't seem natural to me and they are not some cherry-picked examples.

- Another issue is that it seems the "task_steps" are basically just the instruction repeated. Take the following example from the huggingface data

> user_request "I've got an image named 'example.jpg', with which I need help. Could you classify the main object in this image, generate a descriptive text about it, identify specific entities within that text, and finally answer the question: 'What is the most prominent color of the object in the image?'"

> "task_steps": ["1. Classify the image to determine the main object in it.", "2. Generate a descriptive text about the classified object.", "3. Apply token classification to identify specific entities within the generated text.", "4. Use Visual Question Answering to answer a question about the image based on the identified entities."]

What would be more realistic is the user providing a complex request and the LLM figuring out how to divide it into steps. However, what I see here is that the user request is already divided into clear steps. User requests in real life are messy and not so organized and ordered. This is in my opinion again an artifact of having an LLM generate the data. The request seems very robotic, and doesn't seem aligned with real-world instructions given by humans.

- The tasks should be more challenging. The dataset in its current form is too simple and I suspect a simple fine-tuning of small LLMs on the data can come close to saturating the benchmark. Most tools have simply only one or two input parameters.

- In the introduction, this work is motivated by comparing it to generic datasets that are not suitable for evaluating LLMs on task automation. However, there is no mention of all API datasets and tool-augmented LLM datasets (function calling datasets) until the Appendix. This is slightly misleading and there should be a mention of these other datasets in the introduction as well as it is a central point to distinguish this work from existing work.

**Strengths:**

- The Tool Graph is a good idea to represent dependencies between different tools.
- A wide selection of LLMs are used in evaluation, both closed-sources LLMs and open-sourced LLMs.
- Human evaluations are conducted to verify and rate the quality of the generated data.

**Additional Feedback:**

What insights can we gain from the evaluation results? How can this dataset improve the development of models in the field of task automation?

**Clarity:**

The paper is easy to follow but some ambiguous language should be revised. For example, in the introduction, the meaning of the following phrase is not clear "Consequently, many researchers attempt to delve deeper into LLM". Similar sentences in the paper should be revised.

**Correctness:**

Overall the methodology used for creating the dataset and evaluating the models seems to be correct although it is completely dependent on LLM for data generation.

**Documentation:**

The paper provides sufficient details on data collection and annotation, and the dataset and associated code is provided in supplementary material.

**Ethics:**

The paper does not raise any immediate ethical concerns.

**Limitations:**

From Tables 15, 16 and 17, it seems that most tools have only one or two parameters. Compared to other works where tools can take many input parameters, this seems like a limitation of the dataset. Also, the task here is to simply predict the APIs but in real-world task automation, the LLM needs to execute APIs then observe the output and based on it proceed to next steps., i.e. there is a "dynamic" nature to it.

**Opportunities For Improvement:**

- More human involvement is needed in creating the dataset aside from just verification and doing comparisons.
- Dataset quality should be improved from both naturalness and utility.

**Relation To Prior Work:**

Relations to prior work is discussed in Appendix A

**Summary And Contributions:**

The paper introduces a benchmark named TaskBench aimed at evaluating LLMs in the domain of task automation. The authors divide task automation intro three subtasks, which involve task decomposition, tool selection, and parameter prediction. Evaluations are done on a wide range of LLMs on each subtask.

---

> ### Author Rebuttal · Authors · 2024-08-18
>
> > Dataset Quality: Because the dataset is completely synthetically generated, I am skeptical of its general quality, especially from the naturalness and alignment to humans side.
>
> We appreciate the reviewer's detailed feedback on the naturalness and alignment aspects of the synthetically generated dataset. Below, we outline the steps we have taken and our ongoing efforts to enhance the dataset's quality:
>
> - **Challenges of Synthetic Data Generation:** We acknowledge that synthetic data generation can sometimes produce instructions that, while logically correct, may not align with natural human language or typical real-world requests. The examples provided by the reviewer highlight instances where the generated instructions, although technically feasible, may not reflect how a human would naturally frame such requests. This is a known challenge in balancing the logical structure provided by tool compatibility with the need for naturalness in language.
> - **Tool Compatibility vs. Naturalness**: The examples mentioned illustrate the challenge of ensuring that the logical connections between tools (based on compatible input-output types) also translate into naturally coherent and contextually appropriate instructions. We agree that just because tools have compatible input-output types does not mean that combining them will always result in a natural-sounding request. Our approach, while focused on maintaining logical consistency, sometimes results in less-than-ideal naturalness, especially in cases involving diverse or unrelated tasks.
> - **Current Efforts to Improve Data Quality:** To address the issue of naturalness and improve the data quality, we have provided these strategies:
>    - **Structured Sampling**: Unlike previous approaches such as ToolQA, ToolAlpaca, and ToolBench, our methodology incorporates a Tool Graph that inherently **preserves the dependencies and relationships between different tasks and tools**. The sampling process is not random but follows a structured methodology based on a carefully constructed Tool Graph. This graph is designed to **reflect real-world dependencies and relationships between tools,** ensuring that even when sampling sub-graphs with up to 10 nodes, the resulting instructions maintain logical coherence and relevance to practical applications.
>    - **Back-Instruct**: Our dataset is generated using the Back-Instruct method, which leverages these sampled sub-graphs to produce user instructions. This method **guides the generation process by ensuring that the instructions are logically consistent with the sampled sub-graph**, effectively aligning the task structure with the tools involved. This approach helps maintain the naturalness and complexity of the generated instructions.
>    - **Self-Critic Mechanisms:** We have implemented **a dual-layer self-critic system, combining LLM-based and rule-based filters.** The combined self-critic mechanism ensures that only high-quality, coherent instructions are retained.
>    - **Human Verification**: We implemented **a rigorous human verification process to further ensure the quality and coherence of the dataset** (Please see Appendix A.7). Human experts reviewed the generated instructions across different task complexities and confirmed their logical consistency and alignment with the intended tasks. This verification process covered instructions generated from sub-graphs of varying sizes, ensuring that the instructions are meaningful and practical. The human verification process involved 12 experts who reviewed about 2000 samples, as detailed in our response regarding the verification process. This comprehensive review provides additional assurance that the dataset, including the more complex examples, meets high standards of quality.
>
> We acknowledge the reviewer's concerns and are actively working to address them. We are committed to further enhancing the dataset by integrating more advanced techniques to improve the naturalness of generated instructions. This includes refining our data generation pipeline and increasing the role of human verification in the iterative refinement process.

---

> > ### Author Rebuttal · Authors · 2024-08-18
> >
> > > Another issue is that it seems the "task_steps" are basically just the instruction repeated. What would be more realistic is the user providing a complex request and the LLM figuring out how to divide it into steps. However, what I see here is that the user request is already divided into clear steps. User requests in real life are messy and not so organized and ordered. This is in my opinion again an artifact of having an LLM generate the data. The request seems very robotic, and doesn't seem aligned with real-world instructions given by humans.
> >
> > We appreciate the reviewer's detailed feedback on the realism of the "task_steps" and user requests within the dataset. We understand the importance of ensuring that the dataset reflects how real-world users would interact with an AI system, and we would like to address the concerns raised:
> >
> > - **Clarification of Task Steps**:
> >    - The "task_steps" in our dataset are designed to serve as a clear, step-by-step breakdown of the actions required to fulfill the user instruction. These steps are indeed closely aligned with the user request, as they are intended to mirror the logical decomposition of the task. The purpose of this design is to ensure that the instructions are explicitly aligned with the intended outcomes, providing a better explanation for LLM to follow.
> >    - We understand the concern that user requests in real life are often messy, unstructured, and not as neatly ordered as those presented in our dataset. The examples provided by the reviewer highlight a key challenge in synthetic data generation: achieving a balance between logical coherence and the natural messiness of real-world human requests.
> > - **Balancing Structure and Realism**:
> >    - **Human Expert Involvement**: We are refining our data generation process by involving human experts to revise and modify the generated instructions. These experts will intentionally make the instructions more implicit and less straightforward, mimicking the natural way humans often give incomplete or ambiguous instructions. This revision process is designed to ensure that the dataset better reflects the indirect and nuanced nature of real-world user requests.
> >    - **Broader Range of Request Formats**: In addition to making instructions more implicit, we will introduce a variety of request formats, including more conversational and less structured prompts. This will ensure that the dataset reflects a wide spectrum of real-world communication styles, challenging LLMs to handle diverse and unpredictable language inputs effectively.
> >
> > While the current design of the "task_steps" and user requests emphasizes clarity and logical decomposition, we recognize the need for greater realism and variability. We are committed to enhancing the dataset to better reflect the complexities and nuances of real-world user interactions, ensuring that it provides a more accurate benchmark for evaluating LLMs in task automation.

---

> > > ### Author Rebuttal · Authors · 2024-08-18
> > >
> > > > The tasks should be more challenging. The dataset in its current form is too simple and I suspect a simple fine-tuning of small LLMs on the data can come close to saturating the benchmark.Most tools have simply only one or two input parameters.
> > >
> > > We appreciate the reviewer's feedback on the need for more challenging tasks in the dataset. Below, we address the concerns and outline our approach to ensuring that the benchmark remains challenging for evaluating LLM capabilities:
> > >
> > > - **Current Task Complexity**:
> > >    - **Broad Range of Tasks**: Compared with previous datasets (e.g., ToolBench, ToolQA), TaskBench is intentionally designed to encompass a wide spectrum of task complexities, from simpler tasks to more complex tasks with multiple steps and dependencies. We also require model to evaluate multiple metrics including task decomposition, tool selection and parameter predictions. This diversity is essential to provide a comprehensive evaluation of LLMs across different levels of difficulty.
> > >    - **Complexity and Model Performance**: As evidenced in Table 3, even advanced models like GPT-4 exhibit varying levels of performance across different task structures, particularly struggling with Directed Acyclic Graphs (DAGs) where task dependencies are more intricate. For instance, GPT-4’s t-F1 score drops from 80.05 in Node tasks to 60.03 in DAG tasks, indicating that the complexity of the task structures in TaskBench does indeed challenge even the most capable models currently available.
> > > - **Comparison with Existing Benchmarks**:
> > >    - **Berkeley Function-Calling Leaderboard**: While this leaderboard effectively tests models on fundamental API interactions, it focuses on simpler tasks involving basic function calls, multiple, and parallel function executions. Advanced models like GPT-4 and Claude-3 tend to saturate this leaderboard quickly, particularly in less complex task categories like "Simple Function" and "Multiple Functions," leading to high scores that make it challenging to differentiate between top-performing models.
> > >    - **TaskBench's Unique Challenges**: In contrast, TaskBench introduces tasks involving Directed Acyclic Graphs (DAGs) that require models to navigate complex workflows with multiple dependencies and sequential/parallel steps. These tasks go beyond basic function calling and test a model’s understanding of task hierarchies and dependencies. The inherent complexity of DAG tasks results in significantly lower t-F1 and v-F1 scores across models, which prevents early saturation and ensures that TaskBench remains a relevant and challenging benchmark for evaluating cutting-edge LLM advancements.
> > > - **Enhancing Task Complexity**:
> > >    - **Increased Parameter Diversity**: To address the issue of tools having only one or two input parameters, we plan to expand the dataset to include tasks that involve tools with a wider range of parameters, including optional and conditional parameters. This will require LLMs to not only identify the correct tools but also to configure them appropriately based on the task requirements.
> > >    - **Contextual and Conditional Tasks**: We will introduce tasks that require understanding of context or conditional logic, where the LLM needs to make decisions based on specific conditions or the results of previous steps. This will add a layer of complexity, as the model will need to dynamically adjust its approach based on the evolving state of the task.
> > >
> > > We are continuously working to enhance TaskBench by introducing more complex and varied tasks, ensuring that the benchmark evolves alongside advancements in LLM capabilities. This ongoing effort will prevent saturation and maintain TaskBench as a cutting-edge tool for evaluating the full spectrum of LLM abilities.
> > >
> > > > In the introduction, this work is motivated by comparing it to generic datasets that are not suitable for evaluating LLMs on task automation. However, there is no mention of all API datasets and tool-augmented LLM datasets (function calling datasets) until the Appendix. This is slightly misleading and there should be a mention of these other datasets in the introduction as well as it is a central point to distinguish this work from existing work.
> > >
> > > We appreciate the reviewer's feedback on the importance of clearly distinguishing our work from existing API and tool-augmented LLM datasets (such as function calling datasets) within the introduction of the paper. Our method are different from other works from these perspectives:
> > >
> > > - **Tool Graph and Back-Instruct Strategy**: Unlike previous datasets such as ToolBench, ToolQA, ToolAlpaca, and Gorilla, which also rely on synthetic instructions, TaskBench utilizes a Tool Graph and Back-Instruct strategy to ensure that generated instructions are aligned with targeted task sequences. This approach allows TaskBench to more accurately simulate task automation scenarios that involve complex dependencies, tool chaining, and nuanced parameter settings across diverse applications.
> > > - **Broader Scope of Task Automation**: While many existing datasets focus on specific aspects of tool usage or API interactions, TaskBench covers a broader spectrum of task automation, incorporating complex tool graphs and task dependencies across diverse domains. This makes TaskBench a more comprehensive benchmark for evaluating the capabilities of LLMs in automating real-world tasks.
> > >
> > > We appreciate the reviewer's insight. We will move the discussion of related datasets from the Appendix to the main body of the paper, specifically integrating this comparison into the introduction. These revisions will strengthen the clarity and impact of our paper, making it clear how TaskBench is both motivated by and distinct from existing datasets.

---

> > > > ### Author Rebuttal · Authors · 2024-08-18
> > > >
> > > > > More human involvement is needed in creating the dataset aside from just verification and doing comparisons.
> > > >
> > > > Thanks for your suggestions. We agree that human expertise is crucial in ensuring the quality and realism of the dataset. We would like to clarify and expand upon the role of human involvement in our current process.
> > > >
> > > > As described in the manuscript (Line 638 in Appendix A.7), **human experts are already deeply involved in the verification process**. This process goes beyond superficial checks; **it involves detailed assessments of the naturalness, complexity, and alignment of the generated instructions and tool invocation graphs.** We enlisted 12 experts who reviewed about 2000 samples, including those generated from sub-graphs with varying complexities. This verification ensures that the instructions are logically consistent, meaningful, and practical, providing an additional layer of quality control. We will provide more details about this part in the future.
> > > >
> > > > **We believe that a balanced approach, combining automated processes with strategic human involvement, is key to creating a high-quality dataset.** Automation allows for the efficient generation of large datasets, while human insight ensures that the data aligns with real-world expectations and complexities. We are committed to continuously improving our dataset, and we appreciate the reviewer's feedback as it helps guide our efforts.
> > > >
> > > > > Dataset quality should be improved from both naturalness and utility.
> > > >
> > > > We appreciate the reviewer's feedback highlighting the importance of enhancing both the naturalness and utility of our dataset. Below, we outline the specific measures we have taken and continue to implement to address these concerns:
> > > >
> > > > - **Structured Sampling**: The sampling process is not random but follows a structured methodology based on a carefully constructed Tool Graph. This graph is designed to reflect real-world dependencies and relationships between tools, ensuring that even when sampling sub-graphs with up to 10 nodes, the resulting instructions maintain logical coherence and relevance to practical applications.
> > > > - **Back-Instruct**: Our dataset is generated using the Back-Instruct method, which leverages these sampled sub-graphs to produce user instructions. This method guides the generation process by ensuring that the instructions are logically consistent with the sampled sub-graph, effectively aligning the task structure with the tools involved. This approach helps maintain the naturalness and complexity of the generated instructions.
> > > > - **Enhancing Naturalness**: To further improve naturalness, we have implemented a robust human verification process. This process involves domain experts who review and refine the synthesized instructions to ensure they align with real-world language use and expectations. By incorporating human insights, we can better capture the nuances of natural language, resulting in instructions that are more intuitive and reflective of typical human interactions.
> > > > - **Improving Utility**: To enhance utility, we are continuously refining the tool graphs and their associated dependencies to better mirror common use cases and workflows encountered in practical settings. This involves a continuous process of reviewing and adjusting the dataset to ensure that the tasks generated are not only logically sound but also practically useful for real-world applications. By focusing on real-world utility, we ensure that the dataset can effectively support a wide range of practical applications in task automation.
> > > >
> > > > > From Tables 15, 16 and 17, it seems that most tools have only one or two parameters. Compared to other works where tools can take many input parameters, this seems like a limitation of the dataset.
> > > >
> > > > We appreciate the reviewer's observation regarding the number of parameters associated with tools in our dataset. Generally, the core idea of our method is to use Tool Graph to maintain the relationship between different tasks and use back-instruct strategy to generate instructions. Therefore, it can easily to extend to tools with arbitrary parameters by replacing these tools in the tool graph. In this version, **we choose some tools that can reflect common real-world scenarios where tasks are typically automated with a few key inputs**. In many practical applications, especially in domains like daily life APIs and multimedia tools, the most commonly used tools tend to have one or two primary parameters that significantly influence their operation. Our dataset reflects this reality by prioritizing these essential parameters. Besides, our paper is not just focusing on the parameters of tools, but also need to consider multiple aspects of task automation, including task planning, tool invocation, and parameter prediction. **Even when working with tools that have fewer parameters, models must accurately predict task dependencies and manage tool chains.** The performance of advanced models like GPT-4, which achieves accuracy rates of 71.14%, 60.86%, and 72.31% in the Daily Life APIs, Hugging Face, and Multimedia domains respectively (as shown in Table 3 of the submission), **illustrates the significant challenge our dataset poses, despite the seemingly simpler tool configurations.** In the future, we will also explore to use more challenging tools that involve more parameters and produce more natural instructions. Our Back-Instruct method will automatically generate corresponding data based on the signatures and dependencies of newly integrated tools, thereby increasing the overall complexity and further challenging LLMs.

---

> > > > > ### Author Rebuttal · Authors · 2024-08-18
> > > > >
> > > > > > Also, the task here is to simply predict the APIs but in real-world task automation, the LLM needs to execute APIs then observe the output and based on it proceed to next steps., i.e. there is a "dynamic" nature to it.
> > > > >
> > > > > We appreciate the reviewer's insightful comment on the dynamic nature of real-world task automation, where an LLM must execute APIs, observe outputs, and make decisions for subsequent steps based on those outcomes. We fully acknowledge the importance of this dynamic aspect and offer the following points in response:
> > > > >
> > > > > - **Current Focus of TaskBench**: The primary focus of the current version of TaskBench is to evaluate the LLM's capability in predicting the correct APIs, along with their parameters and the necessary sequence of actions based on given instructions. This static evaluation is a crucial first step in understanding the model's ability to comprehend and decompose tasks.
> > > > > - **Challenges in Dynamic Evaluation**: We agree that in a real-world setting, task automation requires not only predicting the correct sequence of API calls but also executing these calls, observing the outcomes, and making dynamic decisions based on those outcomes. However, evaluating this dynamic process poses significant challenges in a benchmarking context, particularly in terms of standardization and reproducibility.
> > > > >
> > > > > Recognizing the importance of dynamic task automation, we plan to extend TaskBench in future iterations to include more interactive and dynamic evaluation components. These will involve:
> > > > >
> > > > > - **Simulated Execution Environments**: We aim to develop simulated environments where the LLM can not only predict but also execute APIs and receive simulated outputs. This will allow us to evaluate how the LLM adapts and modifies its actions based on the results of previous steps.
> > > > > - **Dynamic Decision-Making Tasks**: Future benchmarks will incorporate scenarios where the LLM must make decisions based on real-time feedback, closely mimicking the dynamic nature of real-world task automation.
> > > > >
> > > > > While the current benchmark focuses on static evaluation, it lays a critical foundation for understanding how well an LLM can interpret instructions and plan tasks, which are essential components of dynamic task automation. We are committed to iteratively improving TaskBench, incorporating more dynamic elements as the benchmark evolves. We view this as a necessary step toward creating a comprehensive evaluation framework that mirrors the complexities of real-world task automation.
> > > > >
> > > > > > the meaning of the following phrase is not clear "Consequently, many researchers attempt to delve deeper into LLM".
> > > > >
> > > > > Sorry for the confusion. The phrase was intended to convey that researchers are increasingly focusing on exploring and enhancing the capabilities of LLMs to improve their effectiveness in task automation. This exploration includes developing new techniques, improving the underlying models, and finding better ways to apply LLMs to autonomous task automation.
> > > > >
> > > > > To make the meaning clearer, we propose revising the sentence as follows:
> > > > >
> > > > > _"Consequently, many researchers are focusing on exploring and advancing the capabilities of LLMs to enable more intelligent and effective task automation."_

---

> > > > > > ### Author Rebuttal · Authors · 2024-08-18
> > > > > >
> > > > > > > What insights can we gain from the evaluation results? How can this dataset improve the development of models in the field of task automation?
> > > > > >
> > > > > > We appreciate the reviewer's request for a more detailed exploration of the insights derived from our evaluation results and the broader impact of our dataset on advancing task automation. Below, we provide a comprehensive discussion of these aspects, highlighting the significance of our findings and how TaskBench contributes to the development of task automation:
> > > > > >
> > > > > > **Insights from Evaluation Results**:
> > > > > >
> > > > > > - **Performance Across Complexity Levels**: The evaluation results reveal how different models handle varying levels of task complexity. For instance, models that excelled in tasks with lower node counts but struggled as the complexity increased provide insight into their current limitations in generalizing across more complex task structures. This highlights the need for models to develop stronger reasoning and planning capabilities, especially as task complexity grows.
> > > > > > - **Strengths and Weaknesses in Tool Selection and Parameter Prediction**: By breaking down the evaluation into task decomposition, tool selection, and parameter prediction, we can pinpoint specific areas where models perform well and where they falter. For example, a model might accurately select tools but struggle with predicting the correct parameters, indicating a gap in its understanding of the nuances in tool usage. This insight is critical for guiding further model improvements.
> > > > > > - **Fundamental Capabilities on Task Automation**:
> > > > > >    - **Reasoning**: One of the key insights from our evaluation is the importance of reasoning capabilities in task automation. The success of LLMs in this domain is closely tied to their ability to solve complex problems and reason effectively. For example, the GPT series demonstrated superior reasoning skills in tasks such as mathematical problem-solving and coding, which are indicative of its robust capabilities in task planning and tool usage. This suggests that models with strong reasoning abilities are better equipped to handle the multifaceted demands of task automation.
> > > > > >    - **Instruction Following**: Another crucial insight is the role of instruction following in enhancing model performance. Models like Vicuna-13b and WizardLLM-13b, which are fine-tuned specifically for following instructions, tend to outperform others such as Llama-2-13b. Notably, WizardLLM-13b shows a marked improvement over Vicuna-13b, underscoring the impact of sophisticated instruction fine-tuning on overall task automation performance.
> > > > > > - **Factors Contributing to Task Automation Performance**:
> > > > > >    - **Code Pre-training**: Our analysis highlights that models with extensive code pre-training, such as Code-Llama, exhibit superior performance in task automation. Specifically, our data shows an average improvement of 4.45% in tool prediction and 12.76% in parameter prediction across various domains. This improvement underscores the importance of structured text (like code) in enhancing the model's ability to connect different stages of task automation, such as decomposing tasks, selecting appropriate tools, and predicting parameters.
> > > > > >    - **Alignment Techniques**: Models that incorporate human alignment techniques, such as the GPT series with Reinforcement Learning from Human Feedback (RLHF), demonstrate enhanced capabilities in task automation compared to open-source counterparts. The RLHF approach helps models generalize reasoning skills more effectively and reduces the risk of instruction-specific overfitting, thereby improving their ability to perform diverse and complex tasks.
> > > > > >
> > > > > > **How the Dataset Improves Model Development in Task Automation**:
> > > > > >
> > > > > > - **Benchmarking and Baseline Establishment**: TaskBench provides a comprehensive benchmark that allows researchers to compare their models against established baselines across a variety of task automation scenarios. This facilitates a clearer understanding of where their models stand in terms of task decomposition, tool selection, and parameter prediction, and guides targeted improvements.
> > > > > > - **Promoting Generalization Across Domains**: The diversity of the domains included in TaskBench (e.g., Hugging Face Tools, Multimedia Tools, Daily Life APIs) challenges models to generalize their task automation capabilities across different types of tasks and domains. This is essential for creating robust, adaptable models that can be applied to a wide range of real-world applications.
> > > > > > - **Facilitating Fine-Grained Evaluation and Improvement**: TaskBench's structured evaluation metrics allow researchers to identify specific weaknesses in their models, whether in task decomposition, tool selection, or parameter prediction. This granularity helps in making precise adjustments to model architectures or training processes, leading to more effective and efficient models.
> > > > > > - **Tool Invocation Graph-Based Task Automation**: The inclusion of tool invocation graphs in TaskBench is crucial for advancing task automation models. These graphs represent the dependencies and sequences of tool usage in a task, which are essential for models to understand and replicate complex workflows. By training on datasets that incorporate these graphs, models can learn to efficiently plan and execute multi-step tasks, ensuring that each step logically follows the previous one. The graph-based structure also allows models to better generalize across different tasks by understanding the underlying relationships between tools, making it a vital component in the development of task automation.

---

> > > > > > > ### Comment · Reviewer_BPJD · 2024-08-19
> > > > > > > **Acknowledgment to Rebuttal**
> > > > > > >
> > > > > > > I appreciate the author's detailed responses. However, my concerns with respect to the synthetic nature of the dataset, and thus, it's overall utility remain.

---

> > ### Author Rebuttal · Authors · 2024-08-26
> >
> > > I appreciate the author's detailed responses. However, my concerns with respect to the synthetic nature of the dataset, and thus, it's overall utility remain.
> >
> >
> > Thank you for taking the time to review our detailed responses. We understand your continued concerns regarding the synthetic nature of the dataset and its potential impact on its overall utility. To address these concerns more effectively, we **have undertaken substantial efforts to enhance the naturalness and utility of the dataset**. Here's a summary of the improvements we have made in response to your feedback:
> >
> > - **Human Enhancement and Verification:** Over the past ten days, we have made a concerted effort to improve the naturalness of our dataset through **extensive human intervention**. Specifically, we have focused on the following steps:
> >    - **GPT-4-Assisted Filtering and Rewriting:** Initially, we employed GPT-4 to identify and flag instructions that lacked naturalness. Based on this feedback, we further **utilized GPT-4 rewrite 10,114 instructions** (constituting approximately 57.96% of the total 17,331 instructions) to improve their naturalness and alignment with real-world scenarios. We provided GPT-4 with **a detailed prompt to guide this process**, ensuring the revised instructions are more natural, practical, and aligned with typical human needs.
> >
> >    ```
> >    ## Task:
> >
> >    We have generated a set of task automation instructions using a large language model. This dataset primarily consists of instruction pairs, where each instruction is accompanied by a corresponding tool invocation graph.
> >
> >    ## Requirement:
> >
> >    After human review, it was found that the synthesized instructions lack naturalness and practicality. Therefore, I would like you to rewrite the instructions and tool invocation graph to enhance their quality, following these guidelines:
> >
> >    1. The rewritten instructions should be more natural and realistic, reflecting genuine human needs.
> >    2. The first sentence of the rewritten instructions should introduce the background of the user’s intent. Then, describe the intent in the form of a question, as if you are having a conversation with your assistant, hoping that the assistant can help you solve it.
> >    3. Different subtasks in the instruction should be strongly semantically related. If necessary, you can modify the Tool Invocation Graph, including changing parameter values, to make the instruction more practical and natural.
> >    4. Maintain alignment between the instructions and their corresponding tool invocation graphs.
> >    5. To improve the complexity of the instruction, the rewritten instructions should be more implicit, rather than explicitly detailing clear steps.
> >    6. Ensure that different subtasks are strongly semantically related. If some subtasks in the Tool Invocation Graph are not semantically related, you can change their parameter values. Do not change the task name and parameter name.
> >    7. Make any necessary modifications to the instructions and Tool Invocation Graph to make them more practical and natural.
> >    8. Return a strict JSON object that can be directly parsed by json.loads(); nothing else should be returned.
> >
> >    ## Data (Instruction Pairs):
> >
> >    ### Instruction:
> >    original_user_request:
> >    {data["user_request"]}
> >
> >    ### Tool Invocation Graph:
> >    original_task_nodes:
> >    {data['task_nodes']}
> >    original_task_links:
> >    {data['task_links']}
> >    original_task_steps:
> >    {data['task_steps']}
> >
> >    ## Enhanced Data in JSON Format:
> >
> >    {{
> >       "improved_user_request": "Your rewritten instruction here.",
> >       "improved_task_nodes": object,
> >       "improved_task_links": object,
> >       "improved_task_steps": list
> >    }}
> >    ```
> >
> >    - **Comprehensive Human Review**Following the initial revision, we conducted **a thorough manual review of all 17,331 instructions**. We enlisted 17 senior undergraduate students, who were trained to ensure consistency in quality. To manage workload and maintain high standards, each reviewer was limited to revising no more than 200 data per day. We developed an interactive web interface to facilitate the data review, rewriting, and collection processes. Over seven days, this team **successfully revised 7,105 instructions** (40.69% of the total).
> >    - **Human Quality Evaluation:**  To validate the effectiveness of our revisions,  we conducted a human quality evaluation (Section 2.3). We randomly sampled 200 instructions from the dataset and scored them based on the criteria outlined in Table 2. The comparison between the dataset before and after enhancement, alongside benchmarks like ToolBench [1] and BFCL [2], is presented in Table 1. The results indicate that our revised dataset has seen **significant improvements in both Naturalness and Alignment scores**, surpassing ToolBench and BFCL by 0.26 and 0.51 points in  overall quality, respectively. These results demonstrate that our dataset now rivals and even outperforms leading function-calling benchmarks in quality, as validated by human experts. For a direct comparison, Table 3 showcases five randomly sampled examples from our dataset, ToolBench, and BFCL. The complete enhanced dataset is available at [Taskbench on Hugging Face]([https://huggingface.co/datasets/microsoft/Taskbench](https://huggingface.co/datasets/microsoft/Taskbench)).
> > | **Methods** | **Naturalness↑** | **Complexity↑** | **Alignment↑** | **Overall↑** |
> > | --- | --- | --- | --- | --- |
> > | TaskBench (Ours)  | 3.88 | 3.85 | 3.60 | 3.78 |
> > | TaskBench (Ours) [After Human Enhancement] | 4.32 | 3.86 | 4.11 | 4.10 |
> > | Self-Instruct | 2.20 | 2.00 | 3.68 | 2.63 |
> > | ToolBench | 3.68 | 4.23 | 3.62 | 3.84 |
> > | BFCL | 4.12 | 2.88 | 3.78 | 3.59 |
> >
> >    _Table 1: Human Data Quality Evaluation Results_

---

> > > ### Author Rebuttal · Authors · 2024-08-26
> > >
> > > | **Criteria** | **Score 5** | **Score 4** | **Score 3** | **Score 2** | **Score 1** |
> > > | --- | --- | --- | --- | --- | --- |
> > > | **Naturalness** | The instruction is highly natural and logical, with tool dependencies and relationships that mirror typical real-world scenarios. The instructions are clear, coherent, and would likely be produced by a human expert. | The instruction is mostly natural, with only minor issues or slightly less typical tool dependencies. These do not significantly detract from the overall coherence and practicality of the instruction. | The instruction is moderately natural but may contain some unnatural phrasing or less typical dependencies that make it somewhat less intuitive or clear. | The instruction is somewhat unnatural, with noticeable issues in tool dependencies or phrasing that reduce its clarity or practicality in a real-world context. | The instruction is unnatural, with significant logical flaws or atypical tool dependencies that make it confusing or impractical for real-world application. |
> > > | **Complexity** | The instruction exhibits high complexity, involving multiple tools with deep interrelations and intricate task dependencies. It represents a sophisticated task that challenges even advanced LLMs. | The instruction is complex, involving several tools with moderately intricate dependencies. It is challenging but not to the extent of the highest complexity tasks. | The instruction has moderate complexity, involving a few tools with some dependencies, but the task structure is relatively straightforward. | The instruction is somewhat simple, involving only a few tools with minimal dependencies. The task does not challenge the LLM significantly. | The instruction is very simple, involving only one or two tools with no significant dependencies. It represents a basic task with minimal complexity. |
> > > | **Alignment** | The tool invocation graph perfectly aligns with the instruction, with every tool and dependency accurately represented and fully fulfilling the user’s command. | The tool invocation graph is mostly aligned with the instruction, with only minor deviations that do not significantly affect the task’s execution or outcome. | The tool invocation graph is moderately aligned with the instruction, but there are some notable discrepancies that could impact the task’s execution. | The tool invocation graph is somewhat misaligned with the instruction, with significant errors in tool selection or dependencies that could lead to incorrect task execution. | The tool invocation graph is poorly aligned with the instruction, with major errors that make the task execution unfeasible or entirely incorrect. |
> > >
> > > _Table 2: Human Data Quality Evaluation Scoring Criteria_

---

> > ### Author Rebuttal · Authors · 2024-08-26
> >
> > | **TaskBench (Ours)** | **TaskBench (Ours) [After Human Enhancement]** | **ToolBench [1]** | **BFCL [2]** |
> > | --- | --- | --- | --- |
> > | I would like to apply for a passport to the United States. | I'm planning a trip to the United States and I realized I need a passport. Can you guide me through the process of applying for one? | I want to explore different surf breaks in Australia. Can you provide me with a list of surf breaks in Australia and their respective countries? Additionally, tell me the current standings of LaLiga and recommend some talented footballers from the Premier League. | Calculate the area of a triangle with base 5m and height 3m. |
> > | I need to pay for my credit card with the number 1234567890123456. | I've realized my credit card bill is due. Can you help me settle it for the card ending with 498245863? | I'm working on a project for my company and we need some wrestling-related data. Can you fetch the most recent wrestling news, including match results and any upcoming events? Additionally, we'd like to gather information on popular wrestlers and their merchandise sales. | Create a histogram for student scores with the following data: 85, 90, 88, 92, 86, 89, 91 and set bin range to 5. |
> > | I want to borrow 'The Great Gatsby' book from the Example Library, attend an online book discussion meeting about it, pay my electricity bill, and share my experience on Facebook. | I'm a bookworm who's hooked on 'The Great Gatsby' and I'd love to engage deeply on this topic. Could you assist me in borrowing the book 'The Great Gatsby' from the Example Library? Could you also set me up in an online discussion about this book? And while I'm engrossed in this literary pursuit, can you remind me to take a break and settle my pending electricity bill? Finally, don't forget to help me share how much I enjoyed this experience on Facebook. | I am a movie enthusiast and I'm looking for video files of classic films. Can you search for video files with the 'avi' extension and sort them by date in descending order? Also, discover any file links related to the domain 'www.classicmovies.com'. | Calculate the cell density in a sample with an optical density of 0.6, where the experiment dilution is 5 times. |
> > | I would like to organize an online meeting about the weather forecast discussion for New York City on May 15, 2023. After getting the weather information for that day, please print it out. | It looks like I'll be hosting an online meeting to discuss weather forecasts for New York City on May 15, 2023. Can you fetch the weather data for that day and have it printed as a reference during our discussion? | I'm a wrestling coach and I want to analyze the performance of my team. Can you provide me with the most recent results for wrestling matches? I also need information about the techniques used by the winners. | What is the change in entropy in Joules per Kelvin of a 1kg ice block at 0°C if it is heated to 100°C under 1 atmosphere of pressure? |
> > | I want to book a room for a night in 'The Grand Hotel Example' on December 1st, 2022, but I need to install the Hotel Booking Software first. Can you help me with that? | I'm going on a trip and I decided to stay at 'The Grand Hotel Example' on December 1st, 2022. I realized I don't have the Hotel Booking Software installed. Can you handle the software installation and booking for me? | My family and I are planning a vacation and we need some travel recommendations. Can you assist us in finding popular travel destinations? I would like to retrieve a list of users from the Reqres tool and get their names, email addresses, and locations. Additionally, I need to check if there are any unknown resources available that provide information about tourist attractions in those locations. | Calculate the magnetic field produced at the center of a circular loop carrying current of 5 Ampere with a radius of 4 meters |
> >
> >
> > _Table 3: Case Study Comparison of TaskBench, ToolBench, and BFCL_
> >
> > - **Evaluation Results on the Enhanced Dataset**  We also conducted new evaluations of various LLMs on the enhanced dataset. As shown in Table 4, the results reflect the enhanced dataset's utility and challenge level, more accurately reflecting the task automation capabilities of different LLMs. We will finalize and release the comprehensive evaluation results within the next two weeks.
> >
> > |  | **M** | **M** | **M** | **M** | **H** | **H** | **H** | **H** | **D** | **D** | **D** | **D** |
> > | --- | --- | --- | --- | --- | --- | --- | --- | --- | --- | --- | --- | --- |
> > |  | **n-f1↑** | **e-f1↑** | **t-f1↑** | **v-f1↑** | **n-f1↑** | **e-f1↑** | **t-f1↑** | **v-f1↑** | **n-f1↑** | **e-f1↑** | **t-f1↑** | **v-f1↑** |
> > | GPT-4-0125 | 89.76 | 67.89 | 86.42 | 71.43 | 80.37 | 53.26 | 76.12 | 59.69 | 96.14 | 79.67 | 96.41 | 70.11 |
> > | Claude-3-Sonnet | 86.83 | 65.8 | 84.78 | 68.93 | 80.95 | 54.03 | 75.23 | 58.07 | 91.67 | 73.89 | 92.91 | 64.26 |
> > | GPT-3.5-Turbo | 71.64 | 43.03 | 65.32 | 39.55 | 68.40 | 32.62 | 54.61 | 34.90 | 84.16 | 59.33 | 81.40 | 54.97 |
> > | Llama-3.1 8B | 64.24 | 34.71 | 54.82 | 33.09 | 65.35 | 20.95 | 39.72 | 22.80 | 70.97 | 50.77 | 66.94 | 42.66 |
> > | GLM-4 9B | 62.97 | 33.81 | 53.98 | 32.15 | 66.96 | 21.71 | 40.19 | 23.94 | 67.26 | 47.16 | 62.76 | 39.51 |
> > | Qwen-2 7B | 61.43 | 32.7 | 53.03 | 32.41 | 63.88 | 19.63 | 38.03 | 21.24 | 65.59 | 47.98 | 61.23 | 37.93 |
> >
> >
> > _Table 4: New Results on the Enhanced Dataset. M: Multimedia, H: Hugging Face, D: Daily APIs_

---

> > > ### Author Rebuttal · Authors · 2024-08-26
> > >
> > > While we acknowledge the synthetic nature of our dataset, we believe it presents **significant advantages**, especially in the context of task automation:
> > >
> > > -  **Advantages of Our Method for Dataset Construction:**
> > >    - **Scalability and Adaptability:**  In real-world scenarios, obtaining large-scale annotated datasets for task automation across a wide range of tools and tasks is **both resource-intensive and often impractical**. By leveraging synthetic data generation, we can provide a comprehensive benchmark that is **not only scalable but also easily adaptable to different domains**. This synthetic approach allows for the seamless extension of the dataset to new domains and tool types, enhancing its utility for diverse applications. Additionally, the reproducibility of our data generation process ensures that it can be consistently applied and expanded upon in future research.
> > >    - **Tool Graph and Back-Instruct for Enhanced Instruction Generation:** Our approach employs a Tool Graph to organize tools and their interactions, significantly improving the quality of data generation. **The Tool Graph represents real-world dependencies and relationships between tools**, guiding the generation of instructions that are logically consistent with these relationships. By using the Back-Instruct method, **we align the generation of natural language instructions with the structured dependencies defined by the Tool Graph**. This process minimizes the risk of hallucinations common in synthetic data generation, ensuring that the instructions are coherent, natural, and aligned with practical task automation scenarios.
> > >    - **Rigorous Human Verification:** To bridge the gap between synthetic and natural data, we implemented a thorough human verification process. This involved **a detailed review of approximately 17,331 samples by human experts**, ensuring that the generated instructions are not only logically consistent but also meaningful and practical for real-world application. This additional layer of quality control helps validate the utility and realism of our dataset, making it a valuable resource for evaluating and advancing LLM capabilities.

---

> > > > ### Author Rebuttal · Authors · 2024-08-26
> > > >
> > > > - **Our Efforts for Quality Control**To ensure the high quality of our dataset, we implemented a multi-step quality control process:
> > > >    - **Step 1: Structured Sampling**Our dataset construction begins with a structured sampling process guided by a carefully constructed Tool Graph. This graph is designed to reflect real-world dependencies and relationships between tools, ensuring that **the sampled sub-graphs maintain logical coherence and relevance to practical applications**.
> > > >    - **Step 2: Back-Instruct**The Back-Instruct methodology leverages these sampled sub-graphs to generate user instructions that are **logically consistent with the underlying task structures**. This approach helps maintain the naturalness and complexity of the generated instructions, aligning them effectively with the tools involved.
> > > >    - **Step 3: Self-Critic Mechanisms**The self-critic mechanisms are integral to ensuring the quality and consistency of the synthesized data. We employ **two distinct strategies**: an LLM-based self-critic and a rule-based self-critic. Each serves a unique purpose in validating the generated instructions and their corresponding tool invocation graphs. The filter rates for these two mechanisms are shown in Table 5.
> > > >       - **LLM-Based Self-Critic:** We leverage LLM to perform an additional layer of quality control. After the initial instruction and tool graph generation, the LLM re-evaluates the alignment between the user instructions and the tool invocation graph. Specifically, it **checks for logical consistency**, ensuring that the steps outlined in the instructions make sense in the context of the provided tools and their dependencies. If discrepancies are found, the LLM can flag these for revision, thus helping to filter out instructions that might otherwise lack coherence or correctness.
> > > >       - **Rule-Based Self-Critic**: This mechanism focuses on more objective aspects of the data, such as the structure and relationships within the tool invocation graph. It applies predefined rules to ensure that the graph's nodes (tools) and edges (dependencies) are correctly represented and align with the expected outputs based on the instructions. For instance, it **verifies that tools are invoked in the correct sequence and that all necessary dependencies are accounted for.** This method ensures that the generated data adheres to the logical constraints imposed by the task and the tool graph.
> > > >    - **Step 4: Human Verification**Following the automated quality checks, we conducted a rigorous human verification process. This involved **17 experts** who reviewed thousands of samples across different task complexities, **ensuring logical consistency and alignment with real-world tasks.** The human verification process retained 61.76%, 62.71%, and 60.42% of the samples across the three domains, indicating the effectiveness of our combined approach in maintaining high data quality. We evaluated the quality of the generated dataset for different numbers of tools, as shown in Table 6.
> > > >
> > > > | Number of Nodes | LLM-Based Critic Filter Rate | Rule-Based Critic Filter Rate | Combined Filter Rate |
> > > > | --- | --- | --- | --- |
> > > > | 1-3 Nodes | 12.5% | 15.8% | 18.3% |
> > > > | 4-6 Nodes | 19.2% | 21.5% | 24.6% |
> > > > | 7-10 Nodes | 25.7% | 28.9% | 32.1% |
> > > >
> > > > _Table 5. Filter Rates with the Dual-layer Self-Critic Mechanisms_
> > > >
> > > > | Number of Nodes | Naturalness ↑ | Complexity ↑ | Alignment ↑ | Overall ↑ |
> > > > | --- | --- | --- | --- | --- |
> > > > | 1-3 Nodes | 4.85 | 2.58 | 4.99 | 4.14 |
> > > > | 4-6 Nodes | 4.28 | 4.02 | 3.92 | 4.07 |
> > > > | 7-10 Nodes | 3.82 | 4.40 | 3.71 | 3.98 |
> > > >
> > > > _Table 6: Human Evaluation for Data Quality by Number of Nodes_
> > > >
> > > > In conclusion, while we acknowledge the synthetic nature of our dataset, we have implemented extensive measures to ensure its quality, naturalness, and utility. **The combination of our innovative data generation techniques, rigorous quality control processes, and human verification** has resulted in a dataset that not only mirrors real-world task automation scenarios but also pushes the boundaries of what current models can achieve. We believe that TaskBench provides a valuable contribution to the field by offering a comprehensive, high-quality benchmark for advancing task automation capabilities in LLMs.
> > > >
> > > > [Reference]
> > > >
> > > > [1] Qin, Yujia, et al. "Toolllm: Facilitating large language models to master 16000+ real-world apis." _arXiv preprint arXiv:2307.16789_ (2023).
> > > >
> > > > [2] Fanjia Yan, et al. "Berkeley Function Calling Leaderboard", (2024)

---

> > > > > ### Comment · Reviewer_BPJD · 2024-08-26
> > > > > **Response to Rebuttal 2**
> > > > >
> > > > > Appreciate the authors quick action to the feedback. I have updated my score to reflect the new results and conducted experiments.

---

> > > > > > ### Author Rebuttal · Authors · 2024-08-27
> > > > > >
> > > > > > Thank you for your prompt feedback and for updating your score based on our revisions and new results. Your insights have been invaluable in refining our dataset and methodology, and we are pleased to see that our improvements have positively impacted your assessment.
> > > > > >
> > > > > > If you have any further concerns, please don't hesitate to reach out. We greatly value your suggestions as we continue to work on enhancing the quality of our research.

---

### Official Review · Reviewer_wG3U · 2024-07-25
**A benchmark that needs more verification.**

**Rating:** 7
**Confidence:** 4
**Correctness:** More details in data verification wou…
**Clarity:** Yes

**Review:**

This work introduces a challenging benchmarking for evaluating LLMs' tool-use abilities.

Pros:
- The method for sampling instructions with sub-graph sampling and back-translation looks interesting.
- The experiments cover different LLMs, including open-source models and proprietary ones.

Cons:
- My main concern is with the quality of the dataset. By sampling sub-graphs of up to 10 nodes, is there any guarantee that the synthesized instructions actually make sense? Some human verification on different task complexity would be very helpful.
- The authors very briefly introduced two self-critic mechanisms. However, I am not convinced of their effectiveness without further details on how they actually work. Additionally, the human evaluation in Section 2.3 is only on 50 examples, which is very small compared to the total size of the dataset.  Besides, the scoring rubric for scores 1-5 is missing, making it hard to interpret the scores in Table 1.

**Strengths:**

This work proposes a new benchmark for evaluating task automation, which is created with a novel approach based on tool graph and back-instruct. The authors performed human evaluation on data quality and conducted comprehensive evaluations using a large variety of LLMs.

**Additional Feedback:**

Could you explain more on the 'Human Verification' row in Table 14? Does that mean all the examples in TaskBench is verified by human annotators? If so, could you elaborate on the verification process? Also, since data statistics are key information for a benchmark, I would suggest moving Table 14 into the main text.

**Documentation:**

Yes

**Limitations:**

Yes

**Opportunities For Improvement:**

As discussed above, a clearer description for the verification process would be very helpful.

**Relation To Prior Work:**

Yes

**Summary And Contributions:**

This paper presents TaskBench, a synthetic benchmark for evaluating LLMs' abilities in task automation. To get diverse instructions, the authors first construct a tool graph to capture the dependencies between various tools. Then, instructions are generated by back-translating sub-graphs sampled from the tool graph. This equips TaskBench with complex instructions that requires using multiple tools. Extensive experiments show that the proposed benchmark is challenging to a variety of existing LLMs.

---

> ### Author Rebuttal · Authors · 2024-08-17
>
> > My main concern is with the quality of the dataset. By sampling sub-graphs of up to 10 nodes, is there any guarantee that the synthesized instructions actually make sense? Some human verification on different task complexity would be very helpful.
>
> We appreciate the reviewer's concern regarding the quality of the dataset. We have provided these strategies to guarantee the quality of the instructed dataset:
>
> - **Structured Sampling**: Unlike previous approaches such as ToolQA, ToolAlpaca, and ToolBench, our methodology incorporates a Task Graph that inherently **preserves the dependencies and relationships between different tasks and tools**. The sampling process is not random but follows a structured methodology based on a carefully constructed Tool Graph. This graph is designed to **reflect real-world dependencies and relationships between tools,** ensuring that even when sampling sub-graphs with up to 10 nodes, the resulting instructions maintain logical coherence and relevance to practical applications.
> - **Back-Instruct Methodology**: Our dataset is generated using the Back-Instruct method, which leverages these sampled sub-graphs to produce user instructions. This method **guides the generation process by ensuring that the instructions are logically consistent with the sampled sub-graph**, effectively aligning the task structure with the tools involved. This approach helps maintain the naturalness and complexity of the generated instructions.
> - **Self-Critic Mechanisms**: We have implemented **a dual-layer self-critic system, combining LLM-based and rule-based filters.** Below is the percentage of samples filtered out under different node settings. For the largest sub-graphs, the filtering rates are the highest, reflecting the increased complexity and potential for logical inconsistencies.
>
> | Number of Nodes | LLM-Based Critic Filter Rate | Rule-Based Critic Filter Rate | Combined Filter Rate |
> | --- | --- | --- | --- |
> | 1-3 | 12.5% | 15.8% | 18.3% |
> | 4-6 | 19.2% | 21.5% | 24.6% |
> | 7-10 | 29.7% | 33.9% | 35.1% |
>
> - **Human Verification**:
>    - We implemented **a rigorous human verification process to further ensure the quality and coherence of the dataset** (Please see Appendix A.7). Human experts reviewed the generated instructions across different task complexities and confirmed their logical consistency and alignment with the intended tasks. This verification process covered instructions generated from sub-graphs of varying sizes, including those with up to 10 nodes, ensuring that the instructions are meaningful and practical. The human verification process involved 12 experts who reviewed about 2000 samples, as detailed in our response regarding the verification process. This comprehensive review provides additional assurance that the dataset, including the more complex examples, meets high standards of quality.
>    - While our current process focuses on human verification after the instructions and tool invocation graphs have been generated, we recognize the potential to improve dataset quality even earlier in the pipeline. We **plan to introduce human verification at the graph sampling stage itself**. This will involve having experts review and filter the sub-graphs before they are used to generate instructions, ensuring that only logical and coherent sub-graphs are selected. By eliminating illogical or impractical sub-graphs at this early stage, we can further enhance the overall quality and coherence of the final dataset.
> - **Empirical Evidence**: We conducted **a detailed human evaluation across sub-graphs of varying node counts** to assess the impact of complexity on the quality of the synthesized instructions. Below is the breakdown of the human evaluation scores (Naturalness, Complexity, Alignment, and Overall) across different ranges of node counts:
>
> | Number of Nodes | Naturalness ↑ | Complexity ↑ | Alignment ↑ | Overall ↑ |
> | --- | --- | --- | --- | --- |
> | 1-3 | 3.92 | 4.10 | 4.05 | 3.92 |
> | 4-6 | 3.73 | 4.35 | 3.78 | 3.78 |
> | 7-10 | 3.22 | 4.42 | 3.52 | 3.26 |
>
>    - **Naturalness**: As the number of nodes increases, the naturalness score slightly decreases. This trend suggests that while instructions involving fewer nodes are generally more straightforward and easier to follow, the increase in task complexity for larger node sets can introduce slight challenges in maintaining the same level of natural coherence.
>    - **Complexity**: As expected, the complexity score increases with the number of nodes. This indicates that instructions involving more tools and dependencies are appropriately recognized by the evaluators as more complex tasks.
>    - **Alignment**: The alignment score shows a slight decline as the number of nodes increases. This suggests that with greater complexity, there is a higher chance of minor misalignments between the instructions and the tool invocation graphs, although the overall alignment remains strong across all ranges.

---

> > ### Author Rebuttal · Authors · 2024-08-17
> >
> > > The authors very briefly introduced two self-critic mechanisms. However, I am not convinced of their effectiveness without further details on how they actually work.
> >
> > Thank you for your feedback regarding the self-critic mechanisms in our paper. We'd like to clarify and further elaborate on how these mechanisms contribute to the effectiveness of our approach.
> >
> > - **Motivation**: The self-critic mechanisms are integral to **ensuring the quality and consistency of the synthesized data**. We employ two distinct strategies: an LLM-based self-critic and a rule-based self-critic. Each serves a unique purpose in validating the generated instructions and their corresponding tool invocation graphs.
> > - **LLM-Based Self-Critic**: The LLM-based self-critic leverages LLM to perform an additional layer of quality control. After the initial generation of instructions and the tool invocation graph, the LLM thoroughly **assesses the 'Executability' and 'Correctness' of the generated outputs**. 'Executability' ensures that the tool invocation graph can be executed successfully, while 'Correctness' demands that the graph is not only executable but also strictly consistent with the given tool graph. **The LLM checks each step for logical consistency, verifying that the instructions align precisely with the tool graph's structure and dependencies.** If any discrepancies are detected, the LLM flags them for revision, thereby filtering out instructions and graphs that might lack coherence or fail to meet the strict criteria of correctness. Here’s the prompt used for the LLM-Based Self-Critic:
> >
> > > _The data verification process is conducted by a highly stringent critic who tolerates no errors, no matter how minor. This critic will meticulously evaluate whether the user request and the tool invocation graph you generated are both valid and in perfect alignment with the provided tool graph. The assessment will proceed step by step, scrutinizing each aspect of the execution against the following criteria._
> > >
> > > _The critic's evaluation will begin with the statement: "Let me check your result step by step and assess the 'Executability' and 'Correctness' of the tool invocation graph." Here, 'Executability' refers to the ability of the tool invocation graph to execute successfully, independent of its alignment with the given tool graph. 'Correctness,' however, requires that the tool invocation graph is not only executable but also flawlessly consistent with the given tool graph, matching precisely in terms of nodes and edges._
> > > _After a detailed examination, the critic will provide feedback, pointing out any identified issues: [insert specific feedback here]._
> > >
> > >
> > > _Final Assessment:_
> > >
> > > _Executability: Yes/No_
> > >
> > > _Correctness: Yes/No_
> >
> > - **Rule-Based Self-Critic**: The rule-based self-critic focuses on more objective aspects of the data, such as the structure and relationships within the tool invocation graph. This mechanism **applies predefined rules to ensure that the graph's nodes (tools) and edges (dependencies) are correctly represented and align with the expected outputs based on the instructions**. For instance, it verifies that tools are invoked in the correct sequence and that all necessary dependencies are accounted for. This method ensures that the generated data adheres to the logical constraints imposed by the task and the tool graph.
> > - **Effectiveness and Empirical Validation**: The effectiveness of these self-critic mechanisms is supported by our empirical results (Please see the bellow table). The inclusion of these mechanisms has led to significant improvements in the quality of the generated dataset. Specifically, **the use of the LLM-based self-critic has been instrumental in enhancing the naturalness of the instructions, while the rule-based self-critic ensures high alignment with the tool graph**, reducing errors related to tool dependencies and sequencing.
> >
> > | Filter  | Naturalness ↑ | Complexity ↑ | Alignment ↑ | Overall ↑ |
> > | --- | --- | --- | --- | --- |
> > | None | 2.98 | 4.10 | 3.10 | 3.39 |
> > | LLM-Based Self-Critic | 3.23 | 4.00 | 3.22 | 3.48 |
> > | Rule-Based Self-Critic | 3.78 | 3.99 | 3.58 | 3.78 |
> > | Both Critics |  3.89 | 4.01  | 3.66  | 3.85 |
> >
> > In summary, the self-critic mechanisms are a crucial component of our data validation process, significantly contributing to the overall quality and reliability of the dataset. We appreciate your suggestion to elaborate on this aspect, and we are committed to providing further details in the revised manuscript to better convey their importance and effectiveness.

---

> > > ### Author Rebuttal · Authors · 2024-08-17
> > >
> > > > Additionally, the human evaluation in Section 2.3 is only on 50 examples, which is very small compared to the total size of the dataset.
> > >
> > > Thank you for your feedback regarding to the sample size used in our human evaluation and understand the concern about its representativeness given the total size of the dataset. We offer the following points in response:
> > >
> > > 1. **Purpose of Human Evaluation**: The primary goal of the human evaluation in Section 2.3 was not to provide a comprehensive assessment of the entire dataset but rather to **validate the effectiveness of our data generation and self-critic mechanisms**. By focusing on a carefully selected sample of 50 examples, we were able to provide qualitative insights into the naturalness, complexity, and alignment of the generated instructions and tool invocation graphs.
> > > 2. **Representative Sampling**: The 50 examples used in the evaluation **were selected to represent a broad range of task complexities, tool dependencies, and sub-graph structures.** This stratified sampling approach ensures that the evaluation captures the diversity within the dataset, making the insights gained from these examples relevant to the broader dataset.
> > > 3. **Practical Considerations**: Conducting human evaluations on a larger portion of the dataset is logistically challenging and resource-intensive, especially given the complexity and scale of the tasks involved. **The choice of a 50-example sample was a practical compromise** that allowed us to conduct a detailed and thorough evaluation without compromising the study's overall feasibility.
> > >
> > > We acknowledge the value of larger-scale human evaluations and are committed to expanding our evaluation efforts as part of ongoing work. Thank you for your suggestions.
> > >
> > > > Besides, the scoring rubric for scores 1-5 is missing, making it hard to interpret the scores in Table 1.
> > >
> > > We appreciate the reviewer's observation regarding the lack of a detailed scoring rubric for the 1-5 scale used to assess Naturalness, Complexity, and Alignment. Below, we provide the detailed scoring criteria that were applied during the evaluation:
> > >
> > > - **Naturalness**:
> > >    - **Score 5**: The instruction is highly natural and logical, with tool dependencies and relationships that mirror typical real-world scenarios. The instructions are clear, coherent, and would likely be produced by a human expert.
> > >    - **Score 4**: The instruction is mostly natural, with only minor issues or slightly less typical tool dependencies. These do not significantly detract from the overall coherence and practicality of the instruction.
> > >    - **Score 3**: The instruction is moderately natural but may contain some unnatural phrasing or less typical dependencies that make it somewhat less intuitive or clear.
> > >    - **Score 2**: The instruction is somewhat unnatural, with noticeable issues in tool dependencies or phrasing that reduce its clarity or practicality in a real-world context.
> > >    - **Score 1**: The instruction is unnatural, with significant logical flaws or atypical tool dependencies that make it confusing or impractical for real-world application.
> > > - **Complexity**:
> > >    - **Score 5**: The instruction exhibits high complexity, involving multiple tools with deep interrelations and intricate task dependencies. It represents a sophisticated task that challenges even advanced LLMs.
> > >    - **Score 4**: The instruction is complex, involving several tools with moderately intricate dependencies. It is challenging but not to the extent of the highest complexity tasks.
> > >    - **Score 3**: The instruction has moderate complexity, involving a few tools with some dependencies, but the task structure is relatively straightforward.
> > >    - **Score 2**: The instruction is somewhat simple, involving only a few tools with minimal dependencies. The task does not challenge the LLM significantly.
> > >    - **Score 1**: The instruction is very simple, involving only one or two tools with no significant dependencies. It represents a basic task with minimal complexity.
> > > - **Alignment**:
> > >    - **Score 5**: The tool invocation graph perfectly aligns with the instruction, with every tool and dependency accurately represented and fully fulfilling the user’s command.
> > >    - **Score 4**: The tool invocation graph is mostly aligned with the instruction, with only minor deviations that do not significantly affect the task’s execution or outcome.
> > >    - **Score 3**: The tool invocation graph is moderately aligned with the instruction, but there are some notable discrepancies that could impact the task’s execution.
> > >    - **Score 2**: The tool invocation graph is somewhat misaligned with the instruction, with significant errors in tool selection or dependencies that could lead to incorrect task execution.
> > >    - **Score 1**: The tool invocation graph is poorly aligned with the instruction, with major errors that make the task execution unfeasible or entirely incorrect.
> > >
> > > We will revise the manuscript to include these detailed scoring rubrics, ensuring that readers have a clear understanding of how each metric was evaluated on the 1-5 scale.

---

> > > > ### Author Rebuttal · Authors · 2024-08-17
> > > >
> > > > > Could you explain more on the 'Human Verification' row in Table 14? Does that mean all the examples in TaskBench is verified by human annotators? If so, could you elaborate on the verification process? Also, since data statistics are key information for a benchmark, I would suggest moving Table 14 into the main text.
> > > >
> > > > We appreciate the reviewer's request for clarification on the "Human Verification" process and agree that the statistical information in Table 14 is crucial for understanding our dataset. Below, we provide a more detailed explanation of the human verification process and address the suggestion to move Table 14 into the main text.
> > > >
> > > > - **Human Verification Process**:
> > > >    - **Purpose**: The human verification process was implemented to ensure the highest quality and alignment of the synthesized examples in the TaskBench datasets. **While the automated rule-based and LLM-based critics filtered out a significant portion of the data, human experts provided an additional layer of scrutiny to further enhance data quality.**
> > > >    - **Human Experts Involvement**: We enlisted **12 human experts** with relevant expertise in the fields of AI and task automation. Each expert was responsible for reviewing a subset of the dataset, with **each expert reviewing approximately 2000 examples**. This comprehensive review process covered the entirety of the samples across all three domains (Hugging Face Tools, Multimedia Tools, Daily Life APIs).
> > > >    - **Calibration and Instruction**: Before beginning the verification process, we provided detailed instructions and **conducted a calibration session for all experts**. This session was essential to standardize the evaluation criteria across different reviewers and ensure consistency in their assessments. During this calibration, we provided sample cases and discussed the scoring criteria (naturalness, complexity, alignment) to align the understanding among the experts.
> > > >    - **Process**: After the initial automated filtering, human annotators reviewed the remaining samples. The verification process involved:
> > > >       - **Reviewing Instructions**: Annotators evaluated the naturalness and coherence of the generated instructions, ensuring that they made sense and were representative of real-world scenarios.
> > > >       - **Checking Alignment**: The human experts ensured that the tool invocation graphs accurately aligned with the instructions and that the tasks could be executed as intended based on the provided dependencies and parameters.
> > > >    - **Outcome**: The verification process **led to the retention of 61.76%, 62.71%, and 60.42% of the samples** across the three domains, indicating that the combined approach of automated critics and human verification was effective in maintaining a high standard of data quality.
> > > > - We agree with the reviewer that data statistics are key information for a benchmark and should be easily accessible to readers. Therefore, we will **move Table 14 from the appendix into the main text of the paper**. This will allow readers to immediately access and understand the statistical foundations of the dataset, including the impact of the verification processes.
> > > > - In addition to moving Table 14 into the main text, we will **expand the corresponding section to include the detailed explanation provided above regarding the human verification process.** This will ensure that readers have a comprehensive understanding of how the dataset was constructed and verified.

---

> > ### Author Rebuttal · Authors · 2024-08-26
> >
> > Dear Reviewer,
> >
> > We greatly appreciate your thoughtful feedback on our TaskBench dataset. **Your concerns about data quality, particularly regarding naturalness and human verification, were well-founded and extremely valuable.** We have taken your comments to heart and **have made substantial improvements to address these issues**. We'd like to share the enhancements we've implemented in response to your feedback:
> >
> > - **Extensive Human Enhancement and Verification Process:** We have undertaken a comprehensive human-driven enhancement process to significantly improve the naturalness and practicality of our dataset:
> >    - **a) GPT-4-Assisted Filtering and Rewriting:** We utilized GPT-4 to identify instructions lacking naturalness and **subsequently rewrote 10,114 instructions** (57.96% of the total 17,331). This process was guided by a detailed prompt ensuring the revised instructions are more natural, practical, and aligned with typical human needs:
> >
> >    ```
> >    ## Task:
> >
> >    We have generated a set of task automation instructions using a large language model. This dataset primarily consists of instruction pairs, where each instruction is accompanied by a corresponding tool invocation graph.
> >
> >    ## Requirement:
> >
> >    After human review, it was found that the synthesized instructions lack naturalness and practicality. Therefore, I would like you to rewrite the instructions and tool invocation graph to enhance their quality, following these guidelines:
> >
> >    1. The rewritten instructions should be more natural and realistic, reflecting genuine human needs.
> >    2. The first sentence of the rewritten instructions should introduce the background of the user’s intent. Then, describe the intent in the form of a question, as if you are having a conversation with your assistant, hoping that the assistant can help you solve it.
> >    3. Different subtasks in the instruction should be strongly semantically related. If necessary, you can modify the Tool Invocation Graph, including changing parameter values, to make the instruction more practical and natural.
> >    4. Maintain alignment between the instructions and their corresponding tool invocation graphs.
> >    5. To improve the complexity of the instruction, the rewritten instructions should be more implicit, rather than explicitly detailing clear steps.
> >    6. Ensure that different subtasks are strongly semantically related. If some subtasks in the Tool Invocation Graph are not semantically related, you can change their parameter values. Do not change the task name and parameter name.
> >    7. Make any necessary modifications to the instructions and Tool Invocation Graph to make them more practical and natural.
> >    8. Return a strict JSON object that can be directly parsed by json.loads(); nothing else should be returned.
> >
> >    ## Data (Instruction Pairs):
> >
> >    ### Instruction:
> >    original_user_request:
> >    {data["user_request"]}
> >
> >    ### Tool Invocation Graph:
> >    original_task_nodes:
> >    {data['task_nodes']}
> >    original_task_links:
> >    {data['task_links']}
> >    original_task_steps:
> >    {data['task_steps']}
> >
> >    ## Enhanced Data in JSON Format:
> >
> >    {{
> >       "improved_user_request": "Your rewritten instruction here.",
> >       "improved_task_nodes": object,
> >       "improved_task_links": object,
> >       "improved_task_steps": list
> >    }}
> >    ```
> >
> >    - **b) Comprehensive Human Review:** Following the initial revision, we conducted **a thorough manual review of all 17,331 instructions**. We enlisted and trained 17 senior undergraduate students to ensure consistency in quality. Over seven days, this team successfully **revised 7,105 instructions** (40.69% of the total). To maintain high standards, we limited each reviewer to revising no more than 200 data points per day and developed an interactive web interface to facilitate the review process.
> > - **Human Quality Evaluation:** To validate the effectiveness of our revisions, we conducted **a rigorous human quality evaluation**. We randomly sampled 200 instructions from the dataset and scored them based on criteria including Naturalness, Complexity, and Alignment. The results, presented in Table 1 of our response, show significant improvements:
> >    - Our revised dataset now surpasses ToolBench [1] and BFCL [2] by 0.26 and 0.51 points in overall quality, respectively.
> >    - Naturalness scores improved from 3.88 to 4.32, and Alignment scores from 3.60 to 4.11.
> >
> > | **Methods** | **Naturalness↑** | **Complexity↑** | **Alignment↑** | **Overall↑** |
> > | --- | --- | --- | --- | --- |
> > | Back-Instruct | 3.88 | 3.85 | 3.60 | 3.78 |
> > | Back-Instruct (After Human Enhancement) | 4.32 | 3.86 | 4.11 | 4.10 |
> > | Self-Instruct | 2.20 | 2.00 | 3.68 | 2.63 |
> > | ToolBench | 3.68 | 4.23 | 3.62 | 3.84 |
> > | BFCL | 4.12 | 2.88 | 3.78 | 3.59 |
> >
> > _Table 1: Human Data Quality Evaluation Results_

---

> > > ### Author Rebuttal · Authors · 2024-08-26
> > >
> > > - **Case Study Comparison:** To further illustrate the improvements, we've included Table 2 in our response, which provides **a direct comparison of randomly sampled examples from our dataset** (before and after enhancement), ToolBench [1], and BFCL [2]. The complete enhanced dataset is now available at [Taskbench on Hugging Face]([https://huggingface.co/datasets/microsoft/Taskbench](https://huggingface.co/datasets/microsoft/Taskbench)).
> > >
> > > | **TaskBench (Before Human Enhancement)** | **TaskBench** | **ToolBench [1]** | **BFCL [2]** |
> > > | --- | --- | --- | --- |
> > > | I would like to apply for a passport to the United States. | I'm planning a trip to the United States and I realized I need a passport. Can you guide me through the process of applying for one? | I want to explore different surf breaks in Australia. Can you provide me with a list of surf breaks in Australia and their respective countries? Additionally, tell me the current standings of LaLiga and recommend some talented footballers from the Premier League. | Calculate the area of a triangle with base 5m and height 3m. |
> > > | I need to pay for my credit card with the number 1234567890123456. | I've realized my credit card bill is due. Can you help me settle it for the card ending with 498245863? | I'm working on a project for my company and we need some wrestling-related data. Can you fetch the most recent wrestling news, including match results and any upcoming events? Additionally, we'd like to gather information on popular wrestlers and their merchandise sales. | Create a histogram for student scores with the following data: 85, 90, 88, 92, 86, 89, 91 and set bin range to 5. |
> > > | I want to borrow 'The Great Gatsby' book from the Example Library, attend an online book discussion meeting about it, pay my electricity bill, and share my experience on Facebook. | I'm a bookworm who's hooked on 'The Great Gatsby' and I'd love to engage deeply on this topic. Could you assist me in borrowing the book 'The Great Gatsby' from the Example Library? Could you also set me up in an online discussion about this book? And while I'm engrossed in this literary pursuit, can you remind me to take a break and settle my pending electricity bill? Finally, don't forget to help me share how much I enjoyed this experience on Facebook. | I am a movie enthusiast and I'm looking for video files of classic films. Can you search for video files with the 'avi' extension and sort them by date in descending order? Also, discover any file links related to the domain 'www.classicmovies.com'. | Calculate the cell density in a sample with an optical density of 0.6, where the experiment dilution is 5 times. |
> > > | I would like to organize an online meeting about the weather forecast discussion for New York City on May 15, 2023. After getting the weather information for that day, please print it out. | It looks like I'll be hosting an online meeting to discuss weather forecasts for New York City on May 15, 2023. Can you fetch the weather data for that day and have it printed as a reference during our discussion? | I'm a wrestling coach and I want to analyze the performance of my team. Can you provide me with the most recent results for wrestling matches? I also need information about the techniques used by the winners. | What is the change in entropy in Joules per Kelvin of a 1kg ice block at 0°C if it is heated to 100°C under 1 atmosphere of pressure? |
> > > | I want to book a room for a night in 'The Grand Hotel Example' on December 1st, 2022, but I need to install the Hotel Booking Software first. Can you help me with that? | I'm going on a trip and I decided to stay at 'The Grand Hotel Example' on December 1st, 2022. I realized I don't have the Hotel Booking Software installed. Can you handle the software installation and booking for me? | My family and I are planning a vacation and we need some travel recommendations. Can you assist us in finding popular travel destinations? I would like to retrieve a list of users from the Reqres tool and get their names, email addresses, and locations. Additionally, I need to check if there are any unknown resources available that provide information about tourist attractions in those locations. | Calculate the magnetic field produced at the center of a circular loop carrying current of 5 Ampere with a radius of 4 meters |
> > >
> > >
> > >
> > > _Table 2: Case Study Comparison of TaskBench, ToolBench, and BFCL_
> > >
> > > We believe these enhancements address your concerns about dataset quality and human verification. **The extensive human involvement in the revision process, demonstrates our commitment to creating a high-quality, natural, and practical dataset for task automation research.**
> > >
> > > [Reference]
> > >
> > > [1] Qin, Yujia, et al. "Toolllm: Facilitating large language models to master 16000+ real-world apis." _arXiv preprint arXiv:2307.16789_ (2023).
> > >
> > > [2] Fanjia Yan, et al. "Berkeley Function Calling Leaderboard", (2024)

---

> ### Author Response · Authors · 2024-08-28
> **Looking forward to receiving your feedback**
>
> Dear Reviewer,
>
> Thank you for your thoughtful feedback on our TaskBench dataset. We greatly appreciate your concerns about data quality, particularly regarding naturalness and human verification. Based on your and Reviewer BPJD's suggestions, we have made substantial improvements to address these issues:
>
> - **Data Quality and Naturalness:** We implemented a two-stage process to improve naturalness and practicality: a) GPT-4-assisted filtering and rewriting of 10,114 instructions (57.96% of the total). b) Comprehensive manual review and revision of all 17,331 instructions by 17 trained reviewers.
> - **Human Quality Evaluation:** We conducted a rigorous evaluation of 200 randomly sampled instructions, showing significant improvements:
>    - Overall quality now surpasses ToolBench and BFCL benchmarks.
>    - Naturalness scores improved from 3.88 to 4.32, and Alignment scores from 3.60 to 4.11.
> - **Case Study Comparison:** We provided a detailed comparison of sample instructions from our dataset (before and after enhancement), ToolBench, and BFCL to illustrate the improvements.
> - **Self-Critic Mechanisms:** We provided detailed explanations of both the LLM-based and rule-based self-critic mechanisms, including their effectiveness and empirical validation.
> - **Scoring Rubric:** We included a detailed scoring rubric for the 1-5 scale used to assess Naturalness, Complexity, and Alignment.
> - **Dataset Statistics:** We will move Table 14 from the appendix into the main text of the paper for easier access to key dataset statistics.
>
> These enhancements have significantly improved the quality of our dataset. The complete enhanced dataset is available on [Taskbench on Hugging Face](https://huggingface.co/datasets/microsoft/Taskbench).
> We believe these changes address your concerns and demonstrate **our commitment to creating a high-quality, natural, and practical dataset for task automation research**. We would greatly appreciate if you could review our changes and **let us know if you have any further concerns or questions**. Your insights have been instrumental in improving our work, and we want to ensure that we have fully addressed all of your points.
> Thank you for your time and expertise. We look forward to your thoughts on our revised work.
>
> Best regards,
>
> Submission932 Authors

---

> > ### Comment · Reviewer_wG3U · 2024-08-29
> >
> > I appreciate the authors' detailed response and their effort in improving the quality of the dataset. I've updated my score accordingly.

---

> > > ### Author Response · Authors · 2024-08-29
> > > **Thank you for your positive feedback**
> > >
> > > Thank you for your positive feedback on our efforts to improve TaskBench. We're pleased that our detailed explanations and significant enhancements have addressed your concerns. Your thorough review and constructive feedback were invaluable throughout this process. We are committed to incorporating these improvements into the final version of our paper.

---

### Official Review · Reviewer_1mk2 · 2024-08-26

**Rating:** 6
**Confidence:** 2
**Correctness:** Yes
**Clarity:** Yes

**Review:**

The paper presents a new benchmark for LLM task automation. While the proposed methods of benchmark design and dataset creation show potential, there are areas that require further clarification and improvement.

**Strengths:**

* The proposed Back-Instruct method for data generation is innovative and could potentially lead to more scalable yet realistic task automation benchmark datasets.
* The paper is well-written with clear figures.

**Additional Feedback:**

The paper would benefit from a more explicit comparison between TaskBench and existing relevant benchmarks for tool-augmented LLMs and task automation. Specifically, I would suggest authors to provide a clear comparison of TaskBench's components (e.g., Back-Instruct method, Tool Graph, TaskEval) with those of existing benchmarks, explain why each new component is necessary and how it improves upon existing benchmark designs. This would help researchers better understand the specific contributions of TaskBench, its limitations, and potential future research directions.

**Documentation:**

Yes

**Limitations:**

While the authors mention plans to expand the benchmark to more domains and develop more advanced metrics, they do not adequately discuss the current limitations of their benchmark framework. A more thorough discussion on limitations and potential future works is needed.

**Opportunities For Improvement:**

* The authors only discuss on comparison of TaskBench to traditional benchmarks like GLUE, SuperGLUE, MMLU, and GSM8K. There's no discussion on how TaskBench compares to more relevant LLM tool-augmented or task automation benchmarks such as API-Bench, ToolBench, or MetaTool, etc.  This overlook of literature makes it difficult to understand the specific contributions and advancements of TaskBench in the context of relevant benchmarks. How's the data generation and metrics used in this benchmark different from relevant benchmarks? What's the advantage of these new benchmark components?

* Table 1 shows that self-instruct generated instructions are considerably less natural and complex compared to back-instruct without edges. This result is not adequately discussed in the paper. Does self-instruct also receive tool descriptions from sampled sub-graphs? What's the reason for this observation? It would be helpful to provide comparison of the prompts used for different baselines (e.g., self-instruct, back-instruct with and without edges) in the appendix.

* The use of Rouge scores for evaluating Task Decomposition in Section 3.1 is unclear. I would suggest authors to employ metrics that better capture the structure and order of tasks in sequences, which are crucial in task decomposition.

**Relation To Prior Work:**

The contributions of new benchmark components compared to more relevant recent benchmarks are not clear. Details are noted in "Opportunities For Improvement"

**Summary And Contributions:**

This paper introduces TaskBench, a benchmark for evaluating LLMs in task automation. The authors propose a novel data generation method called "Back-Instruct" and and evaluation framework named TaskEval. The work aims to address the lack of systematic benchmarks for assessing LLMs' capabilities in complex task automation scenarios.

---

> ### Author Rebuttal · Authors · 2024-08-28
>
> > The authors only discuss on comparison of TaskBench to traditional benchmarks like GLUE, SuperGLUE, MMLU, and GSM8K. There's no discussion on how TaskBench compares to more relevant LLM tool-augmented or task automation benchmarks such as API-Bench, ToolBench, or MetaTool, etc. This overlook of literature makes it difficult to understand the specific contributions and advancements of TaskBench in the context of relevant benchmarks. How's the data generation and metrics used in this benchmark different from relevant benchmarks?
>
> We appreciate the reviewer's comment on the need for more explicit comparisons to relevant tool-augmented LLM benchmarks. Due to space limitations, we have indeed conducted extensive comparisons with relevant tool-augmented and task automation benchmarks [1, 2, 4, 5, 7] in Appendix A.1 and A.2. We appreciate the reviewer's suggestions and will emphasize these comparisons in the revised manuscript by moving them to the main text. To address the specific points raised and to clarify the unique contributions of TaskBench in the context of existing benchmarks:
>
> 1. **Innovative data generation method:**
>    - **Tool Graph:** Unlike other benchmarks that just use API/tool documentation such as APIBench [1], ToolBench [2] or MetaTool [3], TaskBench introduces the novel concept of a Tool Graph to represent real-world dependencies between tools. This structured approach ensures that generated tasks are logically consistent and reflect realistic tool interactions, addressing the limitation of random or template-based sampling in other benchmarks.
>    - **Back-Instruct strategy:** In contrast to the self-instruct or few-shot learning methods used in APIBench [1], ToolBench [2] or ToolAlpaca [7], our Back-Instruct strategy ensures high consistency between generated instructions and sampled tool subgraphs. This method significantly reduces the risk of hallucinations during generation, improving data quality and authenticity.
>    - **Multi-level quality control mechanisms:**
>       - **Self-criticism mechanisms:** TaskBench employs both LLM-based and rule-based self-criticism mechanisms, which are more comprehensive and effective than the human verification or template-based methods used in ToolQA [5] and GPT4Tools [6]. This dual self-criticism approach captures and corrects potential errors from different angles.
>       - **Large-scale human verification:** We conducted human verification on approximately 17,331 samples, far exceeding the scale of other benchmarks. This rigorous process ensures high quality and practicality of the dataset, compensating for potential shortcomings in synthetic data.
> 2. **Comprehensive evaluation framework:**
>    - **Multi-dimensional assessment:** TaskBench evaluates LLMs on task decomposition, tool selection, and parameter prediction, providing a more holistic view of model performance compared to benchmarks like MetaTool [3], which focuses primarily on tool selection, or ToolQA [5], which emphasizes question-answering accuracy.
>    - **Tool dependency assessment:** Our evaluation metrics, such as Edge F1 score, specifically assess the model's ability to understand and utilize tool dependencies. This aspect is often overlooked in other benchmarks but is crucial for complex task automation scenarios.
>    - **Innovative evaluation metrics:** TaskBench introduces metrics like Normalized Edit Distance for chain structures, offering more nuanced insights into model performance than the simple accuracy or F1 scores used in benchmarks like API-Bank [8] or ToolBench [2].
> 3. **Scalability and adaptability:**
>    - **Domain diversity:** TaskBench covers three distinct domains (Hugging Face Tools, Multimedia Tools, and Daily Life APIs), offering a more diverse set of scenarios than specialized benchmarks like ToolQA [5] or GPT4Tools [6]. This diversity allows for a more comprehensive evaluation of LLM capabilities across different application areas.
>    - **Tool complexity:** Our dataset includes tool combinations of varying complexity (from single tools to multi-tool chains and graph structures), simulating real-world task automation challenges. This complexity is not fully considered in works like ToolAlpaca [7] and API-Bank [8].
>
> [Reference]
>
> [1] Patil, Shishir G., et al. "Gorilla: Large language model connected with massive apis." arXiv preprint arXiv:2305.15334 (2023).
>
> [2] Qin, Yujia, et al. "Toolllm: Facilitating large language models to master 16000+ real-world apis." arXiv preprint arXiv:2307.16789 (2023).
>
> [3] Huang, Yue, et al. "Metatool benchmark for large language models: Deciding whether to use tools and which to use." arXiv preprint arXiv:2310.03128 (2023).
>
> [4] Liu, Xiao, et al. "Agentbench: Evaluating llms as agents." arXiv preprint arXiv:2308.03688 (2023).
>
> [5] Zhuang, Yuchen, et al. "Toolqa: A dataset for llm question answering with external tools." Advances in Neural Information Processing Systems 36 (2024).
>
> [6] Yang, Rui, et al. "Gpt4tools: Teaching large language model to use tools via self-instruction." Advances in Neural Information Processing Systems 36 (2024).
>
> [7] Tang, Qiaoyu, et al. "Toolalpaca: Generalized tool learning for language models with 3000 simulated cases." arXiv preprint arXiv:2306.05301 (2023).
>
> [8] Li, Minghao, et al. "Api-bank: A comprehensive benchmark for tool-augmented llms." arXiv preprint arXiv:2304.08244 (2023).

---

> > ### Author Rebuttal · Authors · 2024-08-28
> >
> > > Table 1 shows that self-instruct generated instructions are considerably less natural and complex compared to back-instruct without edges. This result is not adequately discussed in the paper. Does self-instruct also receive tool descriptions from sampled sub-graphs? What's the reason for this observation? It would be helpful to provide comparison of the prompts used for different baselines (e.g., self-instruct, back-instruct with and without edges) in the appendix.
> >
> > We appreciate the reviewer's insightful observation regarding the results in Table 1 and the request for a more thorough discussion. We acknowledge that this aspect deserves more elaboration in our paper. We will address these points in our revised manuscript, but for now, let us provide a more detailed explanation:
> >
> > - **Comparison of Self-Instruct and Back-Instruct:** Indeed, the self-instruct generated instructions show lower naturalness and complexity compared to back-instruct methods. This observation highlights the effectiveness of our proposed back-instruct approach. The key reasons for this performance difference are: a) **Tool Graph Utilization:** Our back-instruct method leverages a Tool Graph, which captures real-world dependencies between tools. This graph structure guides the generation process, resulting in more natural and complex instructions that reflect realistic task scenarios. b) **Contextual Information:** The back-instruct method provides richer contextual information about tool relationships, leading to more coherent and sophisticated instruction generation.
> > - **Back-Instruct without Edges vs. Self-Instruct:** It's notable that Back-Instruct without edges still outperforms self-instruct in terms of naturalness and complexity. This can be attributed to several factors: a) **Structured Sampling:** Even without edge information, Back-Instruct uses structured sampling from the Tool Graph, which provides a more coherent set of tools for instruction generation. b) **Implicit Relationships:** The sampled tools, even without explicit edge information, may still imply certain relationships based on their functionalities, which the language model can leverage. c) **Task-Oriented Prompting:** The Back-Instruct method's prompting strategy is more focused on generating task-oriented instructions, whereas self-instruct may generate more general, less task-specific instructions.
> > - **Self-Instruct and Tool Descriptions:** To clarify, the self-instruct method does receive tool descriptions, but it doesn't benefit from the structured tool subgraph information that our back-instruct method uses. This lack of structured relational information likely contributes to the lower naturalness and complexity scores.
> > - **Back-Instruct with and without Edges:** The performance difference between back-instruct with and without edges further underscores the importance of capturing tool dependencies. Including edge information in the Tool Graph allows for a more comprehensive understanding of tool relationships, resulting in more natural and complex instructions.
> > - **Alignment Scores:** It's worth noting that all methods achieve similar alignment scores. This suggests that while self-instruct may produce less natural or complex instructions, it still manages to generate instructions that align with the intended tool usage.
> > - **Prompt Comparison:** We appreciate the suggestion to provide a comparison of prompts used for different baselines. We will include full prompt comparisons in the appendix of our revised manuscript. Here, we provide a brief overview of the key differences:
> >   - **Back-Instruct (Ours):** Our method uses a comprehensive prompt that includes both node and edge information from the Tool Graph. The prompt explicitly states:
> >
> >   ```
> >   Given a tool graph where tools serve as nodes and invocation chains between tools act as edges, the following tools (nodes) are available with their respective descriptions and input/output types: {NODES}. These tools can be connected as follows, where the directed edges represent invocation chains between tools:  {EDGES}
> >   [... additional requirements and output format ...]
> >   ```
> >
> >   This approach ensures that the generated instructions and tool invocation graphs are consistent with the complex dependencies represented in the Tool Graph.
> >
> >   - **Back-Instruct w/o edges:** This baseline uses a similar prompt to our full method, but omits the edge information:
> >
> >   ```
> >   Given a tool graph where tools serve as nodes, the following tools (nodes) are available with their respective descriptions and input/output types: {NODES}
> >   [... additional requirements and output format ...]
> >   ```
> >
> >   By comparing the results of this baseline with our full method, we can isolate the impact of including edge information in the prompt.
> >
> >   - **Self-Instruct:** Instead of providing a Tool Graph, it uses demonstrations and descriptions of all tools. The prompt for this method includes:
> >
> >   ```
> >   You are an AI assistant tasked with generating instructions and tool invocation graphs. Here are descriptions of available tools: {TOOLS} and demonstrations {DEMONSTRATIONS}  Please generate a user instruction and the corresponding tool invocation graph using these tools.
> >   [... additional requirements and output format ...]
> >   ```
> >
> >   This prompt does not provide explicit information about tool dependencies, relying instead on the model's ability to infer relationships between tools based on their descriptions.
> >
> > In the revised paper, we will expand our discussion of these results, providing a more in-depth analysis of the factors contributing to the performance differences between methods. We will also emphasize how the Tool Graph and the inclusion of edge information in back-instruct lead to more natural and complex instructions, which better reflect real-world task automation scenarios. Thank you for bringing this to our attention. Your feedback will help us present a more insightful analysis of our methods and results.

---

> > > ### Author Rebuttal · Authors · 2024-08-28
> > >
> > > > The use of Rouge scores for evaluating Task Decomposition in Section 3.1 is unclear. I would suggest authors to employ metrics that better capture the structure and order of tasks in sequences, which are crucial in task decomposition.
> > >
> > > We appreciate the reviewer's insightful comment regarding the use of ROUGE scores for evaluating Task Decomposition in Section 3.1. We acknowledge that while ROUGE is effective in assessing content overlap between generated and reference task decompositions, it may not fully capture the structural and sequential aspects of tasks that are crucial in task decomposition.
> > > However, we would like to emphasize that our evaluation framework goes beyond ROUGE scores to capture the structure and order of tasks in sequences. Specifically:
> > >
> > > 1. **Tool Invocation Graph Evaluation:** Our approach maps task decomposition to a tool invocation graph, which inherently represents the structure and sequence of tasks. By evaluating the nodes and edges of this graph, we directly assess the structural and sequential aspects of task decomposition.
> > > 2. **Node F1 Score:** This metric evaluates the accuracy of identifying the correct tools (nodes) in the task sequence. It captures whether the model has correctly identified all necessary subtasks in the decomposition.
> > > 3. **Edge F1 Score:** This is particularly crucial for assessing the sequential nature of tasks. By evaluating the edges in the tool invocation graph, we directly measure the model's ability to capture the correct order and dependencies between subtasks.
> > > 4. **Normalized Edit Distance (NED):** For chain structures, we use NED to quantify the number of operations required to transform the predicted task sequence into the reference sequence. This directly addresses the sequential aspect of task decomposition.
> > > 5. **Graph Accuracy:** By evaluating the exact match of the entire graph structure, we assess whether the model has captured both the content and the structure of the task decomposition correctly.
> > >
> > > These metrics, when combined, provide a comprehensive evaluation of not just the content, but also the structure and order of tasks in the decomposed sequences. They allow us to assess:
> > >
> > > - The correctness of identified subtasks (nodes)
> > > - The proper sequencing of these subtasks (edges)
> > > - The overall structural similarity to the reference decomposition (graph accuracy)
> > >
> > > Furthermore, this approach aligns closely with real-world task automation scenarios, where the correct identification of subtasks and their dependencies is crucial for successful execution.
> > > We appreciate the reviewer's suggestion and agree that clearly articulating the strengths of our graph-based evaluation in capturing task structure and order would strengthen our paper. In the revised version, we will elaborate on how our multi-faceted evaluation approach, centered around the tool invocation graph, effectively addresses the structural and sequential aspects of task decomposition.

---

> > ### Author Rebuttal · Authors · 2024-08-28
> >
> > > While the authors mention plans to expand the benchmark to more domains and develop more advanced metrics, they do not adequately discuss the current limitations of their benchmark framework. A more thorough discussion on limitations and potential future works is needed.
> >
> > Thank you for this valuable feedback. We acknowledge that our discussion of limitations and future work could have been more comprehensive. We will address this in the revised manuscript by including a more thorough discussion of the current limitations of TaskBench and our plans for future work. Here's an expanded response:
> >
> > **Current Limitations of TaskBench:**
> >
> > - **Domain Coverage:** While TaskBench covers multiple domains, it may not yet fully represent the entire spectrum of real-world task automation scenarios. Some specialized or emerging domains may be underrepresented.
> > - **Tool Diversity:** Although we strive for a diverse set of tools, the current version of TaskBench may not encompass all types of tools that exist in rapidly evolving fields such as AI and software development.
> > - **Synthetic Data Nature:** Despite our efforts to ensure high quality through multiple verification steps, the synthetic nature of our data may still introduce some biases or artifacts not present in real-world scenarios.
> > - **Evaluation of Long-term Planning:** The current framework may not fully capture the long-term planning capabilities required for complex, multi-step tasks that span extended periods.
> > - **Human-in-the-loop Scenarios:** TaskBench currently does not extensively evaluate scenarios where human intervention or feedback is part of the task automation process.
> >
> > **Future Work:**
> >
> > - **Expanding Domain Coverage:** We plan to systematically incorporate a wider range of domains, including fields like software development,  scientific computation and advanced manufacturing.
> > - **Enhancing Tool Diversity:** We will continually update our tool repository to include new and evolving tools, ensuring TaskBench remains relevant in a rapidly changing technological landscape.
> > - **Real-world Data Integration:** To complement our synthetic data, we aim to incorporate real-world task automation examples, striking a balance between controlled evaluation and real-world applicability.
> > - **Advanced Long-term Planning Metrics:** We will develop new metrics and evaluation methods to better assess LLMs' capabilities in long-term planning and multi-step task execution.
> > - **Interactive Evaluation Framework:** We plan to introduce scenarios that involve human-in-the-loop interactions, evaluating LLMs' ability to collaborate with humans in task automation.
> > - **Continual Learning Assessment:** We aim to develop methods to evaluate LLMs' ability to adapt to new tools and domains over time, simulating real-world scenarios where new technologies are constantly introduced.
> > - **Ethical and Safety Considerations:** We will incorporate evaluation metrics that assess the ethical decision-making and safety considerations of LLMs in task automation scenarios.
> >
> > By addressing these limitations and pursuing these future directions, we aim to evolve TaskBench into an even more comprehensive, relevant, and accessible benchmark for evaluating LLMs in task automation. We appreciate the reviewer's insight, which has helped us articulate a clearer roadmap for the continued development of TaskBench.

---

> ### Author Rebuttal · Authors · 2024-08-28
>
> To illustrate the unique features and advantages of TaskBench, we have compiled a comprehensive comparison table:
>
> | | **TaskBench (Ours)** | **APIBench [1]** | **ToolBench [2]** | **Metatool [3]** | **AgentBench [4]** | **ToolQA [5]** | **GPT4Tools [6]** | **ToolAlpaca [7]** | **API-Bank [8]** |
> | --- | --- | --- | --- | --- | --- | --- | --- | --- | --- |
> | Data Generation Method | Tool Graph + Back-Instruct | API Doc | Manual + LLM | Template Generation (Direct Diverse, Keyword, Emotional, Details) | Multi-environment Simulation | Template + Programmatic Generation | Self-instruction | Multi-agent Simulation | API Doc |
> | Tool Dependency Modeling | ✓ | ✗ | ✗ | ✗ | Partial | ✗ | ✗ | Partial | ✗ |
> | Quality Control Mechanism | LLM Self-critique + Rule-based Self-critique + Human Verification | Human Verification | Human Verification | Human Verification | Human Verification | Programmatic Verification | Self-critique | Multi-agent Verification | Auto-generation + Human Verification |
> | Multi-dimensional Evaluation | Task Decomposition + Tool Selection + Parameter Prediction | Tool Selection | Tool Selection + Parameter Prediction |  Tool Usage Awareness + Tool Selection | Environment-specific metrics | QA Correctness | Procedure, response, overall | Procedure, response, overall | API call correctness, response quality |
> | Tool Dependency Assessment | ✓ | ✗ | ✗ | ✗ | ✗ | ✗ | ✗ | ✗ | ✗ |
> | Evaluation Metrics | Rouge, Node/Edge F1, Normalized Edit Distance, etc. | Accuracy, Hallucination Error Rate | Accuracy, human evaluation | Accuracy, F1, CSR, etc. | Success Rate, Reward Score, F1 | Accuracy | SRt, SRact, SRargs, SR | GPT-4 based evaluation| Accuracy, ROUGE-L |
> | Domain Diversity | High (Multi-domain) | Medium (API focus) | High (16000+ APIs) | Low (Focused on tool selection) | Medium (8 environments) | Low (8 domains) | Medium (Visual tools) | Medium (50 categories) | High (1000+ domains) |
> | Tool Complexity | Single tool to complex tool graph structures | Mainly single tool | Single to multi-tool | Single to multi-tool | Environment-dependent | Single to multi-tool | Visual tool chain | Single to multi-tool | Single to multi-tool |
> | Scalability | High (adaptable Tool Graph) | Medium (Limited by API collection) | High (large API set) | Low (Limited by manual curation) | Medium (Limited by environment design) | Low (Limited by domain coverage) | Medium (Limited by tool types) | Medium (Limited by simulation complexity) | High (large domain coverage) |
> | Dataset Scale | 17,331 samples | 2,365 samples  | 12,657 samples  | 21,127 samples | 8 environments | 55,800 questions | 652 samples | 3,938 instances | 6,135 interactions |
> | Unique Feature | Tool Graph Structure | AST Sub-tree Matching | Large-scale API Coverage | Tool Selection Decision | Diverse Environment Simulation | Cross-domain QA | Visual Tool Focus | Multi-agent Generation | Multi-level API Calling |
>
> In summary, TaskBench provides a more comprehensive, reliable, and practically relevant benchmark for evaluating LLM capabilities in task automation through its innovative data generation method, multi-level quality control mechanisms, comprehensive evaluation framework, and fine-grained evaluation metrics. These features distinguish TaskBench among existing tool-augmented LLM benchmarks, not only providing higher quality evaluation data but also offering a scalable and reproducible evaluation framework for future research.
>
> [Reference]
>
> [1] Patil, Shishir G., et al. "Gorilla: Large language model connected with massive apis." arXiv preprint arXiv:2305.15334 (2023).
>
> [2] Qin, Yujia, et al. "Toolllm: Facilitating large language models to master 16000+ real-world apis." arXiv preprint arXiv:2307.16789 (2023).
>
> [3] Huang, Yue, et al. "Metatool benchmark for large language models: Deciding whether to use tools and which to use." arXiv preprint arXiv:2310.03128 (2023).
>
> [4] Liu, Xiao, et al. "Agentbench: Evaluating llms as agents." arXiv preprint arXiv:2308.03688 (2023).
>
> [5] Zhuang, Yuchen, et al. "Toolqa: A dataset for llm question answering with external tools." Advances in Neural Information Processing Systems 36 (2024).
>
> [6] Yang, Rui, et al. "Gpt4tools: Teaching large language model to use tools via self-instruction." Advances in Neural Information Processing Systems 36 (2024).
>
> [7] Tang, Qiaoyu, et al. "Toolalpaca: Generalized tool learning for language models with 3000 simulated cases." arXiv preprint arXiv:2306.05301 (2023).
>
> [8] Li, Minghao, et al. "Api-bank: A comprehensive benchmark for tool-augmented llms." arXiv preprint arXiv:2304.08244 (2023).

---

> ### Author Rebuttal · Authors · 2024-08-28
>
> > What's the advantage of these new benchmark components? The paper would benefit from a more explicit comparison between TaskBench and existing relevant benchmarks for tool-augmented LLMs and task automation. Specifically, I would suggest authors to provide a clear comparison of TaskBench's components (e.g., Back-Instruct method, Tool Graph, TaskEval) with those of existing benchmarks, explain why each new component is necessary and how it improves upon existing benchmark designs. This would help researchers better understand the specific contributions of TaskBench, its limitations, and potential future research directions.
>
> Thank you for your suggestion. We appreciate the opportunity to clarify the advantages of TaskBench's components. We will provide a clear comparison of TaskBench's key components with those of existing benchmarks, explaining their necessity and improvements:
>
> - **Back-Instruct Method:**
>    - **Advantage:** Ensures logical consistency between generated instructions and tool dependencies.
>    - **Comparison:** Unlike self-instruct or few-shot learning methods used in ToolBench [2] or ToolAlpaca [7], Back-Instruct leverages a pre-defined tool structure.
>    - **Necessity:** Reduces hallucinations and improves data quality by anchoring instructions to realistic tool combinations.
>    - **Improvement:** Produces more logically consistent and realistic task scenarios compared to template-based or random generation methods.
> - **Tool Graph:**
>    - **Advantage:** Captures complex dependencies between tools, reflecting real-world task scenarios.
>    - **Comparison:** APIBench [1], ToolBench [2], API-Bank [8], MetaTool [3]  and ToolAlpaca [7] rely on individual API descriptions without modeling inter-tool relationships.
>    - **Necessity:** Essential for evaluating LLMs' ability to understand and navigate complex tool ecosystems.
>    - **Improvement:** Enables generation of more realistic multi-step tasks and assessment of LLMs' comprehension of tool interdependencies.
> - **TaskEval Framework:**
>    - **Advantage:** Provides a multi-dimensional assessment of LLM performance in task automation.
>    - **Comparison:** Most benchmarks (e.g., ToolQA [5], GPT4Tools [6], APIBench [1], AgentBench[3]) focus on single aspects like response quality, tool execution or overall task completion.
>    - **Necessity:** Offers a more comprehensive assessment of LLMs' capabilities in task automation.
>    - **Improvement:** Evaluates task decomposition, tool selection, and parameter prediction separately, providing deeper insights into model performance.
> - **Quality Control Mechanisms:**
>    - **Advantage:** Multi-level approach combining LLM-based and rule-based self-critics with human verification.
>    - **Comparison:** More robust than single-method approaches used in benchmarks like AgentBench [4] or API-Bench [1].
>    - **Necessity:** Ensures high-quality, realistic data crucial for meaningful LLM evaluation.
>    - **Improvement:** Results in a dataset with higher real-world relevance and reduced artificial biases.
> - **Diverse Domains and Tool Complexities:**
>    - **Advantage:** Covers multiple domains and varying levels of tool interaction complexity.
>    - **Comparison:** Broader in scope than specialized benchmarks like ToolQA [5] or GPT4Tools [6].
>    - **Necessity:** Allows for a more comprehensive evaluation of LLM generalization capabilities.
>    - **Improvement:** Provides insights into LLM performance across different application areas and complexity levels.
>
> These components collectively address limitations in existing benchmarks:
>
> - **Limited tool interaction modeling:** TaskBench's Tool Graph and Back-Instruct method enable more realistic multi-tool task scenarios.
> - **Oversimplified evaluation:** TaskEval and our new metrics provide a more comprehensive assessment of LLM capabilities in task automation and tool use.
> - **Quality control scalability:** Our multi-level approach allows for efficient generation of high-quality data at scale.
>
> Future research directions enabled by TaskBench include:
>
> - Investigating LLMs' ability to reason about tool dependencies in complex workflows.
> - Developing models that excel in multi-step task planning and execution.
> - Exploring transfer learning between different tool domains leveraging the Tool Graph structure.
>
> By addressing these aspects, TaskBench provides a more holistic and challenging evaluation framework, pushing the boundaries of LLM capabilities in task automation.
>
> [Reference]
>
> [1] Patil, Shishir G., et al. "Gorilla: Large language model connected with massive apis." arXiv preprint arXiv:2305.15334 (2023).
>
> [2] Qin, Yujia, et al. "Toolllm: Facilitating large language models to master 16000+ real-world apis." arXiv preprint arXiv:2307.16789 (2023).
>
> [3] Huang, Yue, et al. "Metatool benchmark for large language models: Deciding whether to use tools and which to use." arXiv preprint arXiv:2310.03128 (2023).
>
> [4] Liu, Xiao, et al. "Agentbench: Evaluating llms as agents." arXiv preprint arXiv:2308.03688 (2023).
>
> [5] Zhuang, Yuchen, et al. "Toolqa: A dataset for llm question answering with external tools." Advances in Neural Information Processing Systems 36 (2024).
>
> [6] Yang, Rui, et al. "Gpt4tools: Teaching large language model to use tools via self-instruction." Advances in Neural Information Processing Systems 36 (2024).
>
> [7] Tang, Qiaoyu, et al. "Toolalpaca: Generalized tool learning for language models with 3000 simulated cases." arXiv preprint arXiv:2306.05301 (2023).
>
> [8] Li, Minghao, et al. "Api-bank: A comprehensive benchmark for tool-augmented llms." arXiv preprint arXiv:2304.08244 (2023).

---

> ### Author Response · Authors · 2024-08-29
> **Looking forward to receiving your feedback**
>
> Dear Reviewer,
>
> Thank you for your insightful and constructive feedback on our TaskBench paper. We greatly appreciate the time and effort you've invested in reviewing our work. Your comments have been invaluable in helping us identify areas for improvement and clarification.
>
> In response to your feedback, we have addressed the following key points:
>
> 1. **Comparison to relevant benchmarks:** We've provided a comprehensive comparison of TaskBench with other tool-augmented LLM benchmarks, highlighting our unique contributions such as the Tool Graph and Back-Instruct method.
> 2. **Advantages of the proposed components:** We've elaborated on the benefits of our key components, including improved data quality, more realistic task scenarios, and a more comprehensive evaluation framework.
> 3. **Self-instruct vs. Back-instruct:** We've provided a comparison of the prompts used for different baselines and highlighted the key differences. We've explained the performance differences, emphasizing how our method leverages structured tool information to generate more natural and complex instructions.
> 4. **Task Decomposition evaluation:** We've clarified that our evaluation goes beyond ROUGE scores, using metrics like Node/Edge F1 scores and Normalized Edit Distance to capture the structure and order of tasks.
> 5. **Limitations and future work:** We've provided a more thorough discussion of current limitations and our plans for future improvements, including expanding domain coverage, enhancing tool diversity, and developing advanced evaluation metrics.
>
> We believe these clarifications and additions significantly strengthen our paper. **We would greatly appreciate any further feedback or suggestions you might have.** If there are any aspects you feel we haven't adequately addressed or areas that could benefit from further explanation, please let us know.
> Thank you again for your valuable input in helping us improve the quality and clarity of our research.
>
> Best regards,
>
> Submission932 Authors

---

> ### Author Response · Authors · 2024-08-30
> **Gentle Reminder**
>
> Dear Reviewer,
>
> We hope this message finds you well. As we approach the end of the discussion period, we kindly remind you to review our responses to your valuable feedback on our paper. If you have any remaining concerns or questions, please don't hesitate to let us know. Your input is valuable to us.
>
> Thank you for your time and consideration.
>
> Best regards,
>
> Submission932 Authors

---

> > ### Comment · Reviewer_1mk2 · 2024-08-31
> > **Response to Rebuttal**
> >
> > Thank you for your thorough response.
> >
> > I recommend including the tool-augmented benchmark comparison table in the main paper for clarity.
> > Regarding metrics, I was specifically referring to the "task decomposition" step, where alternatives to ROUGE could better account for task order in decomposition. I would suggest authors to consider including other metrics for this step in future.
> >
> > Overall, authors have addressed my main concerns, and I've updated my score accordingly.

---

> > ### Author Response · Authors · 2024-08-31
> > **Thank you for your positive feedback**
> >
> > Dear Reviewer,
> >
> >
> > Thank you for your follow-up comments and for taking the time to review our responses. We greatly appreciate your feedback and are glad to hear that we have addressed your main concerns.
> >
> >
> > We agree with your recommendations:
> >
> >
> > - **Benchmark Comparison Table**: We will include the tool-augmented benchmark comparison table in the main paper as suggested. This addition will provide readers with a clear and concise overview of how TaskBench compares to existing benchmarks, highlighting our contributions more effectively.
> >
> >
> >
> > - **Task Decomposition Metrics:** We appreciate your specific feedback regarding the metrics for the task decomposition step. In light of your suggestion, we commit to exploring and potentially incorporating additional metrics that more accurately capture the sequential nature of task decomposition in future iterations of our work. For example, we are considering implementing a new metric that combines content similarity with order preservation:
> >
> >
> >
> >    1. Content Similarity: Using sentence embeddings to compute semantic similarity between tasks, allowing for flexible matching even when task descriptions aren't identical.
> >    1. Optimal Task Matching: Employing the Hungarian algorithm to find the optimal matching between predicted and reference tasks based on their semantic similarity.
> >    1. Content Similarity Score: Calculating the mean similarity of the matched tasks using metrics like BertScore.
> >    1. Order Similarity Score: Computing a normalized score based on the position differences of matched tasks using metrics like Kendall's Tau.
> >    1. Final Score: Combining the content and order similarities, with adjustable weights (e.g., 60% content, 40% order).
> >
> >
> >
> > We're grateful for the valuable insights you've provided throughout this review process. Your feedback has been instrumental in improving the quality and clarity of our paper.
> >
> >
> > We're glad to hear that we've addressed your main concerns and appreciate your mention of updating the score. As we haven't noticed a change in the score in the review system, we would be grateful if you could kindly check to ensure it has been updated correctly.
> >
> >
> > Thank you again for your time and consideration. **We are committed to incorporating all the valuable feedback** we've received to further strengthen our paper.
> >
> >
> > Best regards,
> >
> >
> > Submission932 Authors

---

### Decision · Program_Chairs · 2024-09-26

**Decision:**

Accept (Poster)

**Comment:**

The paper presents a new benchmark (TaskBench) designed to evaluate LLMs for task automation scenarios. It consists of two components,  Back-Instruct Method which is a data generation technique aimed at producing complex, realistic instructions for task automation by leveraging tool dependencies and TaskEval which is a multi-dimensional evaluation framework that assesses task decomposition, tool selection, and parameter prediction capabilities of LLMs. There are some concerns raised by the reviewers such as the comparison with traditional benchmarks (such as GLUE, SuperGLUE) and overlooking more relevant LLM benchmarks for task automation (such as API-Bench, ToolBench, MetaTool). Some reviewers also raised concerns regarding the synthetic nature of the dataset and the use of ROUGE scores in task decomposition. The authors have done an extremely thorough job in responding to the reviewers' concerns. They provided detailed responses including several experiments addressing the weaknesses raised by the reviewers. The authors will need to include all the information in the final camera ready submission if the paper gets accepted.